# Mapping routine measles vaccination in low- and middle-income countries

Local Burden of Disease Vaccine Coverage Collaborators*

The safe, highly effective measles vaccine has been recommended globally since 1974, yet in 2017 there were more than 17 million cases of measles and 83,400 deaths in children under 5 years old, and more than 99% of both occurred in low- and middle-income countries (LMICs)[1–4]. Globally comparable, annual, local estimates of routine first-dose measles-containing vaccine (MCV1) coverage are critical for understanding geographically precise immunity patterns, progress towards the targets of the Global Vaccine Action Plan (GVAP), and high-risk areas amid disruptions to vaccination programmes caused by coronavirus disease 2019 (COVID-19)[5–8]. Here we generated annual estimates of routine childhood MCV1 coverage at $5 \times 5$-km$^2$ pixel and second administrative levels from 2000 to 2019 in 101 LMICs, quantified geographical inequality and assessed vaccination status by geographical remoteness. After widespread MCV1 gains from 2000 to 2010, coverage regressed in more than half of the districts between 2010 and 2019, leaving many LMICs far from the GVAP goal of 80% coverage in all districts by 2019. MCV1 coverage was lower in rural than in urban locations, although a larger proportion of unvaccinated children overall lived in urban locations; strategies to provide essential vaccination services should address both geographical contexts. These results provide a tool for decision-makers to strengthen routine MCV1 immunization programmes and provide equitable disease protection for all children.

The safe, highly effective vaccine against measles has been recommended since 1974 by the Expanded Programme on Immunization of the WHO (World Health Organization)[1–3]. A single valid dose of any MCV is approximately 93% effective in providing individuals with lifelong protection from measles[1]. Still, in 2017, there were an estimated 17,767,037 new global cases and 83,439 deaths attributable to measles in children under 5 years old[4]. Although high-income regions, such as the USA and Europe, have recently started experiencing extended measles outbreaks due to a lapse in vaccination coverage, more than 99% of cases and deaths still occur in LMICs[4,9].

Low vaccination rates and increasing vaccine hesitancy[10–12] contribute to the persistence of measles as a major cause of childhood morbidity and mortality. National-level MCV1 estimates from the Global Burden of Diseases, Injuries and Risk Factors Study (GBD) 2019 identified only 72 out of 204 countries in which routine coverage reached approximate herd immunity targets (≥95%) in 2019, and global MCV1 coverage[4,13] was 84.2%. Even in countries with high national coverage, these estimates mask important subnational heterogeneities that may sustain ongoing disease transmission and increase the risk of outbreaks[14–17], especially in light of the current service disruptions associated with the COVID-19 pandemic[7,8]. Global vaccination initiatives, such as the GVAP and Immunization Agenda 2030, recognize the importance of eliminating subnational coverage disparities, aiming for at least 90% of the target population in every country and 80% in every district to be covered[5,6].

## Subnational routine MCV1 coverage

Since 2016, the WHO and UNICEF have collected subnational coverage data through their annual Joint Reporting process, although poor data quality and biases currently limit the use of administrative data to track progress towards GVAP targets[18–20]. For the 101 countries included in this study, 91 reported subnational data in 2018 in a total of 11,311 subnational geographical units. Of these countries, 71 reported MCV1 coverage greater than 100% in at least one unit and 55 reported such coverage in at least a quarter of units. Although researchers have estimated subnational MCV1 coverage in select countries or years for which there have been reliable surveys, to date, no comprehensive analysis of all available vaccine coverage data to produce subnational estimates of MCV1 coverage annually in all LMICs has been undertaken[21–24].

Building from our previous work mapping diphtheria–tetanus–pertussis vaccine coverage in Africa[14], here we present mapped high-spatial-resolution estimates of routine MCV1 coverage across 101 LMICs from 2000 to 2019, aggregated to policy-relevant second-level administrative units (hereafter districts). Using geolocated data on MCV1 coverage from 354 household-based surveys representing approximately 1.70 million children and a suite of environmental, sociodemographic and health-related geospatial and national-level covariates, we extended model-based geostatistical methods that have been used to map child mortality and its main components and risk factors[25–28] to predict MCV1 coverage and uncertainty

*A list of participants and their affiliations appears in the online version of the paper.

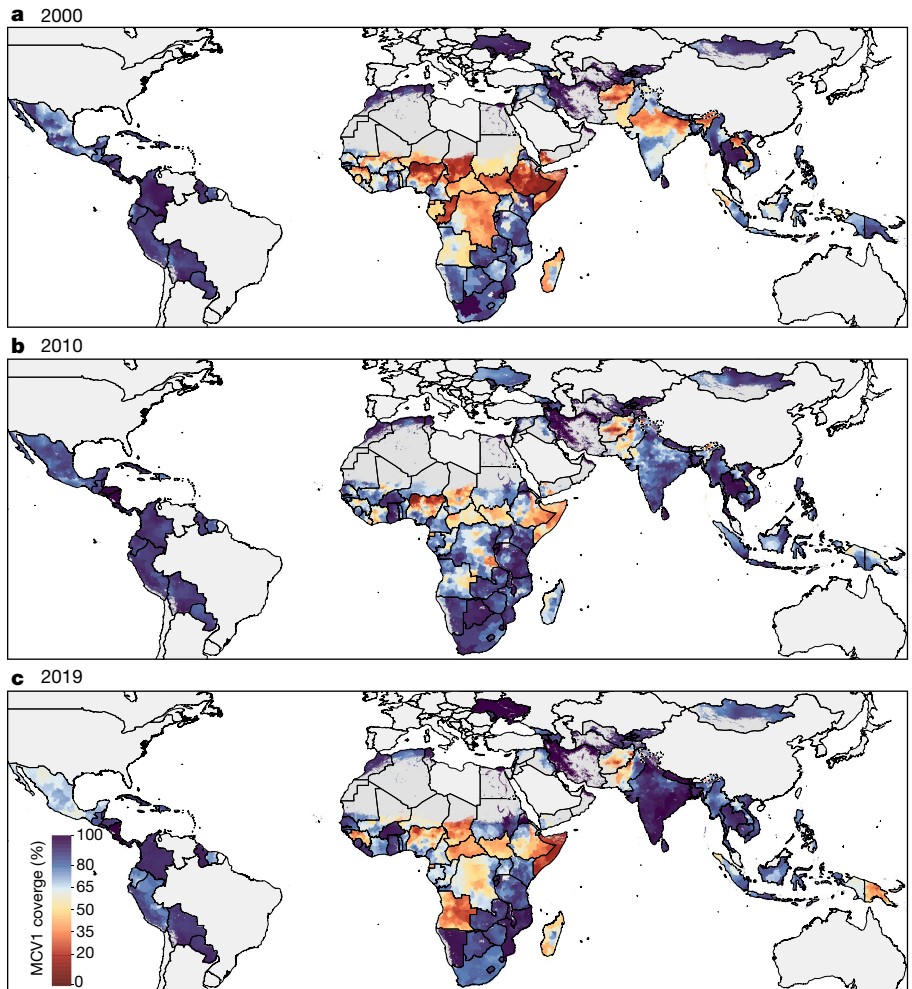

**Fig. 1 | Estimated MCV1 coverage among districts in 101 LMICs, 2000–2019.** **a–c**, MCV1 coverage among target population in districts in 2000 (**a**), 2010 (**b**) and 2019 (**c**). Countries excluded from the analysis and pixels classified as 'barren or sparsely vegetated' based on European Space Agency Climate Change Initiative (ESA-CCI) satellite data or with fewer than 10 people per 1 × 1-km² pixel based on WorldPop estimates are masked in grey[30,50].

(Extended Data Figs. 1, 2), while calibrating estimates to results from GBD 2019. Using these estimates, we assessed trends in geographical inequality, progress towards global targets and differential vaccination status by geographical remoteness.

## Tracking uneven progress

Despite marked global progress between 2000 and 2019, considerable inequalities in routine MCV1 coverage persist, both between and within countries (Fig. 1, Extended Data Figs. 3–7, see also our visualization tool (https://vizhub.healthdata.org/lbd/mcv)). MCV1 coverage among children living in the 101 countries included in this study was 65.6% (95% uncertainty interval, 64.2–67.1%) in 2000 and 81.0% (95% uncertainty interval, 79.2–82.7%) in 2019. Coverage increased at the national level in 69.9% (95% uncertainty interval, 64.4–75.2%) of countries between 2000 and 2019 and in 57.4% (95% uncertainty interval, 50.4–64.6%) of districts (*n* = 20,795 districts).

The three districts with the lowest MCV1 coverage in 2000 were located in Hari Rasu, Ethiopia (4.0% (95% uncertainty interval, 1.1–9.7%)), Gabi Rasu, Ethiopia (4.8% (95% uncertainty interval, 1.4–11.4%)), and Isa, Nigeria (4.9% (95% uncertainty interval, 1.5–10.8%)). In 2000, 60 districts had a coverage below 10%; there was one such district in 2019. The three lowest-coverage districts in 2019 were all located in Afghanistan: Poruns (9.2% (95% uncertainty interval, 2.0–25.5%)),

Wama (12.1% (95% uncertainty interval, 2.8–32.6%)) and Waygal (12.7% (95% uncertainty interval, 3.0–34.2%)).

In the period from 2000 to 2010, there were substantial increases in coverage and progress towards reducing subnational heterogeneity. The period from 2010 to 2019, however, showed slowing progress and, in some cases, regression of coverage compared to the 2000–2010 period (Fig. 2). Between 2000 and 2010, 70.5% (95% uncertainty interval, 66.0–75.4%) of districts increased coverage, but between 2010 and 2019, coverage increased in only 40.1% (95% uncertainty interval, 34.2–46.9%) of districts (Extended Data Fig. 8). This finding persists even when accounting for the starting level of coverage (Supplementary Information section 2.3).

Although district-level MCV1 coverage generally increased between 2000 and 2019, further gains are required to reach both 80% and 95% key coverage targets (Extended Data Fig. 9). In 2000, 38.4% of districts had a high probability (>95% posterior probability) of reaching the GVAP target of 80% district-level MCV1 coverage, which remained stagnant at 33.2% of districts in 2019. Only 15 countries had a high probability of reaching the GVAP target of greater than 80% district-level coverage in all districts.

## Quantifying geographical inequalities

To further assess the effect of geographical heterogeneity in MCV1 coverage, we computed the absolute geographical inequality,

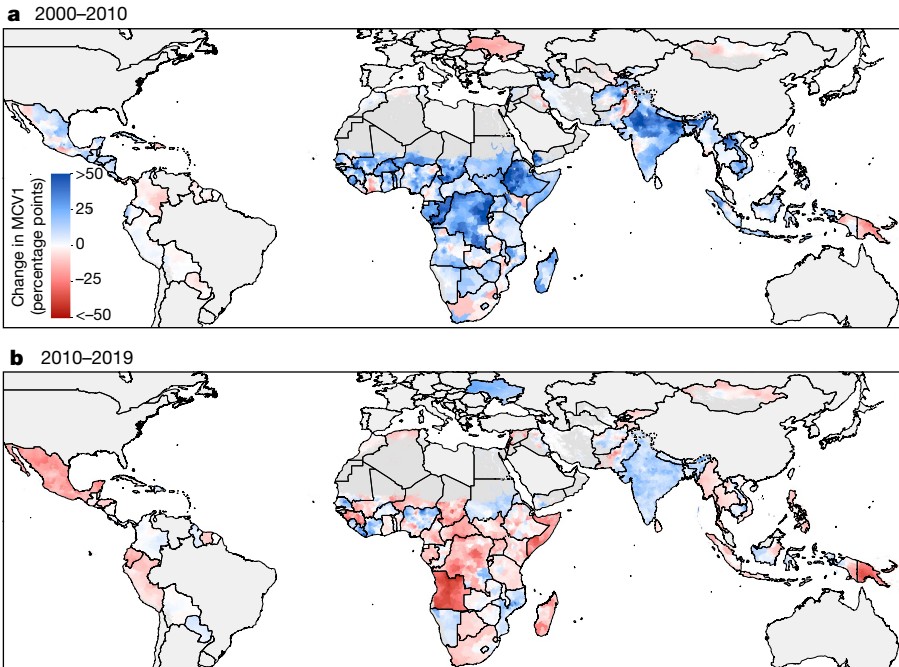

**a** 2000–2010

Change in MCV1
(percentage points)

>50
25
0
−25
<−50

**b** 2010–2019

**Fig. 2 | Estimated absolute changes in MCV1 coverage in the early (2000–2010) and late (2010–2019) study periods. a, b**, Mean district-level absolute differences in MCV1 coverage from 2000 to 2010 (**a**) and from 2010 to 2019 (**b**). Countries excluded from the analysis and pixels classified as 'barren or sparsely vegetated' based on ESA-CCI satellite data or with fewer than 10 people per 1 × 1-km[2] pixel based on WorldPop estimates are masked in grey[30,50].

a Gini-coefficient-derived metric that ranges between zero (perfectly homogenous coverage) and one (perfectly heterogeneous coverage), at the 5 × 5-km[2] level. In 2000, nine countries (Cameroon, Democratic Republic of the Congo, Guinea, India, Laos, Madagascar, Mali, Nigeria and Yemen) had high absolute geographical inequality (above 0.15). In 2019, only five countries had high absolute geographical inequality (Angola, Ethiopia, Madagascar, Nigeria and Pakistan). Nigeria had absolute geographical inequality above or equal to 0.2 in both 2000 and 2019, and 25 countries had increased absolute geographical inequality in 2019 compared with 2000. Notably, absolute geographical inequality decreased considerably in India, from 0.23 in 2000 to 0.07 in 2019.

In general, and as expected, improvements in national-level coverage over time were accompanied by reductions in subnational absolute geographical inequality (Fig. 3). Changes in coverage were negatively correlated ($\rho = -0.47$, Pearson's product moment test statistic = −5.36, $P < 0.001$) with changes in absolute inequality. India is a true exemplar in this trend, with sizeable reductions in inequality occurring as coverage increased. This improvement was not the only pathway for a country, however; some countries with increasing coverage also experienced increasing inequality, such as Chad and Ethiopia. Other countries experienced decreasing coverage alongside increasing inequality, such as Angola.

## Coverage in urban and rural areas

In a post hoc analysis, we compared vaccination status in urban and remote rural settings, using proxies of travel time of ≤30 min and ≥3 h, respectively, to the nearest major city or settlement[29] and the number of children under 5 years old[30] from gridded estimates. In 2019, MCV1 coverage was relatively lower in remote rural areas: in remote rural locations, 33.3% of children were MCV-unvaccinated, compared with 15.2% of children living in urban areas. Owing to the concentration of populations in urban centres, however, more unvaccinated children lived in urban locations (47.9% of all unvaccinated children) than remote rural areas (16.0% of all unvaccinated children) in 2019, although this pattern varied across countries and regions (Fig. 4). For example, in

Chad, 19.3% of unvaccinated children lived in urban locations and 44.4% lived in remote rural locations in 2019. In Mexico in 2019, 72.3% of unvaccinated children lived in urban locations and 3.4% lived in remote rural locations (Extended Data Fig. 10).

Our results show the variability of urban and rural contributions to unvaccinated populations in each country and region. In Latin America and the Caribbean, for example, MCV1 coverage is generally similar between urban and rural settings (Fig. 4); the urban–remote rural distribution of unvaccinated children therefore largely reflects the underlying population distribution. In other regions, the interaction between population and coverage is more complex. In South Asia, for example, 21 times more unvaccinated children live in urban locations compared with remote rural locations. Strategies focused solely on reaching the most unvaccinated children possible in that region, therefore, might reasonably prioritize urban areas. Overall, however, MCV1 coverage in urban areas of South Asia averages 90.7%, compared to only 77.4% in remote rural areas. Approaches that focus primarily on reaching urban areas, therefore, would probably exacerbate existing urban–rural coverage inequalities.

## Discussion

Our MCV1 coverage estimates show overall progress from 2000 to 2019, corresponding to the creation of benchmark targets from the Measles and Rubella Initiative and GVAP, as well as substantial financial support for comprehensive vaccination programming generated by the introduction of Gavi, the Vaccine Alliance[5,6,31–34]. Moreover, 62 out of 101 countries increased national-level MCV1 coverage while reducing subnational geographical inequalities over time, a noteworthy achievement.

This remarkable global progress should be celebrated, but this trend was not universal. Our results show a decline and stagnation in routine MCV1 coverage in certain locations, particularly since 2010, that may be related to conflict, vaccine scepticism, available funding support and supply disruptions[35]. Among countries with stagnant or declining coverage rates, the Central African Republic and Nigeria are experiencing

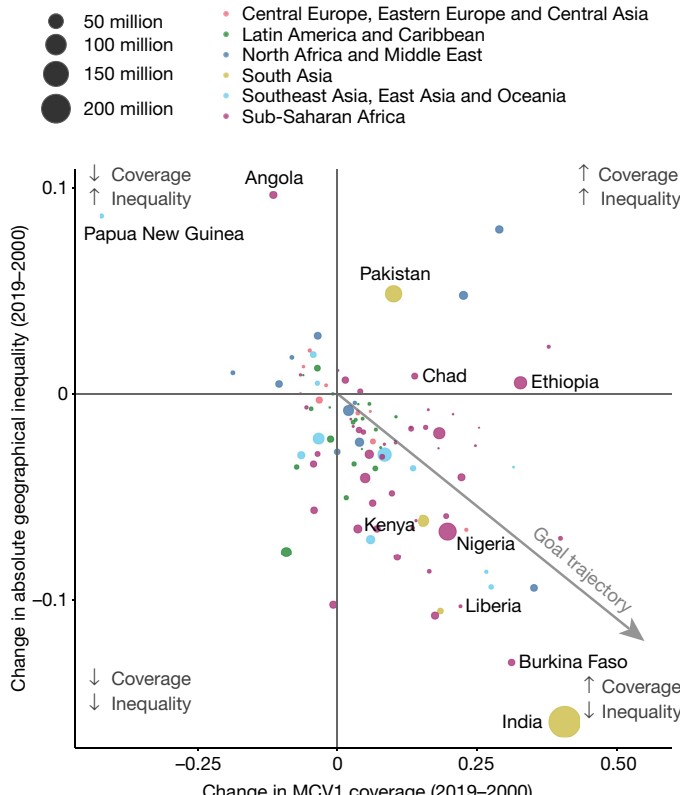

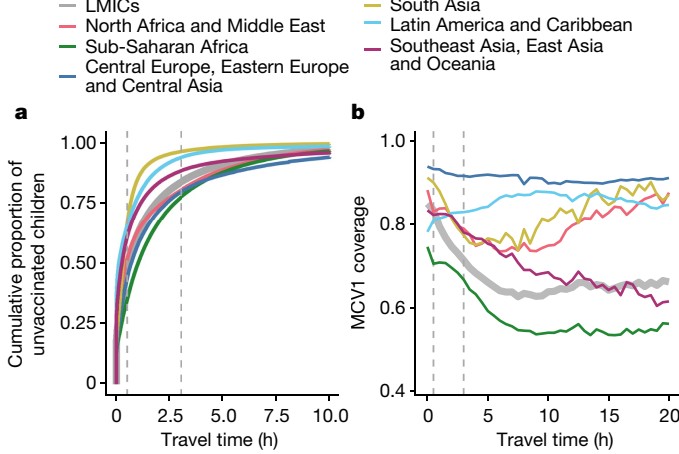

**Fig. 3 | Absolute geographical inequality of MCV1 coverage in 2000 and 2019.** We compared the change in geographical absolute inequality to change in national-level coverage from 2000 to 2019. Points are sized by under-5 population size.

**Fig. 4 | Vaccination status in 2019 and geographical remoteness.** Cumulative proportion of unvaccinated children living within the spectrum of the travel time (in hours) to a major city or settlement per region (left) and coverage among children living within the spectrum of travel time to a major city or settlement per region (right). Vertical dashed grey line shows thresholds for 'urban' and 'remote rural', living within 30 min and at least 3 h from a major city or settlement, respectively.

ongoing political instability and conflict, which serve as major barriers to the success of vaccination programmes[36–38]. Supply disruptions also present a major barrier to achieving and sustaining high levels of MCV coverage. For example, in 2018, the Philippines reported a national-level MCV stockout[39]. The stockout, alongside waning public confidence in vaccination programmes stemming from misinformation related to risks of the Dengvaxia dengue vaccine, contributed to a national-level drop in coverage from 80% to 69% between 2017 and 2018[40]. In Angola, economic turmoil led to a 28% decrease in governmental health spending per capita between 2010 and 2018, which may have also contributed to declines in vaccination coverage[41]. While global immunization initiatives have often focused on low-income countries, districts in middle-income countries also experienced recent declines, emphasizing the need for reliable immunization programmes and monitoring in these nations[42]. In Indonesia, for instance, 3 districts had coverage that reached 95% in 2000, increasing to 13 in 2010, but decreasing back to zero in 2019. In addition, countries with higher than average vaccine scepticism, such as Peru and Moldova, also experienced decreasing coverage rates and increasing within-country inequality[43].

Overlaid on these persistent challenges, the ongoing COVID-19 pandemic has caused the cancellation of supplementary measles immunization campaigns and puts the delivery of critical routine immunization services at risk[7,8]. Baseline subnational estimates of routine MCV1 immunization in LMICs can help to define the geographical areas of highest vulnerability to pandemic-associated disruptions. To mitigate the risk of measles outbreaks in the setting of the COVID-19 pandemic, the maintenance of routine immunization services is crucial[44]—particularly in areas with pre-existing routine immunization weaknesses.

Even before the current pandemic, few countries reached the GVAP target of 80% coverage in all districts by 2019. Stagnant progress

between 2010 and 2019 further suggests that new approaches are needed to reach unvaccinated children and resolve geographical inequalities. As the era of GVAP draws to a close and the Immunization Agenda 2030 begins, these results provide a platform to identify successes and inform strategies for the next decade. India, for instance, saw exemplary improvement in both national-level coverage and geographical equality over time. This may be attributable to specific interventions such as Mission Indradhanush, launched in 2014 with the goal of targeting underserved unvaccinated populations[45]. In addition, India introduced a second dose of MCV (MCV2) in select subnational geographies with MCV1 coverage below 80% starting in 2008, and expanded MCV2 to cover all districts in 2010 through the strengthening of both routine and supplementary immunization programmes[46,47]. The introduction of MCV2 into the national schedule may provide a second opportunity for first-dose vaccination among children who missed the scheduled MCV1 dose. Understanding the specific drivers of simultaneous coverage and equality gains may provide critical insights for the immunization agenda in countries and regions that have fallen behind.

The Equity Reference Group for Immunization highlights the need for increased attention on vaccinating vulnerable children who live in remote rural, urban poor and conflict settings, as well as for equality in coverage by gender[48]. These recommendations suggest that the agenda to leave no child unvaccinated, set by global partners and the Sustainable Development Goals, should transcend geography types and aim to eliminate coverage gaps among children who live in both urban and remote rural areas[49]. These geographically resolved MCV1 estimates provide a tool for decision-makers to plan supplementary immunization activities and routine immunization strengthening programmes, to reach both the urban and remote rural communities where unvaccinated children live.

Despite large improvements made in MCV1 coverage from routine immunization programmes between 2000 and 2019, stalling progress and substantial subnational variation remain in many LMICs, leaving children at risk of preventable death. Policymakers should note where progress is most critically needed to successfully meet global immunization targets and protect the most-vulnerable children against measles. Our subnational estimates of routine MCV1 coverage at policy-relevant scales provide a tool for decision-makers to use in advocating for strong, sustainable immunization programmes that provide equitable protection for all children.

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

Local Burden of Disease Vaccine Coverage Collaborators

Alyssa N. Sbarra[1], Sam Rolfe[1], Jason Q. Nguyen[1], Lucas Earl[1], Natalie C. Galles[1], Ashley Marks[1], Kaja M. Abbas[2], Mohsen Abbasi-Kangevari[3], Hedayat Abbastabar[4], Foad Abd-Allah[5], Ahmed Abdelalim[5], Mohammad Abdollahi[6,7], Kedir Hussein Abegaz[8,9], Hailemariam Abiy Alemu Abiy[10,11], Hassan Abolhassani[12,13], Lucas Guimarães Abreu[14], Michael R. M. Abrigo[15], Abdelrahman I. Abushouk[16,17], Manfred Mario Kokou Accrombessi[18,19], Maryam Adabi[20], Oladimeji M. Adebayo[21], Victor Adekanmbi[22], Olatunji O. Adetokunboh[23,24], Davoud Adham[25], Mohsen Afarideh[26,27], Mohammad Aghaali[28], Tauseef Ahmad[29], Raman Ahmadi[30,31], Keivan Ahmadi[32], Muktar Beshir Ahmed[33,34], Fahad Mashhour Alanezi[35], Turki M. Alanzi[36], Jacqueline Elizabeth Alcalde-Rabanal[37], Birhan Tamene Alemnew[38], Beriwan Abdulqadir Ali[39,40], Muhammad Ali[41], Mehran Alijanzadeh[42], Cyrus Alinia[43], Reza Alipoor[44], Vahid Alipour[45,46], Hesam Alizade[47], Syed Mohamed Aljunid[48,49], Ali Almasi[50], Amir Almasi-Hashiani[51], Hesham M. Al-Mekhlafi[52,53], Khalid A. Altirkawi[54], Bekalu Amare[55], Saeed Amini[56], Mostafa Amini-Rarani[57], Fatemeh Amiri[58], Arianna Maever L. Amit[59,60], Dickson A. Amugsi[61], Robert Ancuceanu[62], Catalina Liliana Andrei[63], Mina Anjomshoa[64], Fereshteh Ansari[65,66], Alireza Ansari-Moghaddam[67], Mustafa Geleto Ansha[68], Carl Abelardo T. Antonio[69,70], Ernoiz Antriyandarti[71], Davood Anvari[72,73], Jalal Arabloo[45], Morteza Arab-Zozani[74], Olatunde Aremu[75], Bahram Armoon[76,77], Afsaneh K. Aryal[78], Afsaneh Arzani[79,80], Mehran Asadi-Aliabadi[81], Samaneh Asgari[82], Zahra Atafar[83], Marcel Ausloos[84,85], Nefsu Awoke[86], Beatriz Paulina Ayala Quintanilla[87], Martin Amogre Ayanore[88], Yared Asmare Aynalem[89], Abbas Azadmehr[90], Samad Azari[45], Ebrahim Babaee[81], Alaa Badawi[91,92], Shahid D. Badiye[93], Mohammad Amin Bahrami[94], Atif Amin Baig[95], Ahad Bakhtiari[96], Senthilkumar Balakrishnan[97], Maciej Banach[98,99], Palash Chandra Banik[100], Aleksandra Barac[101,102], Zahra Baradaran-Seyed[103], Adhanom Gebreegziabher Baraki[104], Sanjay Basu[105,106], Mohsen Bayati[107], Yibeltal Tebekaw Bayou[108], Neeraj Bedi[109,110], Masoud Behzadifar[111], Michelle L. Bell[112], Dessalegn Ajema Berbada[113], Kidanemaryam Berhe[114], Suraj Bhattarai[27], Rakhi Dandona[1,150,151], Lalit Dandona[1,150,151], M. Carolina Danovaro[153], Shiva Borzouei[125], Oliver J. Brady[2], Nicola Luigi Bragazzi[126], Andrey Nikolaevich Briko[127], Nikolay Ivanovich Briko[128], Sharath Burugina Nagaraja[129], Zahid A. Butt[130,131], Luis Alberto Cámera[132,133], Ismael R. Campos-Nonato[134], Josip Car[135,136], Rosario Cárdenas[137], Felix Carvalho[138], João Maurício Castaldelli-Maia[139], Franz Castro[140], Vijay Kumar Chattu[141], Mohammad Chehrazi[142,143], Ken Lee Chin[144,145], Dinh-Toi Chu[146], Aubrey J. Cook[1], Natalie Maria Cormier[1], Brandon Cunningham[1], Saad M. A. Dahlawi[147], Giovanni Damiani[148,149], Rakhi Dandona[1,150,151], Lalit Dandona[1,150,151], M. Carolina Danovaro[153], Emily Dansereau[154], Farah Daoud[1], Aso Mohammad Darwesh[155], Amira Hamed Darwish[156], Jai K. Das[157], Nicole Davis Weaver[1], Jan-Walter De Neve[158], Feleke Mekonnen Demeke[159], Asmamaw Bizuneh Demis[160,161], Edgar Denova-Gutiérrez[162], Assefa Desalew[163], Aniruddha Deshpande[1], Desilu Mahari Desta[164], Samath Dhamminda Dharmaratne[1,151,165], Govinda Prasad Dhungana[166], Mostafa Dianatinasab[167,168], Daniel Diaz[169,170], Isaac Oluwafemi Dipeolu[171], Shirin Djalalinia[171], Hoa Thi Do[172], Fariba Dorostkar[173], Leila Doshmangir[174], Kerrie E. Doyle[175,176], Susanna I. Dunachie[177,178], Andre Rodrigues Duraes[179,180], Mohammad Ebrahimi Kalan[181], Hamed Ebrahimzadeh Leylabadlo[182], Hisham Atan Edinur[183], Andem Effiong[184], Aziz Eftekhari[185,186], Iman El Sayed[187], Maysaa El Sayed Zaki[188], Teshome Bekele Elema[189], Hala Rashad Elhabashy[190], Shaimaa I. El-Jaafary[5], Aisha Elsharkawy[191], Mohammad Hassan Emamian[192], Shymaa Enany[193], Babak Eshrati[81], Khalil Eskandari[194,195], Sharareh Eskandarieh[196], Saman Esmaeilnejad[197,198], Firooz Esmaeilzadeh[199], Alireza Esteghamati[27], Atkilt Esaiyas Etisso[200], Mohammad Farahmand[201], Emerito Jose A. Faraon[69], Mohammad Fareed[202], Roghiyeh Faridnia[203], Andrea Farioli[204], Farshad Farzadfar[205], Nazir Fattahi[206], Mehdi Fazlzadeh[207,208], Seyed-Mohammad Fereshtehnejad[209,210], Eduarda Fernandes[211], Irina Filip[212,213], Florian Fischer[214], Nataliya A. Foigt[215], Morenike Oluwatoyin Folayan[216], Masoud Foroutan[217], Takeshi Fukumoto[218], Nancy Fullman[1], Mohamed M. Gad[16,219], Biniyam Sahiledengle Geberemariyam[220], Tsegaye Tewelde Gebrehiwot[33], Abiyu Mekonnen Gebrehiwot[221], Kidane Tadesse Gebremariam[222], Ketema Bizuwork Gebremedhin[223], Gebreamlak Gebremedhn Gebremeskel[224,225], Assefa Ayalew Gebreslassie[226], Getnet Azeze Gedefaw[227], Kebede Embaye Gezae[228], Keyghobad Ghadiri[229,230], Reza Ghaffari[231], Fatemeh Ghaffarifar[232], Mahsa Ghajarzadeh[233], Reza Ghanei Gheshlagh[234], Ahmad Ghashghaee[45,235], Hesam Ghiasvand[236], Asadollah Gholamian[237,238], Syed Amir Gilani[239,240], Paramjit Singh Gill[241], Alem Girmay[224], Nelson G. M. Gomes[211,242], Sameer Vali Gopalani[243,244], Bárbara Niegia Garcia Goulart[245], Ayman Grada[246], Rafael Alves Guimarães[247], Yuming Guo[144,248], Rahul Gupta[249,250], Nima Hafezi-Nejad[251,252], Arvin Haj-Mirzaian[253,254], Arya Haj-Mirzaian[251], Demelash Woldeyohannes Handiso[255], Asif Hanif[256], Hamidreza Haririan[257], Ahmed I. Hasaballah[258], Md Mehedi Hasan[259,260], Edris Hasanpoor[261], Amir Hasanzadeh[262,263], Soheil Hassanipour[264,265], Hadi Hassankhani[266,267], Reza Heidari-Soureshjani[268], Nathaniel J. Henry[1,269], Claudiu Herteliu[85,270], Fatemeh Heydarpour[271], Gillian I. Hollerich[1], Enayatollah Homaie Rad[272], Praveen Hoogar[273], Naznin Hossain[274], Mostafa Hosseini[275,276], Mehdi Hosseinzadeh[277,278], Mowafa Househ[279], Guoqing Hu[280], Tanvir M. Huda[281,282], Ayesha Humayun[283], Segun Emmanuel Ibitoye[171], Gloria Ikilezi[1], Olayinka Stephen Ilesanmi[284,285], Irena M. Ilic[102], Milena D. Ilic[286], Mohammad Hasan Imani-Nasab[287], Leeberk Raja Inbaraj[288], Usman Iqbal[289], Seyed Sina Naghibi Irvani[290], Sheikh Mohammed Shariful Islam[291,292], M. Mofizul Islam[293], Chinwe Juliana Iwu[24,294], Chidozie C. D. Iwu[295], Farhad Jadidi-Niaragh[296], Morteza Jafarinia[297], Nader Jahanmehr[298,299], Mihajlo Jakovljevic[300,301], Amir Jalali[83], Farzad Jalilian[83], Javad Javidnia[304], Ensiyeh Jenabi[305], Vivekanand Jha[306,307], John S. Ji[308,309], Oommen John[310,311], Kimberly B. Johnson[1], Farahnaz Joukar[264,265], Jacek Jerzy Jozwiak[312], Zubair Kabir[313], Ali Kabir[314], Hamed Kalani[315], Leila R. Kalankesh[316], Rohollah Kalhor[317,318], Zul Kamal[319,320], Tanuj Kanchan[321], Neeti Kapoor[93], Manoochehr Karami[322], Behzad Karami Matin[206], André Karch[323], Salah Eddin Karimi[324], Gbenga A. Kayode[325,326], Ali Kazemi Karyani[206], Peter Njenga Keiyoro[327], Yousef Saleh Khader[328], Morteza Abdullatif Khafaie[329], Mohammad Khammarnia[330], Muhammad Shahzeb Khan[331,332], Ejaz Ahmad Khan[333], Junaid Khan[334], Md Nuruzzaman Khan[335], Khaled Khatab[336,337], Mona M. Khater[7], Mahalaqua Nazli Khatib[340,341], Maryam Khayamzadeh[340,341], Mojtaba Khazaei[342], Salman Khazaei[322], Ardeshir Khosravi[343,344], Jagdish Khubchandani[345], Neda Kianipour[346], Yun Jin Kim[347], Ruth W. Kimokoti[348], Damaris K. Kinyoki[1,151], Adnan Kisa[349,350], Sezer Kisa[351], Tufa Kolola[352], Hamidreza Komaki[353,354], Soewarta Kosen[355], Parvaiz A. Koul[356],

Ai Koyanagi[357,358], Moritz U. G. Kraemer[359,360], Kewal Krishan[361], Barthelemy Kuate Defo[362,363], Manasi Kumar[364,365], Pushpendra Kumar[366], G. Anil Kumar[150], Dian Kusuma[367,368], Carlo La Vecchia[369], Ben Lacey[370,371], Sheetal D. Lad[372], Dharmesh Kumar Lal[150], Felix Lam[373], Faris Hasan Lami[374], Van Charles Lansingh[375,376], Heidi Jane Larson[1,2], Savita Lasrado[377], Shaun Wen Huey Lee[378,379], Paul H. Lee[380], Kate E. LeGrand[1], Tsegaye Lolaso Lenjebo[381], Shanshan Li[382], Xiaofeng Liang[383], Patrick Y. Liu[384], Platon D. Lopukhov[128], Daiane Borges Machado[385,386], Phetole Walter Mahasha[387], Mokhtar Mahdavi Mahdavi[76], Mina Maheri[388], Narayan B. Mahotra[389], Venkatesh Maled[390,391], Shokofeh Maleki[392], Manzoor Ahmad Malik[393,394], Deborah Carvalho Malta[395], Fariborz Mansour-Ghanaei[264,265], Borhan Mansouri[303], Morteza Mansourian[396], Mohammad Ali Mansournia[275], Francisco Rogerlândio Martins-Melo[397], Anthony Masaka[398], Benjamin K. Mayala[1,399], Man Mohan Mehndiratta[400,401], Fereshteh Mehri[402], Kala M. Mehta[403], Peter T. N. Memiah[404], Walter Mendoza[405], Ritesh G. Menezes[406], Meresa Berwo Mengesha[407], Endalkachew Worku Mengesha[408], Tomislav Mestrovic[409,410], Kebadnew Mulatu Mihretie[411], Molly K. Miller-Petrie[1], Edward J. Mills[412], George J. Milne[413], Parvaneh Mirabi[414], Erkin M. Mirrakhimov[415,416], Roya Mirzaei[417,418], Maryam Mirzaei[419], Hamid Reza Mirzaei[420], Hamed Mirzaei[421], Mehdi Mirzaei-Alavijeh[83], Babak Moazen[158,422], Masoud Moghadaszadeh[423,424], Efat Mohamadi[344], Dara K. Mohammad[425,426], Yousef Mohammad[427], Karzan Abdulmuhsin Mohammad[428], Naser Mohammad Gholi Mezerji[429], Abolfazl Mohammadbeigi[28], Abdollah Mohammadian-Hafshejani[430], Reza Mohammadpourhodki[431], Shafiu Mohammed[158,432], Ammas Siraj Mohammed[433], Hussen Mohammed[434], Farnam Mohebi[205,351], Ali H. Mokdad[1,151], Lorenzo Monasta[435], Mohammad Amin Moosavi[436], Mahmood Moosazadeh[437], Ghobad Moradi[438,439], Masoud Moradi[206], Mohammad Moradi-Joo[440], Maziar Moradi-Lakeh[81], Rahmatollah Moradzadeh[51], Paula Moraga[441], Abbas Mosapour[442,443], Simin Mouodi[118], Seyyed Meysam Mousavi[96], Amin Mousavi Khaneghah[444], Ulrich Otto Mueller[445], Atalay Goshu Muluneh[104], Sandra B. Munro[1], Christopher J. L. Murray[1,151], G. V. S. Murthy[447], Saravanan Muthupandian[448], Mehdi Naderi[392], Ahamarshan Jayaraman Nagarajan[449,450], Mohsen Naghavi[1,151], Vinay Nangia[451], Jobert Richie Nansseu[452,453], Vinod C. Nayak[454], Javad Nazari[455], Duduzile Edith Ndwandwe[456], Ionut Negoi[457,458], Josephine N. Ngunjiri[459], Huong Lan Thi Nguyen[460], Chuc T. K. Nguyen[461], Trang Huyen Nguyen[461], Yeshambel T. Nigatu[462,463], Rajan Nikbakhsh[254], Shekoufeh Nikfar[464], Amin Reza Nikpoor[465], Dina Nur Anggraini Ningrum[466,467], Chukwudi A. Nnaji[294,468], In-Hwan Oh[469], Morteza Oladnabi[470], Andrew T. Olagunju[471,472], Jacob Olusegun Olusanya[473], Bolajoko Olubukunola Olusanya[473], Ahmed Omar Bali[474], Muktar Omer Omer[475], Obinna E. Onwujekwe[476], Aaron E. Osgood-Zimmerman[1], Mayowa O. Owolabi[477,478], Mahesh P A[479], Jagadish Rao Padubidri[480], Keyvan Pakshir[481], Adrian Pana[85,482], Anamika Pandey[483], Victoria Pando-Robles[484], Tahereh Pashaei[485], Deepak Kumar Pasupula[486], Angel J. Paternina-Caicedo[487], George C. Patton[488,489], Hamidreza Pazoki Toroudi[490,491], Veincent Christian Filipino Pepito[492], Julia Moreira Pescarini[385], David M. Pigott[1,151], Thomas Pilgrim[493], Meghdad Pirsaheb[206], Mario Poljak[494], Maarten J. Postma[495,496], Hadi Pourjafar[497,498], Farshad Pourmalek[499], Reza Pourmirza Kalhori[500], Sergio I. Prada[501,502], Sanjay Prakash[503], Zahiruddin Quazi Syed[467], Hedley Quintana[140], Navid Rabiee[505], Mohammad Rabiee[506], Amir Radfar[507], Alireza Rafiei[508,509], Fakher Rahim[510,511], Fatemeh Rajati[206], Muhammad Ahmed Rameto[33,512], Kiana Ramezanzadeh[513], Chhabi Lal Ranabhat[514,515], Sowmya J. Rao[516], Davide Rasella[517], Prateek Rastogi[518], Priya Rathi[519], Salman Rawaf[136,520], David Laith Rawaf[521,522], Lal Rawal[523], Reza Rawassizadeh[524], Ramu Rawat[525], Vishnu Renjith[526], Andre M. N. Renzaho[527,528], Bhageerathy Reshmi[307,529], Melese Abate Reta[38,530], Nima Rezaei[13,531], Mohammad Sadegh Rezai[532], Aziz Rezapour[45], Seyed Mohammad Riahi[533], Ana Isabel Ribeiro[534], Jennifer Rickard[535,536], Maria Rios-Blancas[37], Carlos Miguel Rios-González[537,538], Leonardo Roever[539], Morteza Rostamian[540], Salvatore Rubino[541], Godfrey M. Rwegerera[542], Anas M. Saad[543], Seyedmohammad Saadatagah[544], Siamak Sabour[545], Ehsan Sadeghi[206], Sahar Saeedi Moghaddam[205], Shahram Saeidi[83], Rajesh Sagar[546], Amirhossein Sahebkar[547,548], Mohammad Ali Sahraian[196], S. Mohammad Sajadi[549,550], Mohammad Reza Salahshoor[551], Nasir Salam[552], Hosni Salem[553], Marwa Rashad Salem[554], Joshua A. Salomon[555], Hossein Samadi Kafil[182], Evanson Zondani Sambala[294], Abdallah M. Samy[556], Sivan Yegnanarayana Iyer Saraswathy[557,558], Rodrigo Sarmiento-Suárez[559,560], Satish Saroshe[561], Benn Sartorius[151,562], Arash Sarveazad[563], Brijesh Sathian[564,565], Thirunavukkarasu Sathish[566], Lauren E. Schaeffer[1], David C. Schwebel[567], Subramanian Senthilkumaran[568], Hosein Shabaninejad[569,570], Saeed Shahabi[571], Amira A. Shaheen[572], Masood Ali Shaikh[573], Ali S. Shalash[574], Mehran Shams-Beyranvand[575], MohammadBagher Shamsi[576], Morteza Shamsizadeh[577], Kiomars Sharafi[206], Hamid Sharifi[578], Aziz Sheikh[579,580], Abbas Sheikhtaheri[581], Ranjitha S. Shetty[582], Wondimeneh Shibabaw Shiferaw[89], Mika Shigematsu[583], Jae Il Shin[584], Reza Shirkoohi[585,586], Soraya Siabani[587,588], Tariq Jamal Siddiqi[589], Jonathan I. S. Silverberg[590], Biagio Simonetti[591,592], Jasvinder A. Singh[593,594], Dhirendra Narain Sinha[595,596], Abiy H. Sinke[597], Amin Soheili[598], Anton Sokhan[599], Shahin Soltani[206], Moslem Soofi[83], Muluken Bekele Sorrie[113], Ireneous N. Soyiri[600], Adel Spotin[601], Emma Elizabeth Spurlock[1], Chandrashekhar T. Sreeramareddy[602], Agus Sudaryanto[603], Mu'awiyyah Babale Sufiyan[604], Hafiz Ansar Rasul Suleria[605], Rizwan Suliankatchi Abdulkader[606,607], Amir Taherkhani[608], Leili Tapak[429,609], Nuno Taveira[610,611], Parvaneh Taymoori[438,612], Yonatal Mesfin Tefera[613,614], Arash Tehrani-Banihashemi[81,615], Berhane Fseha Teklehaimanot[616], Gebretsadkan Hintsa Tekulu[617], Berhe Etsay Tesfay[616], Zemenu Tadesse Tessema[104], Belay Tessema[618], Kavumpurathu Raman Thankappan[619], Hamid Reza Tohidinik[275,578], Roman Topor-Madry[620,621], Marcos Roberto Tovani-Palone[622,623], Bach Xuan Tran[624], Riaz Uddin[625,626], Irfan Ullah[627], Chukwuma David Umeokonkwo[628], Bhaskaran Unnikrishnan[629], Era Upadhyay[630], Muhammad Shariq Usman[332], Maryam Vaezi[631,632], Sahel Valadan Tahbaz[633,634], Pascual R. Valdez[635,636], Yasser Vasseghian[277], Yousef Veisani[637], Francesco S. Violante[204,638], Sebastian Vollmer[639], Yasir Waheed[640], Jon Wakefield[641,642], Yafeng Wang[643], Yuan-Pang Wang[139], Girmay Teklay Weldesamuel[224], Andrea Werdecker[644], Ronny Westerman[645], Taweewat Wiangkham[646], Kirsten E. Wiens[1], Charles Shey Wiysonge[294,468], Dawit Zewdu Wondafrash[648,649], Tewodros Eshete Wonde[10], Ai-Min Wu[650], Ali Yadollahpour[651], Seyed Hossein Yahyazadeh Jabbari[633], Tomohide Yamada[652], Sanni Yaya[653,654], Vahid Yazdi-Feyzabadi[655,656], Tomas Y. Yeheyis[657], Yigizie Yeshaw[104], Christopher Sabo Yilgwan[658,659], Paul Yip[660,661], Naohiro Yonemoto[662,663], Mustafa Z. Younis[664,665], Zabihollah Yousefi[666], Mahmoud Yousefifard[491], Taraneh Yousefinezhadi[667], Chuanhua Yu[643], Hasan Yusefzadeh[666], Siddhesh Zadey[668], Telma Zahirian Moghadam[669], Leila Zaki[232], Sojib Bin Zaman[282,670], Mohammad Zamani[671], Maryam Zamanian[51], Hamed Zandian[669,672], Alireza Zangeneh[83],

**Fatemeh Zarei**[673], **Taddese Alemu Zerfu**[674,675], **Yunquan Zhang**[676,677], **Zhi-Jiang Zhang**[678], **Xiu-Ju George Zhao**[679,680], **Maigeng Zhou**[681], **Arash Ziapour**[587], **Simon I. Hay**[1,151,682]✉, **Stephen S. Lim**[1,151,682]✉ & **Jonathan F. Mosser**[1,151,682]✉

[1]Institute for Health Metrics and Evaluation, University of Washington, Seattle, WA, USA. [2]Department of Infectious Disease Epidemiology, London School of Hygiene & Tropical Medicine, London, UK. [3]Social Determinants of Health Research Center, Shahid Beheshti University of Medical Sciences, Tehran, Iran. [4]Advanced Diagnostic and Interventional Radiology Research Center, Tehran University of Medical Sciences, Tehran, Iran. [5]Department of Neurology, Cairo University, Cairo, Egypt. [6]The Institute of Pharmaceutical Sciences (TIPS), Tehran University of Medical Sciences, Tehran, Iran. [7]School of Pharmacy, Tehran University of Medical Sciences, Tehran, Iran. [8]Department of Biostatistics, Near East University, Nicosia, Cyprus. [9]Department of Biostatistics and Health Informatics, Madda Walabu University, Bale Robe, Ethiopia. [10]Department of Public Health, Debre Markos University, Debre Markos, Ethiopia. [11]School of Public Health, Bahir Dar University, Bahir Dar, Ethiopia. [12]Department of Laboratory Medicine, Karolinska University Hospital, Huddinge, Sweden. [13]Research Center for Immunodeficiencies, Tehran University of Medical Sciences, Tehran, Iran. [14]Department of Pediatric Dentistry, Federal University of Minas Gerais, Belo Horizonte, Brazil. [15]Department of Research, Philippine Institute for Development Studies, Quezon City, The Philippines. [16]Department of Cardiovascular Medicine, Cleveland Clinic, Cleveland, OH, USA. [17]Department of Medicine, Ain Shams University, Cairo, Egypt. [18]Department of Disease Control, London School of Hygiene & Tropical Medicine, London, UK. [19]Clinical Research and Operations, Foundation for Scientific Research (FORS), Cotonou, Benin. [20]Hamadan University of Medical Sciences, Hamadan, Iran. [21]College of Medicine, University College Hospital, Ibadan, Ibadan, Nigeria. [22]Population Health Sciences, King's College London, London, UK. [23]Centre of Excellence for Epidemiological Modelling and Analysis, Stellenbosch University, Stellenbosch, South Africa. [24]Department of Global Health, Stellenbosch University, Cape Town, South Africa. [25]School of Health, Ardabil University of Medical Science, Ardabil, Iran. [26]Department of Dermatology, Mayo Clinic, Rochester, MN, USA. [27]Endocrinology and Metabolism Research Center, Tehran University of Medical Sciences, Tehran, Iran. [28]Department of Epidemiology and Biostatistics, Qom University of Medical Sciences, Qom, Iran. [29]Department of Epidemiology and Health Statistics, Southeast University, Nanjing, China. [30]Drug Applied Research Center, Tabriz University of Medical Sciences, Tabriz, Iran. [31]Department of Food Science and Technology, University of Tabriz, Tabriz, Iran. [32]Lincoln Medical School, Universities of Nottingham & Lincoln, Lincoln, UK. [33]Department of Epidemiology, , Jimma University, Jimma, Ethiopia. [34]Australian Center for Precision Health, , University of South Australia, Adelaide, South Australia, Australia. [35]Imam Abdulrahman Bin Faisal University, Dammam, Saudi Arabia. [36]Health Information Management and Technology Department, Imam Abdulrahman Bin Faisal University, Dammam, Saudi Arabia. [37]Center for Health System Research, National Institute of Public Health, Cuernavaca, Mexico. [38]Department of Medical Laboratory Science, Woldia University, Woldia, Ethiopia. [39]Erbil Technical Health College, Erbil Polytechnic University, Erbil, Iraq. [40]School of Pharmacy, Tishk International University, Erbil, Iraq. [41]Department of Biotechnology, Quaid-i-Azam University, Islamabad, Pakistan. [42]Social Determinants of Health Research Center, Qazvin University of Medical Sciences, Qazvin, Iran. [43]Department of Health Care Management and Economics, Urmia University of Medical Sciences, Urmia, Iran. [44]Student Research Committee, Hormozgan University of Medical Sciences, Bandar Abbas, Iran. [45]Health Management and Economics Research Center, Iran University of Medical Sciences, Tehran, Iran. [46]Health Economics Department, Iran University of Medical Sciences, Tehran, Iran. [47]Infectious and Tropical Disease Research Center, Hormozgan University of Medical Sciences, Bandar Abbas, Iran. [48]Department of Health Policy and Management, Kuwait University, Safat, Kuwait. [49]International Centre for Casemix and Clinical Coding, National University of Malaysia, Bandar Tun, Razak, Malaysia. [50]Department of Environmental Health Engineering, Kermanshah University of Medical Sciences, Kermanshah, Iran. [51]Department of Epidemiology, Arak University of Medical Sciences, Arak, Iran. [52]Medical Research Center, Jazan University, Jazan, Saudi Arabia. [53]Department of Parasitology, Sana'a University, Sana'a, Yemen. [54]Pediatric Intensive Care Unit, King Saud University, Riyadh, Saudi Arabia. [55]Department of Pharmacology, Mekelle University, Mekelle, Ethiopia. [56]Health Services Management Department, Arak University of Medical Sciences, Arak, Iran. [57]Health Management and Economics Research Center, Isfahan University of Medical Sciences, Isfahan, Iran. [58]Department of Radiology and Nuclear Medicine, Kermanshah University of Medical Sciences, Kermanshah, Iran. [59]Department of Epidemiology and Biostatistics, University of the Philippines Manila, Manila, The Philippines. [60]School of Public Health, Johns Hopkins University, Baltimore, MD, USA. [61]Maternal and Child Wellbeing, African Population and Health Research Center, Nairobi, Kenya. [62]Pharmacy Department, Carol Davila University of Medicine and Pharmacy, Bucharest, Romania. [63]Cardiology Department, Carol Davila University of Medicine and Pharmacy, Bucharest, Romania. [64]Social Determinants of Health Research Center, Rafsanjan University of Medical Sciences, Rafsanjan, Iran. [65]Research Center for Evidence Based Medicine, Tabriz University of Medical Sciences, Tabriz, Iran. [66]Razi Vaccine and Serum Research Institute, Agricultural Research, Education, and Extension Organization (AREEO), Tehran, Iran. [67]Department of Epidemiology and Biostatistics, Zahedan University of Medical Sciences, Zahedan, Iran. [68]Department of Public Health, Debre Berhan University, Debre Berhan, Ethiopia. [69]Department of Health Policy and Administration, University of the Philippines Manila, Manila, The Philippines. [70]Department of Applied Social Sciences, Hong Kong Polytechnic University, Hong Kong, China. [71]Agribusiness Study Program, Sebelas Maret University, Surakarta, Indonesia. [72]Department of Parasitology, Mazandaran University of Medical Sciences, Sari, Iran. [73]Department of Parasitology, Iranshahr University of Medical Sciences, Iranshahr, Iran. [74]Social Determinants of Health Research Center, Birjand University of Medical Sciences, Birjand, Iran. [75]Department of Public Health, Birmingham City University, Birmingham, UK. [76]Social Determinants of Health Research Center, Saveh University of Medical Sciences, Saveh, Iran. [77]Social Determinants of Health Research Center, Yasuj University of Medical Sciences, Yasuj, Iran. [78]Monitoring Evaluation and Operational Research Project, Abt Associates Nepal, Lalitpur, Nepal. [79]School of Nursing and Midwifery, Babol University of Medical Sciences, Babol, Iran. [80]Babol University of Medical Sciences, Babol, Iran. [81]Preventive Medicine and Public Health Research Center, Iran University of Medical Sciences, Tehran, Iran. [82]Prevention of Metabolic Disorders Research Center, Shahid Beheshti University of Medical Sciences, Tehran, Iran. [83]Social Development and Health Promotion Research Center, Kermanshah University of Medical Sciences, Kermanshah, Iran. [84]School of Business, University of Leicester, Leicester, UK. [85]Department of Statistics and Econometrics, Bucharest University of Economic Studies, Bucharest, Romania. [86]Department of Nursing, Wolaita Sodo University, Wolaita Sodo, Ethiopia. [87]The Judith Lumley Centre, La Trobe University, Melbourne, Victoria, Australia. [88]Department of Health Policy Planning and Management, University of Health and Allied Sciences, Ho, Ghana. [89]Department of Nursing, Debre Berhan University, Debre Berhan, Ethiopia. [90]Cellular and Molecular Biology Research Center, Babol University of Medical Sciences, Babol, Iran. [91]Public Health Risk Sciences Division, Public Health Agency of Canada, Toronto, Ontario, Canada. [92]Department of Nutritional Sciences, University of Toronto, Toronto, Ontario, Canada. [93]Department of Forensic Science, Government Institute of Forensic Science, Nagpur, India. [94]Department of Healthcare Management and Education, Shiraz University of Medical Sciences, Shiraz, Iran. [95]Unit of Biochemistry, Sultan Zainal Abidin University (Universiti Sultan Zainal Abidin), Kuala Terengganu, Malaysia. [96]Department of Health Policy, Management, and Economics, Tehran University of Medical Sciences, Tehran, Iran. [97]Department of Medical Microbiology, Haramaya University, Harar, Ethiopia. [98]Department of Hypertension, Medical University of Lodz, Lodz, Poland. [99]Polish Mothers' Memorial Hospital Research Institute, Lodz, Poland. [100]Department of Non-communicable Diseases, Bangladesh University of Health Sciences, Dhaka, Bangladesh. [101]Clinic for Infectious and Tropical Diseases, Clinical Center of Serbia, Belgrade, Serbia. [102]Faculty of Medicine, University of Belgrade, Belgrade, Serbia. [103]Razi Vaccine and Serum Research Institute, Agricultural Research, Education and Extension Organization (AREEO), Karaj, Iran. [104]Department of Epidemiology and Biostatistics, University of Gondar, Gondar, Ethiopia. [105]Center for Primary Care, Harvard University, Boston, MA, USA. [106]School of Public Health, Imperial College London, London, UK. [107]Health Human Resources Research Center, Shiraz University of Medical Sciences, Shiraz, Iran. [108]Monitoring, Evaluation and Research Department, JSI Research & Training Institute, Addis Ababa, Ethiopia. [109]Department of Community Medicine, Gandhi Medical College Bhopal, Bhopal, India. [110]Jazan University, Jazan, Saudi Arabia. [111]Social Determinants of Health Research Center, Lorestan University of Medical Sciences, Khorramabad, Iran. [112]School of the Environment, Yale University, New Haven, CT, USA. [113]Department of Public Health, Arba Minch University, Arba Minch, Ethiopia. [114]Department of Nutrition and Dietetics, Mekelle University, Mekelle, Ethiopia. [115]Department of Global Health, Global Institute for Interdisciplinary Studies, Kathmandu, Nepal. [116]Centre for Global Child Health, University of Toronto, Toronto, Ontario, Canada. [117]Centre of Excellence in Women & Child Health, Aga Khan University, Karachi, Pakistan. [118]Social Determinants of Health Research Center, Babol University of Medical Sciences, Babol, Iran. [119]Department of Pediatrics and Child Health Nursing, Bahir Dar University, Bahir Dar, Ethiopia. [120]Global Health Division, Research Triangle Institute International, Research Triangle Park, NC, USA. [121]School of Medicine, University of Nottingham, Nottingham, UK. [122]Department of Neurology, Institute of Post-Graduate Medical Education and Research and Seth Sukhlal Karnani Memorial Hospital, Kolkata, India. [123]Department of Veterinary Medicine, Islamic Azad University, Kermanshah, Iran. [124]Department of Biomedical Sciences, Nazarbayev University, Nur-Sultan City, Kazakhstan. [125]Department of Endocrinology, Hamadan University of Medical Sciences, Hamadan, Iran. [126]University of Genoa, Genoa, Italy. [127]Department of Biomedical Technologies, Bauman Moscow State Technical University, Moscow, Russia. [128]Department of Epidemiology and Evidence Based Medicine, I. M. Sechenov First Moscow State Medical University, Moscow, Russia. [129]Department of Community Medicine, Employee State Insurance Post Graduate Institute of Medical Sciences and Research, Bangalore, India. [130]School of Public Health and Health Systems, University of Waterloo, Waterloo, Ontario, Canada. [131]Al Shifa School of Public Health, Al Shifa Trust Eye Hospital, Rawalpindi, Pakistan. [132]Internal Medicine Department, Hospital Italiano de Buenos Aires, Buenos Aires, Argentina. [133]Board of Directors, Argentine Society of Medicine, Buenos Aires, Argentina. [134]Health and Nutrition Research Center, National Institute of Public Health, Cuernavaca, Mexico. [135]Centre for Population Health Sciences, Nanyang Technological University, Singapore, Singapore. [136]Department of Primary Care and Public Health, Imperial College London, London, UK. [137]Department of Health Care, Metropolitan Autonomous University, Mexico City, Mexico. [138]Research Unit on Applied Molecular Biosciences (UCIBIO), University of Porto, Porto, Portugal. [139]Department of Psychiatry, University of São Paulo, São Paulo, Brazil. [140]Gorgas Memorial Institute for Health Studies, Panama City, Panama. [141]Department of Medicine, University of Toronto, Toronto, Ontario, Canada. [142]Department of Biostatistics and Epidemiology, Babol University of Medical Sciences, Babol, Iran. [143]Epidemiology Research Center, Royan Institute, Tehran, Iran. [144]Department of Epidemiology and Preventive Medicine, Monash University, Melbourne, Victoria, Australia. [145]Melbourne Medical School, University of Melbourne, Parkville, Victoria, Australia. [146]Faculty of Biology, Hanoi National University of Education, Hanoi, Vietnam. [147]Environmental Health Department, Imam Abdulrahman Bin Faisal University, Dammam, Saudi Arabia. [148]Clinical Dermatology, IRCCS Istituto Ortopedico Galeazzi, University of Milan, Milan, Italy. [149]Department of Dermatology, Case Western Reserve University, Cleveland, OH, USA. [150]Public Health Foundation of India, Gurugram, India. [151]Department of Health Metrics Sciences, School of Medicine, University of Washington, Seattle, WA, USA. [152]Indian Council of Medical Research, New Delhi, India. [153]Immunization, Vaccines and Biologicals (IVB), World Health Organization (WHO), Geneva, Switzerland. [154]Global Delivery Programs, Bill & Melinda Gates Foundation, Seattle, WA, USA. [155]Department of Information Technology, University of Human Development, Sulaymaniyah, Iraq. [156]Department of Pediatrics, Tanta University, Tanta, Egypt. [157]Division of Women and Child Health, Aga Khan University, Karachi, Pakistan. [158]Heidelberg Institute of Global Health (HIGH), Heidelberg University, Heidelberg, Germany. [159]Department of Medical Laboratory Sciences, Bahir Dar University, Bahir Dar, Ethiopia. [160]Department of Nursing, Woldia University, Woldia, Ethiopia. [161]School of Nursing, Jimma University, Jimma, Ethiopia. [162]Center for Nutrition and Health Research, National Institute of Public Health, Cuernavaca, Mexico. [163]School of Nursing and Midwifery, Haramaya University, Harar, Ethiopia. [164]School of Pharmacy, Mekelle University, Mekelle, Ethiopia. [165]Department of Community Medicine, University of Peradeniya, Peradeniya, Sri Lanka. [166]Department of Microbiology, Far Western University, Mahendranagar, Nepal. [167]Department of Epidemiology and Biostatistics, Shahroud University of Medical Sciences, Shahroud, Iran. [168]Department of Epidemiology, Shiraz University of Medical Sciences, Shiraz, Iran. [169]Center of Complexity Sciences, National Autonomous University of Mexico, Mexico City, Mexico. [170]Faculty of Veterinary Medicine and

Zootechnics, Autonomous University of Sinaloa, Culiacán Rosales, Mexico. [171]Department of Health Promotion and Education, University of Ibadan, Ibadan, Nigeria. [172]Institute of Health Economics and Technology, Hanoi, Vietnam. [173]Department of Medical Laboratory Sciences, Iran University of Medical Sciences, Tehran, Iran. [174]Department of Health Policy and Management, Tabriz University of Medical Sciences, Tabriz, Iran. [175]School of Medicine, Western Sydney University, Sydney, New South Wales, Australia. [176]Health Sciences, Royal Melbourne Institute of Technology University, Melbourne, Victoria, Australia. [177]Centre for Tropical Medicine and Global Health, University of Oxford, Oxford, UK. [178]Mahidol-Oxford Tropical Medicine Research Unit, Bangkok, Thailand. [179]School of Medicine, Federal University of Bahia, Salvador, Brazil. [180]Department of Internal Medicine, Escola Bahiana de Medicina e Saúde Pública, Salvador, Brazil. [181]Department of Epidemiology, Florida International University, Miami, FL, USA. [182]Department of Bacteriology and Virology, Tabriz University of Medical Sciences, Tabriz, Iran. [183]School of Health Sciences, Universiti Sains Malaysia, Kubang Kerian, Malaysia. [184]Centre Clinical Epidemiology and Biostatistics, University of Newcastle, Newcastle, New South Wales, Australia. [185]Department of Pharmacology and Toxicology, Maragheh University of Medical Sciences, Maragheh, Iran. [186]Department of Pharmacology and Toxicology, The John Paul II Catholic University of Lublin, Lublin, Poland. [187]Biomedical Informatics and Medical Statistics Department, Alexandria University, Alexandria, Egypt. [188]Reference Laboratory of Egyptian Universities Hospitals, Ministry of Higher Education and Research, Cairo, Egypt. [189]Department of Food Science and Nutrition, Arsi University, Asella, Ethiopia. [190]Neurophysiology Department, Cairo University, Cairo, Egypt. [191]Endemic Medicine and Hepatogastroentrology Department, Cairo University, Cairo, Egypt. [192]Ophthalmic Epidemiology Research Center, Shahroud University of Medical Sciences, Shahroud, Iran. [193]Department of Microbiology and Immunology, Suez Canal University, Ismailia, Egypt. [194]Department of Medicinal Chemistry, Kerman University of Medical Sciences, Kerman, Iran. [195]Pharmaceutics Research Center, Kerman University of Medical Sciences, Kerman, Iran. [196]Multiple Sclerosis Research Center, Tehran University of Medical Sciences, Tehran, Iran. [197]Department of Physiology, Tarbiat Modares University, Tehran, Iran. [198]Tehran Medical Sciences Branch, Islamic Azad University, Tehran, Iran. [199]Department of Public Health, Maragheh University of Medical Sciences, Maragheh, Iran. [200]Unit of Medical Physiology, Hawassa University, Hawassa, Ethiopia. [201]School of Public Health, Tehran University of Medical Sciences, Tehran, Iran. [202]College of Medicine, Imam Mohammad Ibn Saud Islamic University, Riyadh, Saudi Arabia. [203]Department of Medical Parasitology, Mazandaran University of Medical Sciences, Sari, Iran. [204]Department of Medical and Surgical Sciences, University of Bologna, Bologna, Italy. [205]Non-communicable Diseases Research Center, Tehran University of Medical Sciences, Tehran, Iran. [206]Research Center for Environmental Determinants of Health, Kermanshah University of Medical Sciences, Kermanshah, Iran. [207]Department of Environmental Health Engineering, Ardabil University of Medical Science, Ardabil, Iran. [208]Department of Environmental Health Engineering, Tehran University of Medical Sciences, Tehran, Iran. [209]Department of Neurobiology, Karolinska Institute, Stockholm, Sweden. [210]Division of Neurology, University of Ottawa, Ottawa, Ontario, Canada. [211]Associated Laboratory for Green Chemistry (LAQV), University of Porto, Porto, Portugal. [212]Psychiatry Department, Kaiser Permanente, Fontana, CA, USA. [213]School of Health Sciences, A. T. Still University, Mesa, AZ, USA. [214]Institute of Gerontological Health Services and Nursing Research, Ravensburg-Weingarten University of Applied Sciences, Weingarten, Germany. [215]Institute of Gerontology, National Academy of Medical Sciences of Ukraine, Kyiv, Ukraine. [216]Department of Child Dental Health, Obafemi Awolowo University, Ile-Ife, Nigeria. [217]Department of Medical Parasitology, Abadan Faculty of Medical Sciences, Abadan, Iran. [218]Department of Dermatology, Kobe University, Kobe, Japan. [219]Gillings School of Global Public Health, University of North Carolina Chapel Hill, Chapel Hill, NC, USA. [220]Department of Public Health, Madda Walabu University, Bale Robe, Ethiopia. [221]Menelik-II College of Medical and Health Sciences, Kotebe Metropolitan University, Addis Ababa, Ethiopia. [222]School of Public Health, Mekelle University, Mekelle, Ethiopia. [223]Department of Nursing and Midwifery, Addis Ababa University, Addis Ababa, Ethiopia. [224]Department of Nursing, Aksum University, Aksum, Ethiopia. [225]Department of Nursing, Mekelle University, Mekelle, Ethiopia. [226]Department of Reproductive Health, Mekelle University, Mekelle, Ethiopia. [227]Department of Midwifery, Woldia University, Woldia, Ethiopia. [228]Department of Biostatistics, Mekelle University, Mekelle, Ethiopia. [229]Infectious Disease Research Center, Kermanshah University of Medical Sciences, Kermanshah, Iran. [230]Pediatric Department, Kermanshah University of Medical Sciences, Kermanshah, Iran. [231]Medical Education Research Center, Tabriz University of Medical Sciences, Tabiz, Iran. [232]Department of Parasitology and Entomology, Tarbiat Modares University, Tehran, Iran. [233]Department of Neurology, Tehran University of Medical Sciences, Tehran, Iran. [234]Faculty of Nursing and Midwifery, Kurdistan University of Medical Sciences, Sanandaj, Iran. [235]Student Research Committee, Iran University of Medical Sciences, Tehran, Iran. [236]Institute of Health Research, University of Exeter, Exeter, UK. [237]Young Researchers and Elite Club, Islamic Azad University, Rasht, Iran. [238]Department of Biology, Islamic Azad University, Tehran, Iran. [239]Faculty of Allied Health Sciences, The University of Lahore, Lahore, Pakistan. [240]Afro-Asian Institute, Lahore, Pakistan. [241]Medical School, University of Warwick, Coventry, UK. [242]Department of Chemistry, University of Porto, Porto, Portugal. [243]Hudson College of Public Health, University of Oklahoma Health Sciences Center, Oklahoma City, OK, USA. [244]Department of Health and Social Affairs, Government of the Federated States of Micronesia, Palikir, Federated States of Micronesia. [245]Postgraduate Program in Epidemiology, Federal University of Rio Grande do Sul, Porto Alegre, Brazil. [246]Department of Dermatology, Boston University, Boston, MA, USA. [247]Institute of Tropical Pathology and Public Health (IPTSP), Federal University of Goias, Goiânia, Brazil. [248]Department of Epidemiology, Binzhou Medical University, Yantai City, China. [249]Medical Resources, March of Dimes, Arlington, VA, USA. [250]Health Policy, Management and Leadership, West Virginia University School of Public Health, Morgantown, WV, USA. [251]Department of Radiology and Radiological Sciences, Johns Hopkins University, Baltimore, MD, USA. [252]School of Medicine, Tehran University of Medical Sciences, Tehran, Iran. [253]Department of Pharmacology, Tehran University of Medical Sciences, Tehran, Iran. [254]Obesity Research Center, Shahid Beheshti University of Medical Sciences, Tehran, Iran. [255]Department of Public Health, Wachemo University, Hossana, Ethiopia. [256]University Institute of Public Health, The University of Lahore, Lahore, Pakistan. [257]Tabriz University of Medical Sciences, Tabriz, Iran. [258]Department of Zoology and Entomology, Al Azhar University, Cairo, Egypt. [259]Institute for Social Science Research, The University of Queensland, Indooroopilly, Queensland, Australia. [260]ARC Centre of Excellence for Children and Families over the Life Course, The University of Queensland, Indooroopilly, Queensland, Australia. [261]Department of Healthcare Management, Maragheh University of Medical Sciences, Maragheh, Iran. [262]Department of Microbiology, Maragheh University of Medical Sciences, Maragheh, Iran. [263]Department of Microbiology, Tehran University of Medical Sciences, Tehran, Iran. [264]Gastrointestinal and Liver Diseases Research Center, Guilan University of Medical Sciences, Rasht, Iran. [265]Caspian Digestive Disease Research Center, Guilan University of Medical Sciences, Rasht, Iran. [266]School of Nursing and Midwifery, Tabriz University of Medical Sciences, Tabriz, Iran. [267]Independent Consultant, Tabriz, Iran. [268]School of Nursing and Midwifery, Tehran University of Medical Sciences, Tehran, Iran. [269]Big Data Institute, University of Oxford, Oxford, UK. [270]School of Business, London South Bank University, London, UK. [271]Medical Biology Research Center, Kermanshah University of Medical Sciences, Kermanshah, Iran. [272]Guilan Road Trauma Research Center, Guilan University of Medical Sciences, Rasht, Iran. [273]Centre for Bio Cultural Studies (CBiCS), Manipal Academy of Higher Education, Manipal, India. [274]Department of Pharmacology, Bangladesh Industrial Gases Limited, Tangail, Bangladesh. [275]Department of Epidemiology and Biostatistics, Tehran University of Medical Sciences, Tehran, Iran. [276]Pediatric Chronic Kidney Disease Research Center, Tehran University of Medical Sciences, Tehran, Iran. [277]Institute of Research and Development, Duy Tan University, Da Nang, Vietnam. [278]Department of Computer Science, University of Human Development, Sulaymaniyah, Iraq. [279]College of Science and Engineering, Hamad Bin Khalifa University, Doha, Qatar. [280]Department of Epidemiology and Health Statistics, Central South University, Changsha, China. [281]School of Public Health, University of Sydney, Sydney, New South Wales, Australia. [282]Maternal and Child Health Division, International Centre for Diarrhoeal Disease Research, Bangladesh, Dhaka, Bangladesh. [283]Department of Public Health and Community Medicine, Shaikh Khalifa Bin Zayed Al-Nahyan Medical College, Lahore, Pakistan. [284]Department of Community Medicine, University of Ibadan, Ibadan, Nigeria. [285]Department of Community Medicine, University College Hospital, Ibadan, Ibadan, Nigeria. [286]Department of Epidemiology, University of Kragujevac, Kragujevac, Serbia. [287]Department of Public Health, Lorestan University of Medical Sciences, Khorramabad, Iran. [288]Division of Community Health and Family Medicine, Bangalore Baptist Hospital, Bangalore, India. [289]College of Public Health, Taipei Medical University, Taipei, Taiwan. [290]Research Institute for Endocrine Sciences, Shahid Beheshti University of Medical Sciences, Tehran, Iran. [291]Institute for Physical Activity and Nutrition, Deakin University, Burwood, Victoria, Australia. [292]Sydney Medical School, University of Sydney, Sydney, New South Wales, Australia. [293]School of Psychology and Public Health, La Trobe University, Melbourne, Victoria, Australia. [294]South African Medical Research Council, Cape Town, South Africa. [295]School of Health Systems and Public Health, University of Pretoria, Pretoria, South Africa. [296]Department of Immunology, Tabriz University of Medical Sciences, Tabriz, Iran. [297]Department of Immunology, Isfahan University of Medical Sciences, Isfahan, Iran. [298]School of Management and Medical Education, Shahid Beheshti University of Medical Sciences, Tehran, Iran. [299]Safety Promotion and Injury Prevention Research Center, Shahid Beheshti University of Medical Sciences, Tehran, Iran. [300]N. A. Semashko Department of Public Health and Healthcare, I. M. Sechenov First Moscow State Medical University, Moscow, Russia. [301]Department of Global Health, Economics and Policy, University of Kragujevac, Kragujevac, Serbia. [302]Health Institute, Kermanshah University of Medical Sciences, Kermanshah, Iran. [303]Substance Abuse Prevention Research Center, Kermanshah University of Medical Sciences, Kermanshah, Iran. [304]Department of Medical Mycology, Mazandaran University of Medical Sciences, Sari, Iran. [305]Autism Spectrum Disorders Research Center, Hamadan University of Medical Sciences, Hamadan, Iran. [306]The George Institute for Global Health, New Delhi, India. [307]Manipal Academy of Higher Education, Manipal, India. [308]Environmental Research Center, Duke Kunshan University, Kunshan, China. [309]Nicholas School of the Environment, Duke University, Durham, NC, USA. [310]Renal and Cardiovascular Division, The George Institute for Global Health, New Delhi, India. [311]Department of Medicine, University of New South Wales, Sydney, New South Wales, Australia. [312]Department of Family Medicine and Public Health, University of Opole, Opole, Poland. [313]School of Public Health, University College Cork, Cork, Ireland. [314]Minimally Invasive Surgery Research Center, Iran University of Medical Sciences, Tehran, Iran. [315]Infectious Diseases Research Center, Golestan University of Medical Sciences, Gorgan, Iran. [316]School of Management and Medical Informatics, Tabriz University of Medical Sciences, Tabriz, Iran. [317]Institute for Prevention of Non-communicable Diseases, Qazvin University of Medical Sciences, Qazvin, Iran. [318]Health Services Management Department, Qazvin University of Medical Sciences, Qazvin, Iran. [319]Department of Pharmacy, Shaheed Benazir Bhutto University, Upper Dir, Pakistan. [320]School of Pharmacy, Shanghai Jiao Tong University, Shanghai, China. [321]Department of Forensic Medicine and Toxicology, All India Institute of Medical Sciences, Jodhpur, India. [322]Department of Epidemiology, Hamadan University of Medical Sciences, Hamadan, Iran. [323]Institute for Epidemiology and Social Medicine, University of Münster, Münster, Germany. [324]Social Determinants of Health Research Center, Tabriz University of Medical Sciences, Tabriz, Iran. [325]International Research Center of Excellence, Institute of Human Virology Nigeria, Abuja, Nigeria. [326]Julius Centre for Health Sciences and Primary Care, Utrecht University, Utrecht, The Netherlands. [327]Open, Distance and eLearning Campus, University of Nairobi, Nairobi, Kenya. [328]Department of Public Health, Jordan University of Science and Technology, Irbid, Jordan. [329]Social Determinants of Health Research Center, Ahvaz Jundishapur University of Medical Sciences, Ahvaz, Iran. [330]Health Promotion Research Center, Zahedan University of Medical Sciences, Zahedan, Iran. [331]Department of Internal Medicine, John H. Stroger, Jr Hospital of Cook County, Chicago, IL, USA. [332]Department of Internal Medicine, Dow University of Health Sciences, Karachi, Pakistan. [333]Department of Epidemiology and Biostatistics, Health Services Academy, Islamabad, Pakistan. [334]Department of Population Studies, International Institute for Population Sciences, Mumbai, India. [335]Department of Population Science, Jatiya Kabi Kazi Nazrul Islam University, Mymensingh, Bangladesh. [336]Faculty of Health and Wellbeing, Sheffield Hallam University, Sheffield, UK. [337]College of Arts and Sciences, Ohio University, Zanesville, OH, USA. [338]Department of Medical Parasitology, Cairo University, Cairo, Egypt. [339]Global Evidence Synthesis Initiative, Datta Meghe Institute of Medical Sciences, Wardha, India. [340]Shahid Beheshti University of Medical Sciences, Tehran, Iran. [341]The Iranian Academy of Medical Sciences, Tehran, Iran. [342]Department of Neurology, Hamadan University of Medical Sciences, Hamadan, Iran. [343]Deputy for Public Health, Ministry of Health and Medical Education, Tehran, Iran. [344]Health Equity Research Center, Tehran University of Medical Sciences, Tehran, Iran. [345]Department of Public Health, New Mexico State University, Las

Cruces, NM, USA. [346]Department of Public Health, Kermanshah University of Medical Sciences, Kermanshah, Iran. [347]School of Traditional Chinese Medicine, Xiamen University Malaysia, Sepang, Malaysia. [348]Department of Nutrition, Simmons University, Boston, MA, USA. [349]School of Health Sciences, Kristiania University College, Oslo, Norway. [350]Global Community Health and Behavioral Sciences, Tulane University, New Orleans, LA, USA. [351]Department of Nursing and Health Promotion, Oslo Metropolitan University, Oslo, Norway. [352]Department of Public Health, Ambo University, Ambo, Ethiopia. [353]Neurophysiology Research Center, Hamadan University of Medical Sciences, Hamadan, Iran. [354]Brain Engineering Research Center, Institute for Research in Fundamental Sciences, Tehran, Iran. [355]Independent Consultant, Jakarta, Indonesia. [356]Department of Internal and Pulmonary Medicine, Sheri Kashmir Institute of Medical Sciences, Srinagar, India. [357]CIBERSAM, San Juan de Dios Sanitary Park, Sant Boi de Llobregat, Spain. [358]Catalan Institution for Research and Advanced Studies (ICREA), Barcelona, Spain. [359]Department of Zoology, University of Oxford, Oxford, UK. [360]Harvard Medical School, Harvard University, Boston, MA, USA. [361]Department of Anthropology, Panjab University, Chandigarh, India. [362]Department of Demography, University of Montreal, Quebec, Canada. [363]Department of Social and Preventive Medicine, University of Montreal, Montreal, Quebec, Canada. [364]Department of Psychiatry, University of Nairobi, Nairobi, Kenya. [365]Division of Psychology and Language Sciences, University College London, London, UK. [366]International Institute for Population Sciences, Mumbai, India. [367]Imperial College Business School, Imperial College London, London, UK. [368]Faculty of Public Health, University of Indonesia, Depok, Indonesia. [369]Department of Clinical Sciences and Community Health, University of Milan, Milan, Italy. [370]Nuffield Department of Population Health, University of Oxford, Oxford, UK. [371]National Institute of Health Research (NIHR), Tehran University of Medical Sciences, Tehran, Iran. [372]Department of Pediatrics, Post Graduate Institute of Medical Education and Research, Chandigarh, India. [373]Department of Essential Medicines and Health Products, Clinton Health Access Initiative, Boston, MA, USA. [374]Department of Community and Family Medicine, University of Baghdad, Baghdad, Iraq. [375]HelpMeSee, New York, NY, USA. [376]Mexican Institute of Ophthalmology, Queretaro, Mexico. [377]Department of Otorhinolaryngology, Father Muller Medical College, Mangalore, India. [378]School of Pharmacy, Monash University, Bandar Sunway, Malaysia. [379]School of Pharmacy, Taylor's University Lakeside Campus, Subang Jaya, Malaysia. [380]School of Nursing, Hong Kong Polytechnic University, Hong Kong, China. [381]School of Public Health, Wolaita Sodo University, Wolaita Sodo, Ethiopia. [382]School of Public Health and Preventive Medicine, Monash University, Melbourne, Victoria, Australia. [383]Chinese Center for Disease Control and Prevention, Beijing, China. [384]David Geffen School of Medicine, University of California Los Angeles, Los Angeles, CA, USA. [385]Center for Integration of Data and Health Knowledge, Oswald Cruz Foundation (FIOCRUZ), Salvador, Brazil. [386]Centre for Global Mental Health (CGMH), London School of Hygiene & Tropical Medicine, London, UK. [387]Grants, Innovation and Product Development Unit, South African Medical Research Council, Cape Town, South Africa. [388]Department of Public Health, Urmia University of Medical Science, Urmia, Iran. [389]Department of Clinical Physiology, Tribhuvan University, Kathmandu, Nepal. [390]Department of Forensic Medicine, Rajiv Gandhi University of Health Sciences, Dharwad, India. [391]Department of Forensic Medicine, Shri Dharmasthala Manjunatheshwara University, Dharwad, India. [392]Clinical Research Development Center, Kermanshah University of Medical Sciences, Kermanshah, Iran. [393]Department of Humanities and Social Sciences, Indian Institute of Technology, Roorkee, Roorkee, India. [394]Department of Development Studies, International Institute for Population Sciences, Mumbai, India. [395]Department of Maternal and Child Nursing and Public Health, Federal University of Minas Gerais, Belo Horizonte, Brazil. [396]Department of Health Education and Promotion, Iran University of Medical Sciences, Tehran, Iran. [397]Campus Caucaia, Federal Institute of Education, Science and Technology of Ceará, Caucaia, Brazil. [398]Faculty of Health and Education, Botho University-Botswana, Gaborone, Botswana. [399]ICF International, DHS Program, Rockville, MD, USA. [400]Neurology Department, Janakpuri Super Specialty Hospital Society, New Delhi, India. [401]Department of Neurology, Govind Ballabh Institute of Medical Education and Research, New Delhi, India. [402]Nutrition Health Research Center, Iran University of Medical Sciences, Hamadan, Iran. [403]Department of Epidemiology and Biostatistics, University of California San Francisco, San Francisco, CA, USA. [404]Institute of Human Virology, University of Maryland, Baltimore, MD, USA. [405]Peru Country Office, United Nations Population Fund (UNFPA), Lima, Peru. [406]Forensic Medicine Division, Imam Abdulrahman Bin Faisal University, Dammam, Saudi Arabia. [407]Department of Midwifery, Adigrat University, Adigrat, Ethiopia. [408]Department of Reproductive Health and Population Studies, Bahir Dar University, Bahir Dar, Ethiopia. [409]Clinical Microbiology and Parasitology Unit, Dr Zora Profozic Polyclinic, Zagreb, Croatia. [410]University Centre Varazdin, University North, Varazdin, Croatia. [411]Department of Epidemiology and Biostatistics, Bahir Dar University, Bahir Dar, Ethiopia. [412]Department of Health Research Methods, Evidence and Impact, McMaster University, Hamilton, Ontario, Canada. [413]Department of Computer Science and Software Engineering, University of Western Australia, Perth, Western Australia, Australia. [414]Fatemeh Zahra Infertility and Reproductive Health Center, Babol University of Medical Sciences, Babol, Iran. [415]Internal Medicine Programme, Kyrgyz State Medical Academy, Bishkek, Kyrgyzstan. [416]Department of Atherosclerosis and Coronary Heart Disease, National Center of Cardiology and Internal Disease, Bishkek, Kyrgyzstan. [417]Comprehensive Research Laboratory, Iran University of Medical Sciences, Tehran, Iran. [418]Water Quality Research Center, Tehran University of Medical Sciences, Tehran, Iran. [419]Department of Rehabilitation and Sports Medicine, Kermanshah University of Medical Sciences, Kermanshah, Iran. [420]Department of Medical Immunology, Tehran University of Medical Sciences, Tehran, Iran. [421]Research Center for Biochemistry and Nutrition in Metabolic Diseases, Kashan University of Medical Sciences, Kashan, Iran. [422]Institute of Addiction Research (ISFF), Frankfurt University of Applied Sciences, Frankfurt, Germany. [423]Biotechnology Research Center, Tabriz University of Medical Sciences, Tabriz, Iran. [424]Molecular Medicine Research Center, Tabriz University of Medical Sciences, Tabriz, Iran. [425]Department of Forestry, Salahaddin University-Erbil, Erbil, Iraq. [426]Department of Medicine-Huddinge, Karolinska Institute, Stockholm, Sweden. [427]Internal Medicine Department, King Saud University, Riyadh, Saudi Arabia. [428]Department of Biology, Salahaddin University-Erbil, Erbil, Iraq. [429]Department of Biostatistics, Hamadan University of Medical Sciences, Hamadan, Iran. [430]Department of Epidemiology and Biostatistics, Shahrekord University of Medical Sciences, Shahrekord, Iran. [431]Department of Nursing, Mashhad University of Medical Sciences, Mashhad, Iran. [432]Health Systems and Policy Research Unit, Ahmadu Bello University, Zaria, Nigeria. [433]School of Pharmacy, Haramaya University, Harar,

Ethiopia. [434]Department of Public Health, Dire Dawa University, Dire Dawa, Ethiopia. [435]Clinical Epidemiology and Public Health Research Unit, Burlo Garofolo Institute for Maternal and Child Health, Trieste, Italy. [436]Department of Molecular Medicine, National Institute of Genetic Engineering and Biotechnology, Tehran, Iran. [437]Health Sciences Research Center, Mazandaran University of Medical Sciences, Sari, Iran. [438]Social Determinants of Health Research Center, Kurdistan University of Medical Sciences, Sanandaj, Iran. [439]Department of Epidemiology and Biostatistics, Kurdistan University of Medical Sciences, Sanandaj, Iran. [440]National Center for Health Insurance Research, Iran Health Insurance Organization, Tehran, Iran. [441]Computer, Electrical, and Mathematical Sciences and Engineering Division, King Abdullah University of Science and Technology, Thuwal, Saudi Arabia. [442]Department of Clinical Biochemistry, Babol University of Medical Sciences, Babol, Iran. [443]Department of Clinical Biochemistry, Tarbiat Modares University, Tehran, Iran. [444]Department of Food Science, University of Campinas (Unicamp), Campinas, Brazil. [445]Federal Institute for Population Research, Wiesbaden, Germany. [446]Center for Population and Health, Wiesbaden, Germany. [447]Indian Institute of Public Health, Public Health Foundation of India, Hyderabad, India. [448]Department of Microbiology and Immunology, Mekelle University, Mekelle, Ethiopia. [449]Research and Analytics Department, Initiative for Financing Health and Human Development, Chennai, India. [450]Department of Research and Analytics, Bioinsilico Technologies, Chennai, India. [451]Suraj Eye Institute, Nagpur, India. [452]Department for the Control of Disease, Epidemics, and Pandemics, Ministry of Public Health, Yaoundé, Cameroon. [453]Department of Public Heath, University of Yaoundé I, Yaoundé, Cameroon. [454]Department of Forensic Medicine and Toxicology, Manipal Academy of Higher Education, Manipal, India. [455]Department of Pediatrics, Arak University of Medical Sciences, Arak, Iran. [456]Cochrane South Africa, South African Medical Research Council, Cape Town, South Africa. [457]Department of General Surgery, Carol Davila University of Medicine and Pharmacy, Bucharest, Romania. [458]Department of General Surgery, Emergency Hospital of Bucharest, Bucharest, Romania. [459]Department of Biological Sciences, University of Embu, Embu, Kenya. [460]Institute for Global Health Innovations, Duy Tan University, Da Nang, Vietnam. [461]Center of Excellence in Behavioral Medicine, Nguyen Tat Thanh University, Ho Chi Minh City, Vietnam. [462]Institute for Mental Health and Policy, Centre for Addiction and Mental Health, Toronto, Ontario, Canada. [463]Department of Clinical Epidemiology, Institute for Clinical Evaluative Sciences, Ottawa, Ontario, Canada. [464]Department of Pharmacoeconomics and Pharmaceutical Administration, Tehran University of Medical Sciences, Tehran, Iran. [465]Hormozgan University of Medical Sciences, Bandar Abbas, Iran. [466]Public Health Department, Universitas Negeri Semarang, Kota Semarang, Indonesia. [467]Graduate Institute of Biomedical Informatics, Taipei Medical University, Taipei, Taiwan. [468]School of Public Health and Family Medicine, University of Cape Town, Cape Town, South Africa. [469]Department of Preventive Medicine, Kyung Hee University, Dongdaemun-gu, South Korea. [470]Gorgan Congenital Malformations Research Center, Golestan University of Medical Sciences, Gorgan, Iran. [471]Department of Psychiatry and Behavioural Neurosciences, McMaster University, Hamilton, Ontario, Canada. [472]Department of Psychiatry, University of Lagos, Lagos, Nigeria. [473]Centre for Healthy Start Initiative, Lagos, Nigeria. [474]Diplomacy and Public Relations Department, University of Human Development, Sulaimaniyah, Iraq. [475]Department of Public Health, Jigjiga University, Jijiga, Ethiopia. [476]Department of Pharmacology and Therapeutics, University of Nigeria Nsukka, Enugu, Nigeria. [477]Department of Medicine, University of Ibadan, Ibadan, Nigeria. [478]Department of Medicine, University College Hospital, Ibadan, Ibadan, Nigeria. [479]Department of Respiratory Medicine, Jagadguru Sri Shivarathreeswara Academy of Health Education and Research, Mysore, India. [480]Department of Forensic Medicine, Manipal Academy of Higher Education, Mangalore, India. [481]Department of Parasitology and Mycology, Shiraz University of Medical Sciences, Shiraz, Iran. [482]Department of Health Metrics, Center for Health Outcomes & Evaluation, Bucharest, Romania. [483]Department of Research, Public Health Foundation of India, Gurugram, India. [484]Infectious Disease Research Center, National Institute of Public Health, Cuernavaca, Mexico. [485]Environmental Health Research Center, Kurdistan University of Medical Sciences, Sanandaj, Iran. [486]Division of General Internal Medicine, University of Pittsburgh Medical Center, Pittsburgh, PA, USA. [487]School of Medicine, University of Sinu, Cartagena, Colombia. [488]Department of Pediatrics, University of Melbourne, Melbourne, Victoria, Australia. [489]Population Health Theme, Murdoch Childrens Research Institute, Melbourne, Victoria, Australia. [490]Department of Physiology, Iran University of Medical Sciences, Tehran, Iran. [491]Physiology Research Center, Iran University of Medical Sciences, Tehran, Iran. [492]Center for Research and Innovation, Ateneo De Manila University, Pasig City, The Philippines. [493]Department of Cardiology, University of Bern, Bern, Switzerland. [494]Institute of Microbiology and Immunology, University of Ljubljana, Ljubljana, Slovenia. [495]University Medical Center Groningen, University of Groningen, Groningen, The Netherlands. [496]School of Economics and Business, University of Groningen, Groningen, The Netherlands. [497]Department of Nutrition and Food Sciences, Maragheh University of Medical Sciences, Maragheh, Iran. [498]Dietary Supplements and Probiotic Research Center, Alborz University of Medical Sciences, Karaj, Iran. [499]School of Population and Public Health, University of British Columbia, Vancouver, British Columbia, Canada. [500]Department of Emergency Medicine, Kermanshah University of Medical Sciences, Kermanshah, Iran. [501]Clinical Research Center, Valle del Lili Foundation (Centro de Investigaciones Clinicas, Fundación Valle del Lili), Cali, Colombia. [502]PROESA, ICESI University (Centro PROESA, Universidad ICESI), Cali, Colombia. [503]Department of Neurology, Smt. B.K.S. Medical Institute and Research Center, Vadodara, India. [504]Department of Community Medicine, Datta Meghe Institute of Medical Sciences, Wardha, India. [505]Department of Chemistry, Sharif University of Technology, Tehran, Iran. [506]Biomedical Engineering Department, Amirkabir University of Technology, Tehran, Iran. [507]College of Medicine, University of Central Florida, Orlando, FL, USA. [508]Department of Immunology, Mazandaran University of Medical Sciences, Sari, Iran. [509]Molecular and Cell Biology Research Center, Mazandaran University of Medical Sciences, Sari, Iran. [510]Thalassemia and Hemoglobinopathy Research Center, Ahvaz Jundishapur University of Medical Sciences, Ahvaz, Iran. [511]Metabolomics and Genomics Research Center, Tehran University of Medical Sciences, Tehran, Iran. [512]College of Medicine & Health Sciences, University of Gondar, Gondar, Ethiopia. [513]Department of Pharmacology, Shahid Beheshti University of Medical Sciences, Tehran, Iran. [514]Research Department, Policy Research Institute, Kathmandu, Nepal. [515]Health and Public Policy Department, Global Center for Research and Development, Kathmandu, Nepal. [516]Department of Oral Pathology, Srinivas Institute of Dental Sciences, Mangalore, India. [517]Institute of Collective Health, Federal

University of Bahia, Salvador, Brazil. [518]Department of Forensic Medicine and Toxicology, Manipal Academy of Higher Education, Mangalore, India. [519]Kasturba Medical College, Mangalore, Manipal Academy of Higher Education, Manipal, India. [520]Academic Public Health England, Public Health England, London, UK. [521]WHO Collaborating Centre for Public Health Education and Training, Imperial College London, London, UK. [522]University College London Hospitals, London, UK. [523]School of Health, Medical and Applied Sciences, CQ University, Sydney, New South Wales, Australia. [524]Department of Computer Science, Boston University, Boston, MA, USA. [525]Department of Mathematical Demography & Statistics, International Institute for Population Sciences, Mumbai, India. [526]School of Nursing and Midwifery, Royal College of Surgeons in Ireland - Bahrain, Muharraq Governorate, Bahrain. [527]School of Social Sciences and Psychology, Western Sydney University, Penrith, New South Wales, Australia. [528]Translational Health Research Institute, Western Sydney University, Penrith, New South Wales, Australia. [529]Department of Health Information Management, Manipal Academy of Higher Education, Manipal, India. [530]Department of Medical Microbiology, University of Pretoria, Pretoria, South Africa. [531]Network of Immunity in Infection, Malignancy and Autoimmunity (NIIMA), Universal Scientific Education and Research Network (USAERN), Tehran, Iran. [532]Pediatric Infectious Diseases Research Center, Mazandaran University of Medical Sciences, Sari, Iran. [533]Cardiovascular Diseases Research Center, Birjand University of Medical Sciences, Birjand, Iran. [534]Epidemiology Research Unit Institute of Public Health (EPIUnit-ISPUP), University of Porto, Porto, Portugal. [535]Department of Surgery, University of Minnesota, Minneapolis, MN, USA. [536]Department of Surgery, University Teaching Hospital of Kigali, Kigali, Rwanda. [537]Research Department, Faculty of Medical Sciences, National University of Caaguazu, Coronel Oviedo, Paraguay. [538]Department of Research and Publications, National Institute of Health, Asunción, Paraguay. [539]Department of Clinical Research, Federal University of Uberlândia, Uberlândia, Brazil. [540]School of Medicine, Gonabad University of Medical Sciences, Gonabad, Iran. [541]Department of Biomedical Sciences, University of Sassari, Sassari, Italy. [542]Department of Internal Medicine, University of Botswana, Gaborone, Botswana. [543]Heart and Vascular Institute, Cleveland Clinic, Cleveland, OH, USA. [544]Department of Cardiovascular Medicine, Mayo Clinic, Rochester, MN, USA. [545]Department of Epidemiology, Shahid Beheshti University of Medical Sciences, Tehran, Iran. [546]Department of Psychiatry, All India Institute of Medical Sciences, New Delhi, India. [547]Halal Research Center of IRI, Food and Drug Administration of the Islamic Republic of Iran, Tehran, Iran. [548]Neurogenic Inflammation Research Center, Mashhad University of Medical Sciences, Mashhad, Iran. [549]Department of Phytochemistry, Soran University, Soran, Iraq. [550]Department of Nutrition, Cihan University-Erbil, Erbil, Iraq. [551]Department of Anatomical Sciences, Kermanshah University of Medical Sciences, Kermanshah, Iran. [552]Department of Microbiology, Central University of Punjab, Bathinda, India. [553]Urology Department, Cairo University, Cairo, Egypt. [554]Public Health and Community Medicine Department, Cairo University, Giza, Egypt. [555]Center for Health Policy & Center for Primary Care and Outcomes Research, Stanford University, Stanford, CA, USA. [556]Department of Entomology, Ain Shams University, Cairo, Egypt. [557]Department of Community Medicine, PSG Institute of Medical Sciences and Research, Coimbatore, India. [558]PSG-FAIMER South Asia Regional Institute, Coimbatore, India. [559]Department of Health and Society, Faculty of Medicine, University of Applied and Environmental Sciences, Bogota, Colombia. [560]National School of Public Health, Carlos III Health Institute, Madrid, Spain. [561]Department of Community Medicine, Mahatma Gandhi Memorial Medical College, Indore, India. [562]Faculty of Infectious and Tropical Diseases, London School of Hygiene & Tropical Medicine, London, UK. [563]Colorectal Research Center, Iran University of Medical Sciences, Tehran, Iran. [564]Department of Geriatrics and Long Term Care, Hamad Medical Corporation, Doha, Qatar. [565]Faculty of Health & Social Sciences, Bournemouth University, Bournemouth, UK. [566]Population Health Research Institute, McMaster University, Hamilton, Ontario, Canada. [567]Department of Psychology, University of Alabama at Birmingham, Birmingham, AL, USA. [568]Emergency Department, Manian Medical Centre, Erode, India. [569]Population Health Sciences Institute, Newcastle University, Newcastle Upon Tyne, UK. [570]Department of Health Services Management, Iran University of Medical Sciences, Tehran, Iran. [571]Health Policy Research Center, Shiraz University of Medical Sciences, Shiraz, Iran. [572]Public Health Division, An-Najah National University, Nablus, Palestine. [573]Independent Consultant, Karachi, Pakistan. [574]Neurology Department, Ain Shams University, Cairo, Egypt. [575]School of Medicine, Alborz University of Medical Sciences, Karaj, Iran. [576]Department of Sports Medicine and Rehabilitation, Kermanshah University of Medical Sciences, Kermanshah, Iran. [577]Faculty of Caring Science, Work Life and Social Welfare, University of Borås, Borås, Sweden. [578]HIV/STI Surveillance Research Center and WHO Collaborating Center for HIV Surveillance, Kerman University of Medical Sciences, Kerman, Iran. [579]Centre for Medical Informatics, University of Edinburgh, Edinburgh, UK. [580]Division of General Internal Medicine, Harvard University, Boston, MA, USA. [581]Health Information Management, Iran University of Medical Sciences, Tehran, Iran. [582]Department of Community Medicine, Manipal Academy of Higher Education, Manipal, India. [583]National Institute of Infectious Diseases, Tokyo, Japan. [584]College of Medicine, Yonsei University, Seoul, South Korea. [585]Cancer Research Institute, Tehran University of Medical Sciences, Tehran, Iran. [586]Cancer Biology Research Center, Tehran University of Medical Sciences, Tehran, Iran. [587]Department of Health Education and Health Promotion, Kermanshah University of Medical Sciences, Kermanshah, Iran. [588]School of Health, University of Technology Sydney, Sydney, New South Wales, Australia. [589]Department of Medicine, Dow University of Health Sciences, Karachi, Pakistan. [590]Department of Dermatology, George Washington University, Washington, DC, USA. [591]Department of Law, Economics, Management and Quantitative Methods, University of Sannio, Benevento, Italy. [592]WSB University in Gdańsk, Gdansk, Poland. [593]School of Medicine, University of Alabama at Birmingham, Birmingham, AL, USA. [594]Medicine Service, USA Department of Veterans Affairs (VA), Birmingham, AL, USA. [595]Department of Epidemiology, School of Preventive Oncology, Patna, India. [596]Department of Epidemiology, Healis Sekhsaria Institute for Public Health, Mumbai, India. [597]Program Services Unit, Pathfinder International, Addis Ababa, Ethiopia. [598]Nursing Care Research Center, Semnan University of Medical Sciences, Semnan, Iran. [599]Department of Infectious Diseases, Kharkiv National Medical University, Kharkiv, Ukraine. [600]Hull York Medical School, University of Hull, Hull, UK. [601]Department of Parasitology and Mycology, Tabriz University of Medical Sciences, Tabriz, Iran. [602]Division of Community Medicine, International Medical University, Kuala Lumpur, Malaysia. [603]Nursing, Muhammadiyah University of Surakarta, Surakarta, Indonesia. [604]Department of Community Medicine, Ahmadu Bello University, Zaria, Nigeria. [605]Department of Agriculture and Food Systems, University of Melbourne, Melbourne, Victoria, Australia. [606]Department of Statistics, Manonmaniam Sundaranar University, Abishekapatti, India. [607]National Institute of Epidemiology, Indian Council of Medical Research, Chennai, India. [608]Research Center for Molecular Medicine, Hamadan University of Medical Sciences, Hamadan, Iran. [609]Non-communicable Diseases Research Center, Hamadan University of Medical Sciences, Hamadan, Iran. [610]University Institute 'Egas Moniz', Monte da Caparica, Portugal. [611]Research Institute for Medicines, University of Lisbon, Lisbon, Portugal. [612]Department of Public Health, Kurdistan University of Medical Sciences, Sanandaj, Iran. [613]School of Public Health, University of Adelaide, Adelaide, South Australia, Australia. [614]Department of Environmental Health, Wollo University, Dessie, Ethiopia. [615]Department of Community and Family Medicine, Iran University of Medical Sciences, Tehran, Iran. [616]Department of Public Health, Adigrat University, Adigrat, Ethiopia. [617]Department of Pharmacognosy, Mekelle University, Mekelle, Ethiopia. [618]Department of Medical Microbiology, University of Gondar, Gondar, Ethiopia. [619]Department of Public Health and Community Medicine, Central University of Kerala, Kasaragod, India. [620]Institute of Public Health, Jagiellonian University Medical College, Kraków, Poland. [621]Agency for Health Technology Assessment and Tariff System, Warsaw, Poland. [622]Department of Pathology and Legal Medicine, University of São Paulo, Ribeirão Preto, Brazil. [623]Modestum, London, UK. [624]Department of Health Economics, Hanoi Medical University, Hanoi, Vietnam. [625]Institute for Physical Activity and Nutrition, Deakin University, Melbourne, Queensland, Australia. [626]School of Health and Rehabilitation Sciences, The University of Queensland, Brisbane, Queensland, Australia. [627]Department of Allied Health Sciences, Iqra National University, Peshawar, Pakistan. [628]Department of Community Medicine, Alex Ekwueme Federal University Teaching Hospital Abakaliki, Abakaliki, Nigeria. [629]Kasturba Medical College, Manipal Academy of Higher Education, Mangalore, India. [630]Amity Institute of Biotechnology, Amity University Rajasthan, Jaipur, India. [631]Alzahra Teaching Hospital, Tabriz University of Medical Sciences, Tabriz, Iran. [632]Women's Reproductive Health Research Center, Tabriz University of Medical Sciences, Tabriz, Iran. [633]Clinical Cancer Research Center, Milad General Hospital, Tehran, Iran. [634]Department of Microbiology, Islamic Azad University, Tehran, Iran. [635]Argentine Society of Medicine, Buenos Aires, Argentina. [636]Velez Sarsfield Hospital, Buenos Aires, Argentina. [637]Psychosocial Injuries Research Center, Ilam University of Medical Sciences, Ilam, Iran. [638]Occupational Health Unit, Sant'Orsola Malpighi Hospital, Bologna, Italy. [639]Department of Economics, University of Göttingen, Göttingen, Germany. [640]Foundation University Medical College, Foundation University Islamabad, Islamabad, Pakistan. [641]Department of Statistics, University of Washington, Seattle, WA, USA. [642]Department of Biostatistics, University of Washington, Seattle, WA, USA. [643]Department of Epidemiology and Biostatistics, Wuhan University, Wuhan, China. [644]Demographic Change and Aging Research Area, Federal Institute for Population Research, Wiesbaden, Germany. [645]Competence Center of Mortality-Follow-Up of the German National Cohort, Federal Institute for Population Research, Wiesbaden, Germany. [646]Department of Physical Therapy, Naresuan University, Phitsanulok, Thailand. [647]School of Pharmacy, Aksum University, Aksum, Ethiopia. [648]Department of Pharmacology and Toxicology, Mekelle University, Mekelle, Ethiopia. [649]Department of Pharmacology, Addis Ababa University, Addis Ababa, Ethiopia. [650]Department of Orthopaedics, Wenzhou Medical University, Wenzhou, China. [651]Psychology Department, University of Sheffield, Sheffield, UK. [652]Department of Diabetes and Metabolic Diseases, University of Tokyo, Tokyo, Japan. [653]School of International Development and Global Studies, University of Ottawa, Ottawa, Ontario, Canada. [654]The George Institute for Global Health, University of Oxford, Oxford, UK. [655]Health Services Management Research Center, Kerman University of Medical Sciences, Kerman, Iran. [656]Department of Health Management, Policy, and Economics, Kerman University of Medical Sciences, Kerman, Iran. [657]School of Nursing, Hawassa University, Hawassa, Ethiopia. [658]Pediatrics Department, University of Jos, Jos, Nigeria. [659]Department of Pediatrics, Jos University Teaching Hospital, Jos, Nigeria. [660]Centre for Suicide Research and Prevention, University of Hong Kong, Hong Kong, China. [661]Department of Social Work and Social Administration, University of Hong Kong, Hong Kong, China. [662]Department of Neuropsychopharmacology, National Center of Neurology and Psychiatry, Kodaira, Japan. [663]Department of Public Health, Juntendo University, Tokyo, Japan. [664]Department of Health Policy and Management, Jackson State University, Jackson, MS, USA. [665]School of Medicine, Tsinghua University, Beijing, China. [666]Department of Environmental Health, Mazandaran University of Medical Sciences, Sari, Iran. [667]Injury Prevention and Safety Promotion Research Center, Shahid Beheshti University of Medical Sciences, Tehran, Iran. [668]Duke Global Health Institute, Duke University, Durham, NC, USA. [669]Social Determinants of Health Research Center, Ardabil University of Medical Science, Ardabil, Iran. [670]The School of Clinical Sciences at Monash Health, Monash University, Melbourne, Victoria, Australia. [671]Student Research Committee, Babol University of Medical Sciences, Babol, Iran. [672]Department of Community Medicine, Ardabil University of Medical Science, Ardabil, Iran. [673]Department of Health Education, Tarbiat Modares University, Tehran, Iran. [674]College of Medicine and Health Sciences, Dilla University, Dilla, Ethiopia. [675]Public Health Department, University of Edinburgh, Edinburgh, UK. [676]School of Public Health, Wuhan University of Science and Technology, Wuhan, China. [677]Hubei Province Key Laboratory of Occupational Hazard Identification and Control, Wuhan University of Science and Technology, Wuhan, China. [678]School of Medicine, Wuhan University, Wuhan, China. [679]School of Biology and Pharmaceutical Engineering, Wuhan Polytechnic University, Wuhan, China. [680]School of Health Sciences, Wuhan University, Wuhan, China. [681]National Center for Chronic and Noncommunicable Disease Control and Prevention, Chinese Center for Disease Control and Prevention, Beijing, China. [682]These authors jointly supervised this work: Simon I. Hay, Stephen S. Lim, Jonathan F. Mosser. ✉e-mail: sihay@uw.edu; stevelim@uw.edu; jmosser@uw.edu

## Methods

### Data reporting

As this is a modelling study, no statistical methods were used to predetermine sample size, the experiments were not randomized and the investigators were not blinded to allocation during experiments and outcome assessment.

### Overview

Building from our previous study of diphtheria–tetanus–pertussis vaccination coverage in Africa[14], we fitted a geostatistical model with correlated errors across space and time to predict $5 \times 5$-km$^2$ level estimates of MCV1 coverage from 2000 to 2019 using a suite of geospatial and national-level covariates for 101 LMICs. This overall process has been summarized in Extended Data Fig. 1. We spatially aggregated estimates using population-weighted averages to second administrative units from a modified version of the Database of Global Administrative Units (GADM), referred to as districts, and performed post hoc analyses to assess geographical inequality to examine progress towards GVAP targets, absolute geographical inequality and vaccination status as a function of geographical remoteness[5]. This study is compliant with the Guidelines for Accurate and Transparent Health Estimates Reporting (GATHER) recommendations[51] (Supplementary Table 1).

We defined routine MCV1 coverage as evidence of receipt of at least one dose of a MCV from either a home-based record (HBR) or parental recall among the target population in concordance with country-specific vaccination schedules in 2019[52]. Despite our best efforts to remove doses delivered through supplemental immunizion activities (SIAs) (Supplementary Information section 1.3.4), there is likely to be residual misclassification of some SIA doses due to the limitations of the available data, and these estimates of routine coverage should be viewed in the context of this limitation.

Countries were selected for this analysis if they were a LMIC or were a 'Decade of Vaccine' priority country with available subnational survey data on MCV1 coverage between 2000 and 2019[53]. We defined LMICs based on the socio-demographic index, a metric combining education, fertility and income to summarize development, as determined by GBD 2019[54]. For 13 countries (Bhutan, Brazil, China, Dominica, Georgia, Grenada, Libya, Oman, Palestine, Saint Lucia, Saint Vincent and the Grenadines, Seychelles and Venezuela), no available subnational vaccine coverage data met the inclusion and exclusion criteria; these countries were therefore excluded from this analysis. A full list of included countries is provided in Supplementary Table 3. Countries were assigned to one of 13 continuous geographical modelling regions. These regions were adapted from regions defined by GBD 2019, which are constructed to group countries together by epidemiological similarity and geographical proximity (Extended Data Fig. 2).

### Data

Using the Global Health Data Exchange (GHDx)[55], we identified and compiled a total of 354 population-based household surveys from 101 LMICs from 2000 to 2019 containing individual MCV1 vaccination status and subnational geolocation information. Surveys were included if they contained MCV1 coverage information and subnational geolocation, and excluded if they contained areal data and were missing key survey design variables (strata, primary sampling units and design weights), did not include children aged 12–59 months, contained no subnational individual-level geographical information or if coverage estimates were implausible (Supplementary Tables 4, 5).

Coverage was computed at the cluster level when global positioning system (GPS) data were available. If GPS information was not collected or was not available, we calculated mean coverage at the most-granular geographical area possible while accounting for sampling weights and survey design. These aggregated coverage estimates were then included in the geospatial modelling process using a

previously described method[14,56] that leverages population weights and a $k$-means clustering algorithm to propose a set of GPS coordinates as a proxy for locations where survey data collection could have occurred (Supplementary Fig. 3). These coordinates were then used to represent the areal data in the geospatial model. The following data were extracted from each survey source: vaccine card or HBR doses, parental recall vaccine doses, age (in months), survey weight and design variables, and GPS cluster or areal location. Individuals with evidence of vaccination either from HBR or recall were considered to have been vaccinated. Individuals were excluded from the analysis if they were missing age, spatial or survey design information or were outside of the study age or year range. The study included all years between 2000 and 2019. A comprehensive overview of data from all study geographies included can be found in Supplementary Figs. 1, 2.

Individual age, in months, at the time of survey collection was used to assign each child to a birth cohort (12–23 months, 24–35 months, 36–47 months and 48–59 months). Data corresponding to each birth cohort were included in the modelling process in the year in which that birth cohort was aged 0–12 months old. For countries recommending MCV1 within the first year of life, we included data from children aged 12–47 months. If the first dose was not recommended until the second year of life, we included data from children aged 24–59 months. A full list of schedules by country can be found in Supplementary Table 2. This yielded a dataset of 1,697,570 total children. This method allows the inclusion of additional individuals, which increases overall geographical representation but requires assumptions such as negligible catch-up vaccination and no differential mortality or migration. However, the overall influence of including older cohorts in our model on the key findings appeared to be minor (Supplementary Information section 2.2). Additional information on the benefits and limitations of this approach can be found in the Supplementary Information sections 1.3.4, 2.2, Supplementary Figs. 18–20 and Supplementary Tables 14, 15.

We included 26 geospatial covariates as possible predictors of MCV1 coverage in the modelling process, including maternal education, access to major cities or settlements, a binary urban or rural indicator, total population and a suite of 22 environmental covariates (Supplementary Fig. 4 and Supplementary Table 6). Four national-level covariates were also included: lag-distributed income, prevalence of the completion of the fourth antenatal care visit among pregnant women, mortality due to war and terror, and bias-adjusted national-level administrative data on MCV1 coverage reported through the WHO/UNICEF Joint Reporting Form (Supplementary Information section 1.5.1). For each region, an optimized set of geospatial covariates was selected from these 26 possible covariates, using a variance inflation factor (VIF) algorithm[57] in which covariates were selected with a VIF < 3. This method was used to ensure non-collinearity between covariates within each region to facilitate model convergence. Selected covariates varied by region (Supplementary Table 7).

Other spatial data used in our analyses included gridded population estimates, administrative boundaries and gridded estimates of travel time to major cities or settlements. These sources are described in detail in Supplementary Information section 1.3.

### Geostatistical model

First, stacked generalization was used to capture potential nonlinear and complex relationships between covariates and vaccination coverage. This approach has previously been shown to increase the predictive accuracy of geospatial models[58]. Using the optimized set of covariates selected for each region by the VIF algorithm, three different child models—generalized additive models, lasso regression and boosted regression trees—were fit, with each model predicting MCV1 coverage as the outcome of interest. When fitting boosted regression trees, country-level fixed effects were included to allow relationships between coverage and covariates to differ by country. In this initial modelling step, there were no explicitly defined temporal or spatial

effects included in the models beyond those inherently present in the covariate patterns and correlations between covariates.

Each child model was fit using fivefold cross-validation to avoid overfitting. This generated out-of-sample predictions of coverage for each location and year per region. Each model in each region was also fit using the full set of vaccine coverage outcome data, which yielded a corresponding set of in-sample predictions. The predictions of MCV1 coverage obtained from each child model were in turn used as predictors in the second-step geostatistical model described below. Out-of-sample predictions from each child model were used as explanatory covariates when fitting the geostatistical model. In-sample predictions from each model and region were used when generating predictions from the fitted geostatistical model.

After the first step (stacked generalization), a second-step Bayesian geostatistical modelling framework was used to model vaccination coverage as counts in a binomial space with a logit link through a generalized linear regression with explicit spatial and temporal terms. This second step leverages the covariate relationships estimated through stacked generalization while also accounting for additional correlation in coverage across space and time.

A separate model of MCV1 coverage was fit for each of the 13 regions as defined below:

$$C_d | p_{i(d),t(d)}, N_d \sim \text{Binomial}(p_{i(d),t(d)}, N_d) \forall \text{ observed clusters } d$$

$$\text{logit}(p_{i,t}) = \beta_0 + X_{i,t}\beta + Z_{i,t} + \epsilon_{\text{country}_{(i)}} + \epsilon_{i,t} \forall \ i \in \text{spatial domain } \forall$$
$$t \in \text{time domain}$$

$$\sum_{h=1}^{3} \beta_h = 1$$

$$\epsilon_{i,t} \sim N(0, \sigma^2_{\text{nugget}})$$

$$\epsilon_{\text{country}(i)} \sim N(0, \sigma^2_{\text{country}(i)})$$

$$Z \sim \text{GP}(0, \Sigma^{\text{space}} \otimes \Sigma^{\text{time}})$$

$$\Sigma^{\text{space}} = \frac{2^{1-\nu}\sigma^2_{\text{space}}}{\Gamma(\nu)} \times \left(\frac{\sqrt{8}}{\delta}D\right)^\nu \times K_\nu\left(\frac{\sqrt{8}}{\delta}D\right)$$

$$\Sigma^{\text{time}}_{j,k} = \rho^{|k-j|}$$

This model, adopted from widely used Bayesian hierarchical models[59,60], has been described in detail in other work[14,25,56,61,62]. In brief, this method estimates the number of children, $C$, in cluster $d$ at location $i$ and time $t$ with sample size $N$ that have been vaccinated with a specific antigen-dose combination. $p_{i(d),t(d)}$ is the proportion of children vaccinated with MCV1 among the target age population in cluster $d$. Each child model generates a prediction $X_{i,t}$ for each child model $h$. Residual terms $\epsilon_*$ are unique to each particular location in space and time across all modelled geographies and years.

In this generalized linear regression framework, the proportion of children vaccinated $p_{i,t}$ is modelled using the out-of-sample predictions of vaccine coverage $X_{i,t}$ from each of three stacked generalization child models ($h$) as explanatory variables. The $\beta_h$ coefficients are constrained to sum to 1, via the 'extraconstr' R-INLA parameter[63], to improve computational tractability[58] and are representative of the predictive weighting used in the stacking process.

$\epsilon_{\text{country}(i)}$ represents a country-level random effect. $\epsilon_{i,t}$ represents an independent nugget effect for irreducible error for a given observation, which accounts for true variation that is unable to be captured by the model and variation from measurement error. $Z_{i,t}$ represents a correlated spatiotemporal error term, for any residual autocorrelation across space and time that remains after accounting for the predictive capacity of the stacked-modelled covariates, country-specific variation in vaccine coverage and observation-specific irreducible error.

These additional spatiotemporal residuals $Z_{i,t}$ were modelled as a three-dimensional spatiotemporal Gaussian process with a mean of zero and a covariance matrix formed from the Kronecker product of spatial and temporal covariance kernels. The temporal covariance $\Sigma^{\text{time}}$ was modelled via an annual autoregressive order 1 function from all study years from 2000 to 2019, where $\rho$ is the autocorrelation function and $k$ and $j$ are points in the annual time series. The spatial covariance $\Sigma^{\text{space}}$ was assumed to be an isotropic, stationary Matérn function, where $\Gamma$ is the gamma function, $K_\nu$ is the modified Bessel function of the second kind of order $\nu > 0$, $\sigma^2_{\text{space}}$ is the marginal variance, $\nu$ is a scaling constant, $\delta$ is a range parameter with a penalized complexity prior, and $D$ is a spatial distance matrix[64], measured along the great circle in kilometres. The generalized linear model was fitted using an integrated nested Laplace approximation in R-INLA with a stochastic partial differential equation (SPDE) solver in package SPDE[65]. Additional detailed information on priors, spatial mesh construction and model fitting is provided in Supplementary Information sections 1.4.2–1.4.6 and Supplementary Fig. 5. This process produces a set of 1,000 posterior draws, each representing an estimate of vaccine coverage for each location and year— in other words, a set of 1000 candidate maps of coverage from 2000 to 2019.

## Post-estimation

To leverage data from additional national-level sources, including administrative data, and maintain internal consistency, the set of candidate maps was calibrated to MCV1 coverage estimates produced for GBD 2019. This post hoc calibration preserves the overall spatial variation of estimates, while ensuring that the population-weighted averages of the geospatial estimates are equivalent to those produced by GBD[4]. This step allows for the calibrated estimates to reflect information from data sources that are only available at the national level, such as surveys for which no subnational data are available, which are included in the GBD estimates but excluded from the geospatial model described in the 'Geostatistical model' section. A description of the estimation of MCV1 coverage for GBD 2019 can be found in Supplementary Information section 1.5.1.

In this calibration process, each 5 × 5-km² pixel in each modelled region was first assigned to a second-level administrative unit. In locations in which boundary definitions transect a given pixel, the fraction of area of that pixel belonging to each overlapping second-level administrative unit was calculated. Because of the nested hierarchy of administrative units, this additionally allowed for the assignment of pixels and partial pixels to first administrative units and countries. Assuming that the population density within each pixel was uniform, WorldPop population values of children under 5 years old were divided for each whole or partial pixel proportional to fractional area. After pixel and partial pixel populations were assigned, population-level estimates were calibrated to GBD population estimates for each country and year.

Calibration methods similar to those used in this study have been described previously[14]. To ensure vaccination coverage estimates post-calibration remained between 0 and 100%, calibration was performed in logit space such that for each country $c$ and year $t$, national-level estimates of coverage from GBD ($V_{\text{GBD},c,t}$) and population-weighted national averages of coverage from the model-based geostatistical (MBG) model ($V_{\text{MBG},c,t}$) can be related via a country-year-specific calibration factor ($k_{c,t}$) in the following equation:

$$\text{logit}(V_{\text{GBD},c,t}) = \text{logit}(V_{\text{MBG},c,t}) + k_{c,t}$$

Calibration factors were applied to each $5 \times 5$-$km^2$ pixel and partial pixel per draw per country-year. Pixels that were fractionally assigned to multiple countries were combined using a weighted average proportional to the fraction of each area. This process resulted in a set of calibrated draw-level estimates of vaccination coverage, which were used for all subsequent analyses.

Population-weighted averages of coverage for each pixel or partial pixel within a first or second administrative unit were then calculated. Fractional pixel membership was determined as described above. This process was repeated for each of the 1,000 posterior pixel-level draws, which yielded 1,000 posterior draws of MCV1 coverage per administrative unit per year. Estimates for first and second administrative units with uncertainty were derived from mean, 2.5th and 97.5th percentiles.

## Model validation
We assessed the predictive performance of the models using fivefold out-of-sample cross-validation. We stratified data by first and second administrative units and ran models leaving out one-fifth of the spatially stratified data at a time. Predicted estimates of MCV1 coverage were then compared to the withheld observed data by calculating the mean error, root mean square error, correlation and other predictive validity metrics for all years for which survey data were available (2000–2018). Fitted model parameters can be found in Supplementary Table 8. Metrics and validity figures can be found in Supplementary Tables 9–12 and Supplementary Figs. 6–13, respectively. Additional information regarding uncertainty of estimates can be found in Supplementary Figs. 14–17.

## Post hoc geospatial inequality analyses
Lorenz curves were generated using the relationship between the number of children and the number of vaccinated children for each pixel. Pixel-level Gini coefficients were calculated for 2000 and 2019 from corresponding Lorenz curves[66,67] (Supplementary Table 13). Absolute geographical inequality per country was calculated from the national-level Gini coefficients and national MCV1 coverage using the following formula:

$$\text{Absolute geographical inequality} = 2 \times \text{coverage} \times \text{Gini}$$

We chose to use the absolute geographical inequality metric to represent inequality over the Gini coefficient alone. As the mean is related to Gini, we wanted to account for this relationship. Estimates are scaled by 2 as this puts the absolute geographical inequality coefficient back to the same scale as the mean[68].

Additionally, we assessed vaccination status as a function of geographical remoteness. Using a gridded surface of travel time to major cities or settlements, we classified each $5 \times 5$-$km^2$ pixel as remote rural, urban or neither[29]. Pixels with travel times of less than 30 min were classified as urban, and pixels with travel times greater than 3 h were classified as remote rural. Overlaid with a gridded population surface from WorldPop[30], the number of unvaccinated children per pixel was also calculated.

We constructed concentration curves of the cumulative proportion of unvaccinated children as well as plots of MCV1 coverage by travel time to assess patterns across countries and regions. Country-specific concentration curves of the cumulative proportion of unvaccinated children as a function of travel time for select countries are shown in Extended Data Fig. 10. Summary metrics, such as the proportion of unvaccinated individuals living in each urban and remote rural location, were computed.

## Limitations
This work is subject to several limitations. First, the primary data used in this analysis came from child-level survey data with varying degrees of representativeness, consistency, accuracy and comparability, from both HBR and parental recall[69,70]. The magnitude and direction of recall bias varies, and we therefore were unable to correct for it[71]. We estimate coverage using data from children aged 12–59 months, and while we accounted for target age at vaccination, this does not fully account for differential mortality due to vaccine status or catch-up vaccination. We aim to estimate routine coverage and have excluded doses delivered via SIAs from the analysed survey data wherever possible (Supplementary Information section 1.3.4), but misclassification of SIA doses is still likely, particularly in cases of parental recall—especially for older children—and in cases in which survey methodology does not distinguish clearly between SIA and routine doses.

In data-sparse areas for which covariate relationships may not fully capture coverage patterns, results may be biased. Additionally, data representativeness among vulnerable populations, such as those living in urban slums or migrant populations, might vary due to data collection in survey design. We include as much data on MCV1 coverage as possible, including data that are only geo-resolved to areal locations. The methodology that we used to assign areal data to specific locations for modelling could lead to oversmoothing in final estimates, obscure relationships between coverage and covariates, and underestimate uncertainty, but this method has been shown to have a higher predictive validity compared with the exclusion of the data[72]. Limitations due to data availability should not be taken lightly and should reinforce to stakeholders and policymakers the need for additional resources to collect high-quality data that are representative of all populations, especially the most vulnerable for being unvaccinated, and to increase the quality of routinely collected subnational administrative data.

Because the estimates that we used to assess geographical remoteness in post hoc analyses were also used as spatial covariates in the geospatial model, these results are limited by the possibility of circularity and subsequent confounding. In addition, we used a stacked generalization method to allow for complex and nonlinear relationships between covariates and vaccination coverage. These methods are optimized for prediction, not causal inference. For that reason, these results cannot be used to identify the specific effect of any particular covariate on MCV1 coverage. In addition, owing to limitations in the underlying data and computational feasibility, we were unable to incorporate several potentially important sources of uncertainty into this analysis, including from covariates, population estimates, the incorporation of areal data and the stacked generalization process.

We fitted our geostatistical models using R-INLA, as opposed to a full Markov chain Monte Carlo sampler. Although using a more traditional Bayesian model fitting approach that takes true samples from the posterior typically results in increased parameter identifiability, the Laplace approximation approach used by R-INLA is more computationally feasible given our current modelling scale. Our model is separable, yet symmetric, across time and space. This approach assumes that, for each region, the covariance has the same functional form between years and locations regardless of the locations themselves; the use of a non-separable covariance function could relax these assumptions[73,74]. However, owing to the additional computational challenges associated with fitting a non-separable model, as well as data sparsity in several regions throughout space and time, we determined that fitting a non-separable model would be challenging and complex, and would probably yield little benefit compared to our current modelling approach.

In some settings with high levels of natural immunity (derived from previous infection), greater than 95% vaccination coverage may not be required to prevent disease transmission[75]. These estimates only focus on the first routine dose of MCV, and immunity can also be obtained through later vaccination via SIA or natural infection. In an ideal long-term measles elimination scenario, all immunity would be vaccine-derived, and no natural infections would occur. A 95% coverage target for routine immunization, therefore, still has practical programmatic relevance.

Finally, our study describes spatial heterogeneity in coverage, but not pockets of low coverage within social or age groupings that can facilitate ongoing disease transmission, particularly in densely populated areas, despite nominally high average vaccine coverage[76]. Although these results provide a powerful tool for policymakers to identify weaknesses in routine immunization systems and plan for SIA, they should be used in conjunction with other data sources that can be used to make decisions about vaccine policy, including analyses of cost effectiveness, determinants of high or low coverage, and specific coverage initiatives to reduce disease burden.

### Reporting summary

Further information on research design is available in the Nature Research Reporting Summary linked to this paper.

### Data availability

The findings of this study are supported by data available in public online repositories and data publicly available upon request from the data provider. A detailed table of data sources and availability can be found in Supplementary Table 4 and at http://ghdx.healthdata.org/lbd-publication-data-input-sources. Administrative boundaries were modified from the Database for Global Administrative Areas (GADM) dataset[77]. Populations were retrieved from WorldPop[30], and gridded estimates of travel time to the nearest city or settlement were available online from a previously published study[29]. This study complies with the GATHER recommendations[51].

### Code availability

This study is compliant with the GATHER recommendations[51]; as such, all computer code is available from GitHub (https://github.com/ihmeuw/lbd/tree/mcv1-lmic-2020). All maps and figures presented in this study are generated by the authors using RStudio (R version 3.6.1), ArcGIS Desktop 10.6 and Python 2.7.

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

**Acknowledgements** This work was primarily supported by grants from the Bill & Melinda Gates Foundation (OPP1182474, OPP11093011 and OPP1132415). S.I.H. is funded by additional grants from the Bill & Melinda Gates Foundation (OPP1119467 and OPP1106023). The opinions expressed in this paper are those of the authors and not necessarily those of the World Health Organization. J.-W.D.N. was supported by the Alexander von Humboldt Foundation. C.H. is partially supported by a grant of the Romanian National Authority for Scientific Research and Innovation, CNDS-UEFISCDI, project number PN-III-P4-ID-PCCF-2016-0084, and a grant co-funded by the European Fund for Regional Development through Operational Program for Competitiveness, Project ID P_40_382. Y.J.K. acknowledges support by the Research Management Centre, Xiamen University Malaysia (XMUMRF/2018-C2/ITCM/0001). K. Krishan is supported by a DST PURSE Grant and UGC Centre of Advanced Study awarded to the Department of Anthropology, Panjab University, Chandigarh, India. B.L. acknowledges support from the NIHR Oxford Biomedical Research Centre and the BHF Centre of Research Excellence, Oxford. M.A.M. acknowledges NIGEB and NIMAD grants. A. Sheikh acknowledges support by Health Data Research UK. S.B.Z. acknowledges support from the Australian Government research training program (RTP) for his academic career.

**Author contributions** J.F.M., A.N.S., S.S.L. and S.I.H. conceived and planned the study. A.N.S., J.F.M., S.R., J.Q.N. and N.C.G. identified and vetted data for this analysis. S.R. and J.Q.N. extracted, processed and geo-positioned the data. A.N.S. carried out the statistical analyses with assistance and input from J.F.M., S.S.L. and C.J.L.M. A.N.S., L.E., S.R. and J.Q.N. prepared figures and tables. A.N.S. wrote the first draft of the manuscript with assistance from J.F.M, S.S.L, S.I.H. and M.K.M.-P. and all authors contributed to subsequent revisions. All authors provided intellectual input into aspects of this study.

**Competing interests** This study was funded by the Bill & Melinda Gates Foundation. Authors employed by the Bill & Melinda Gates Foundation provided feedback on initial maps and drafts of this manuscript. Otherwise, the funders of the study had no role in study design, data collection, data analysis, data interpretation, or writing of the final report. The corresponding authors had full access to all the data in the study and had final responsibility for the decision to submit for publication. O.O.A. is supported by DSI-NRF Centre of Excellence for Epidemiological Modelling and Analysis (SACEMA). C.A.T.A. reports personal fees from Johnson & Johnson (The Philippines), outside the submitted work. M.L.B. reports grants from the US Environmental Protection Agency, the National Institutes of Health (NIH) and the Wellcome Trust Foundation, during the conduct of the study. M.L.B. also reports honoraria and/or travel reimbursements from the NIH (for the review of grant proposals), *American Journal of Public Health* (participation as editor), Global Research Laboratory and Seoul National University, Royal Society London UK, Ohio University, Atmospheric Chemistry Gordon Research Conference, Johns Hopkins Bloomberg School of Public Health, Arizona State University, Ministry of the Environment Japan, Hong Kong Polytechnic University, University of Illinois–Champaign, and University of Tennessee–Knoxville. S. Basu reports grants from the NIH, grants from the US Centers for Disease Control and Prevention, grants from the US Department of Agriculture, grants from Robert Wood Johnson Foundation, personal fees from Research Triangle Institute, personal fees from Collective Health, personal fees from KPMG, personal fees from HealthRight360, personal fees from *PLOS Medicine*, personal fees from *The New England Journal of Medicine*, outside the submitted work. F.D. reports grants from the Bill & Melinda Gates Foundation, during the conduct of the study. A. Deshpande reports grants from the Bill & Melinda Gates Foundation, during the conduct of the study. S.J.D. reports grants from The Fleming Fund at the UK Department of Health & Social Care, during the conduct of the study. S.M.S.I. received funding from the National Heart Foundation of Australia. J.J.J. reports personal fees from AMGEN, personal fees from ALAB, personal fees from TEVA, personal fees from SYNEXUS, personal fees from BOEHRINGER INGELHEIM and personal fees from VALEANT, outside the submitted work. H.J.L. reports grants from GSK, outside the submitted work. W.M. is Program Analyst in Population and Development at the United Nations Population Fund (UNFPA), an institution which does not necessarily endorse this study. T. Pilgrim reports grants and personal fees from Biotronik, grants and personal fees from Boston Scientific and grants from Edwards Lifesciences, outside the submitted work. M.J.P. reports grants and personal fees from MSD, GSK, Pfizer, Boehringer Ingelheim, BMS, Novavax, Astra Zeneca, Sanofi, IQVIA and other pharmaceutical industries, personal fees from Quintiles, Novartis, Pharmerit and Seqirus, grants from Bayer, BioMerieux, WHO, EU, FIND, Antilope, DIKTI, LPDP, Budi, and other from Ingress Health, Pharmacoeconomics Advice Groningen (PAG Ltd), Asc Academics, outside the submitted work. M.J.P. holds stocks in Ingress Health and PAG Ltd and is advisor to Asc Academics, all of which are pharmacoeconomic consultancy companies, outside of submitted work. J. A. Singh reports personal fees from Crealta/Horizon, Medisys, Fidia, UBM LLC, Trio Health, Medscape, WebMD, Clinical Care Options, Clearview Healthcare Partners, Putnam Associates, Spherix, Practice Point Communications, the NIH and the American College of Rheumatology, personal fees from the speaker's bureau of Simply Speaking, stock options in Amarin Pharmaceuticals and Viking Pharmaceuticals, non-financial support from the steering committee of OMERACT, an international organization that develops measures

for clinical trials and receives arm's length funding from 12 pharmaceutical companies, outside of the submitted work. J. A. Singh serves on the FDA Arthritis Advisory Committee, is a member of the Veterans Affairs Rheumatology Field Advisory Committee, and is the editor and the Director of the UAB Cochrane Musculoskeletal Group Satellite Center on Network Meta-analysis, all outside the submitted work. R.U. reports other financial activities from Deakin University, outside the submitted work. J.F.M. reports grants from the Bill and Melinda Gates Foundation, during the conduct of the study.

**Additional information**
**Correspondence and requests for materials** should be addressed to S.I.H., S.S.L. or J.F.M.

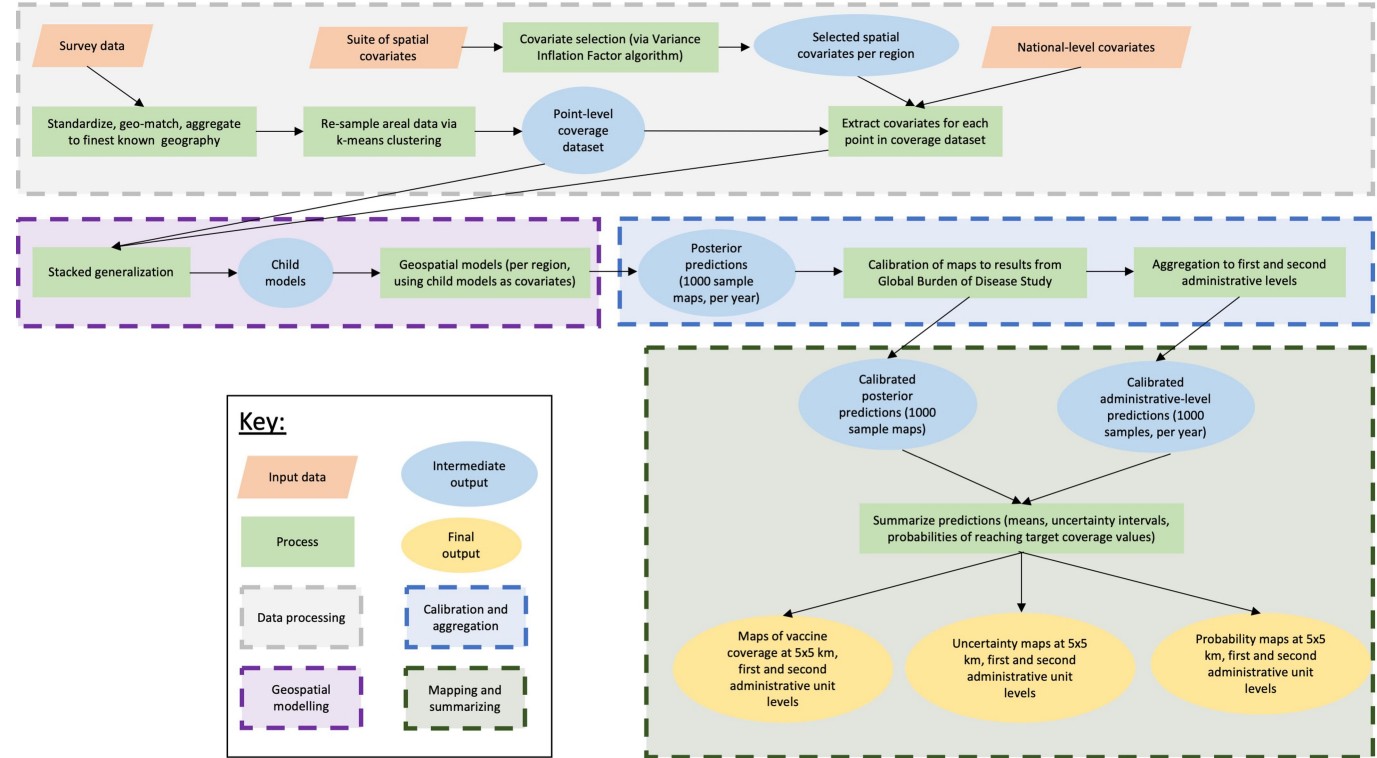

**Extended Data Fig. 1 | Data processing and geospatial modelling flowchart.** Survey data and the suite of covariates used in modelling are first compiled and processed (orange and grey). The modelling process (purple) consists of data being used in a stacked generalization ensemble modelling process via boosted regression tree, lasso and generalized additive models, fitting the second-stage spatiotemporal model using integrated nested Laplace approximation, and finally calibration to GBD estimates (blue). Steps in dark green and outputs in yellow indicate the post-estimation process when the full posterior distribution of predications is transformed to both 5 × 5-km² and first and second administrative-unit-level maps and their various final results. Intermediate outputs throughout the process are shown in blue and overall processes are shown in light green.

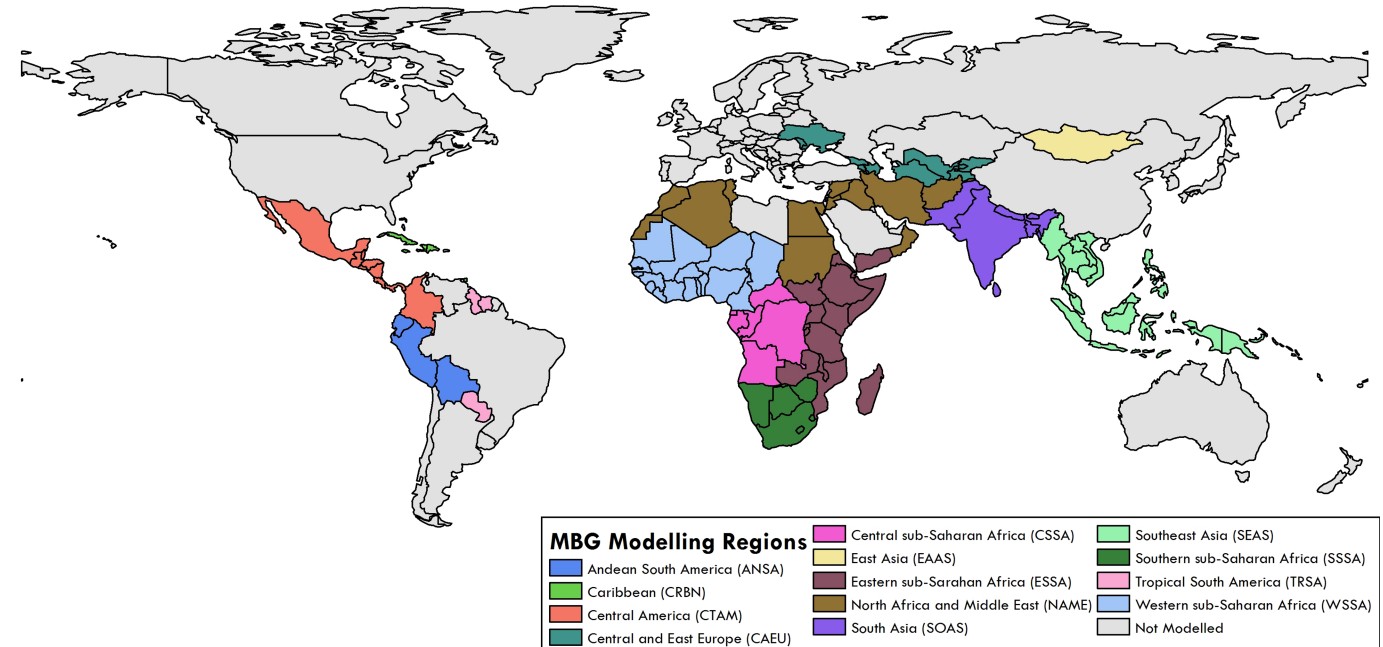

**Extended Data Fig. 2 | Regions of countries used in modelling.** Analyses were divided into 13 regions based on the GBD super-regions to allow for locations similar in data availability and patterns of vaccine coverage to be analysed using similar covariate and modelling relationships. Each colour represents a different region, as described in the legend.

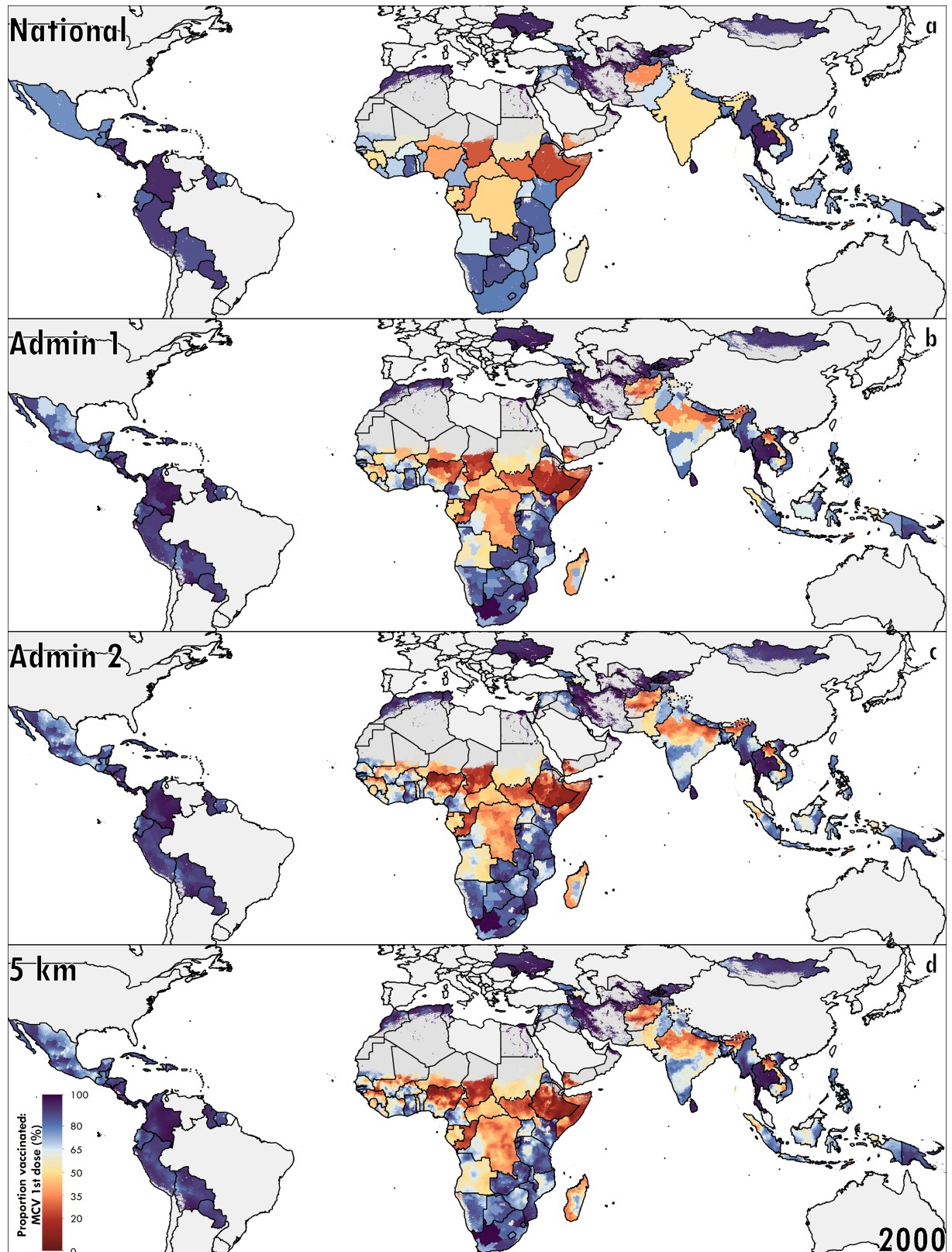

**Extended Data Fig. 3 | National, first- and second-administrative-unit level, and pixel-level MCV1 coverage, 2000. a–d**, Posterior means are represented at the national (**a**), first-administrative-unit (**b**), second-administrative-unit (**c**) and 5 × 5-km² pixel (**d**) levels. Pixels that are grey in colour are either not included in the analysis, or have been classified as being 'barren or sparsely vegetated' or had fewer than 10 people per 1 × 1-km² pixel[30,50].

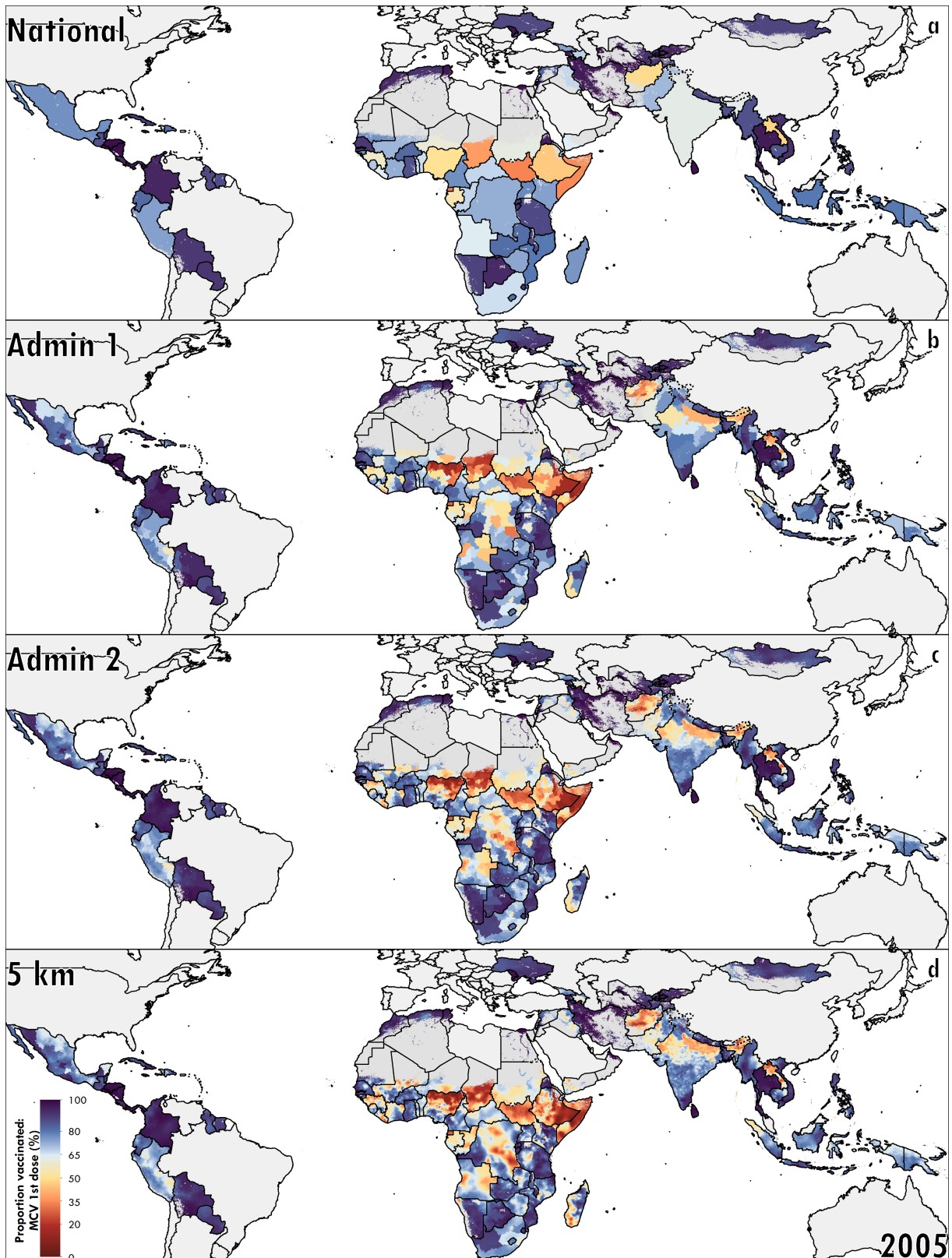

**Extended Data Fig. 4 | National, first- and second-administrative level, and pixel-level MCV1 coverage, 2005. a–d**, Posterior means are represented at the national (**a**), first-administrative-unit (**b**), second-administrative-unit (**c**) and 5 × 5-km² pixel (**d**) levels. Pixels that are grey in colour are either not included in the analysis, or have been classified as being 'barren or sparsely vegetated' or had fewer than 10 people per 1 × 1-km² pixel[30,50].

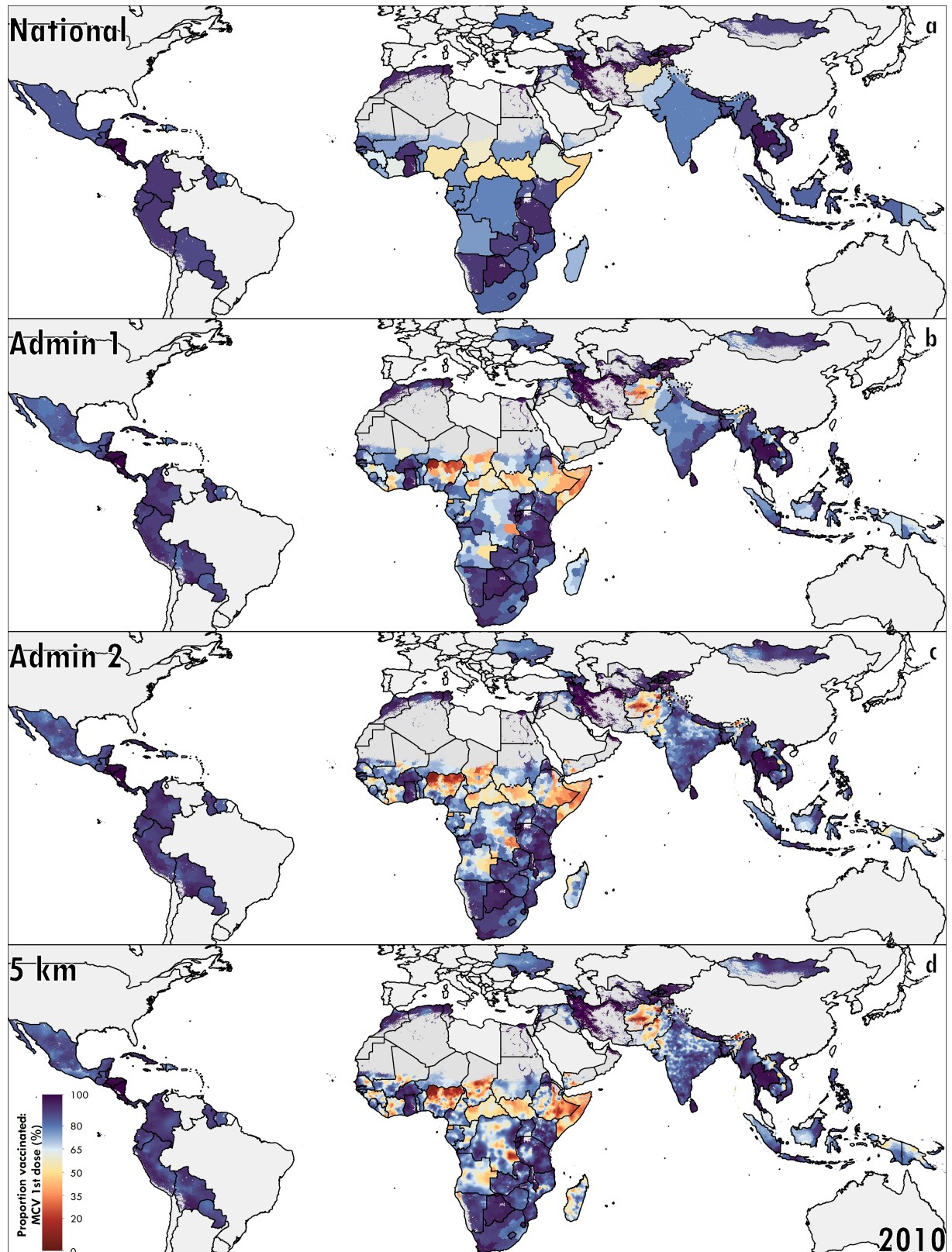

**Extended Data Fig. 5 | National, first- and second-administrative-unit level, and pixel-level MCV1 coverage, 2010. a–d,** Posterior means are represented at the national (**a**), first-administrative-unit (**b**), second-administrative-unit (**c**) and 5 × 5-km² pixel (**d**) levels. Pixels that are grey in colour are either not included in the analysis, or have been classified as being 'barren or sparsely vegetated' or had fewer than 10 people per 1 × 1-km² pixel[30,50].

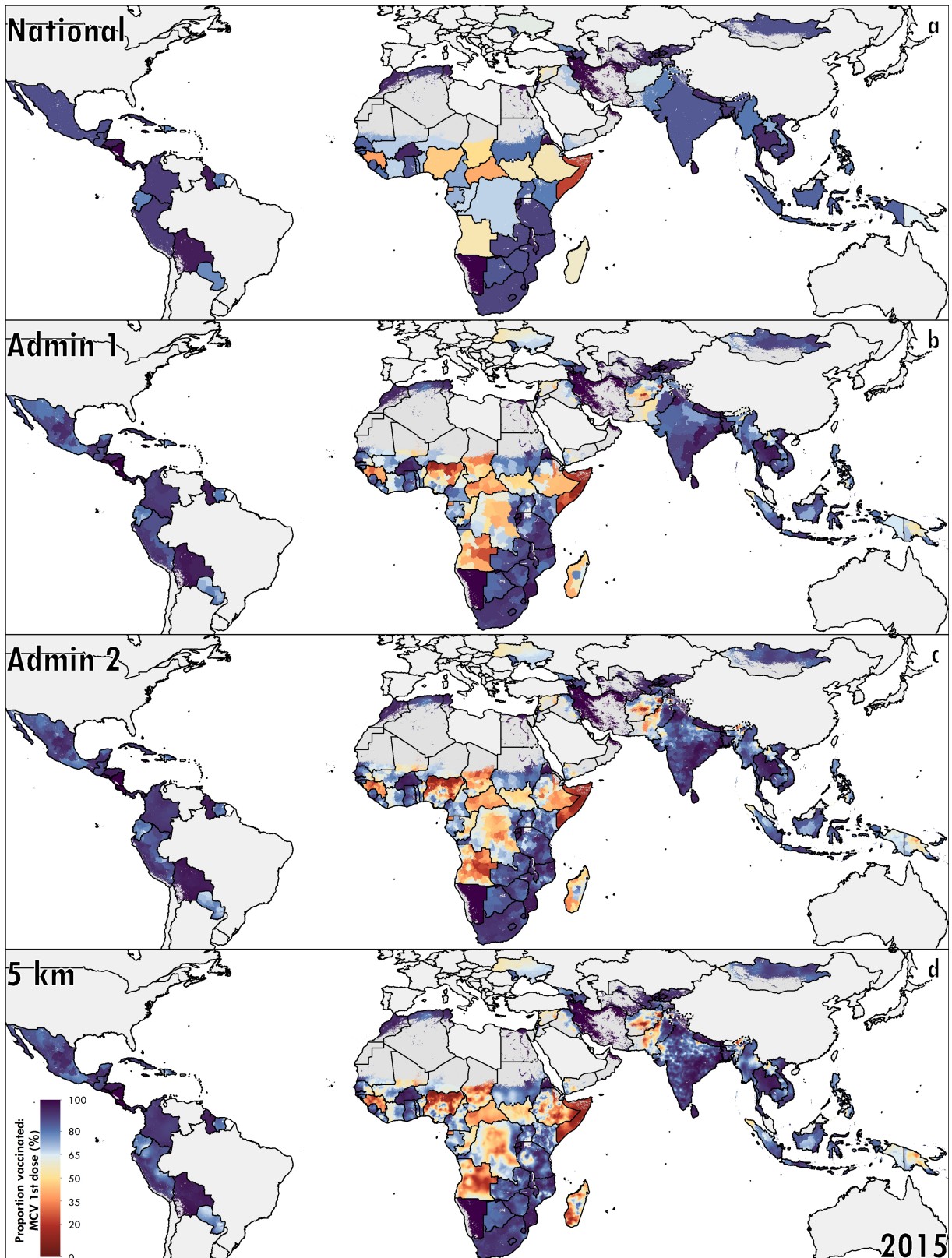

**Extended Data Fig. 6 | National, first- and second-administrative-unit level, and pixel-level MCV1 coverage, 2015. a–d**, Posterior means are represented at the national (**a**), first-administrative-unit (**b**), second-administrative-unit (**c**) and 5 × 5-km² pixel (**d**) levels. Pixels that are grey in colour are either not included in the analysis, or have been classified as being 'barren or sparsely vegetated' or had fewer than 10 people per 1 × 1-km² pixel[30,50].

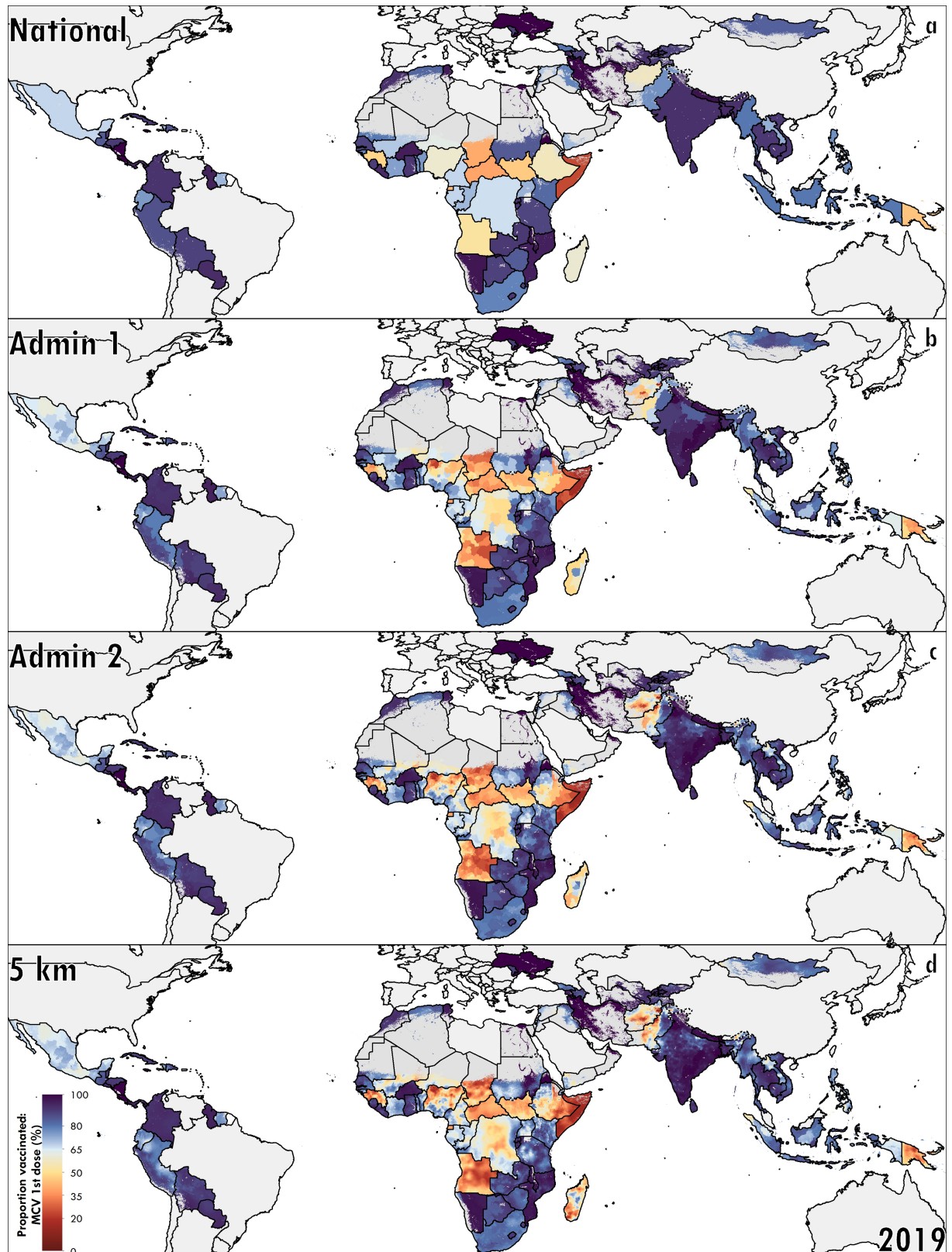

**Extended Data Fig. 7 | National, first- and second-administrative-unit level, and pixel-level MCV1 coverage, 2019. a–d,** Posterior means are represented at the national (**a**), first-administrative-unit (**b**), second-administrative-unit (**c**) and 5 × 5-km² pixel (**d**) levels. Pixels that are grey in colour are either not included in the analysis, or have been classified as being 'barren or sparsely vegetated' or had fewer than 10 people per 1 × 1-km² pixel[30,50].

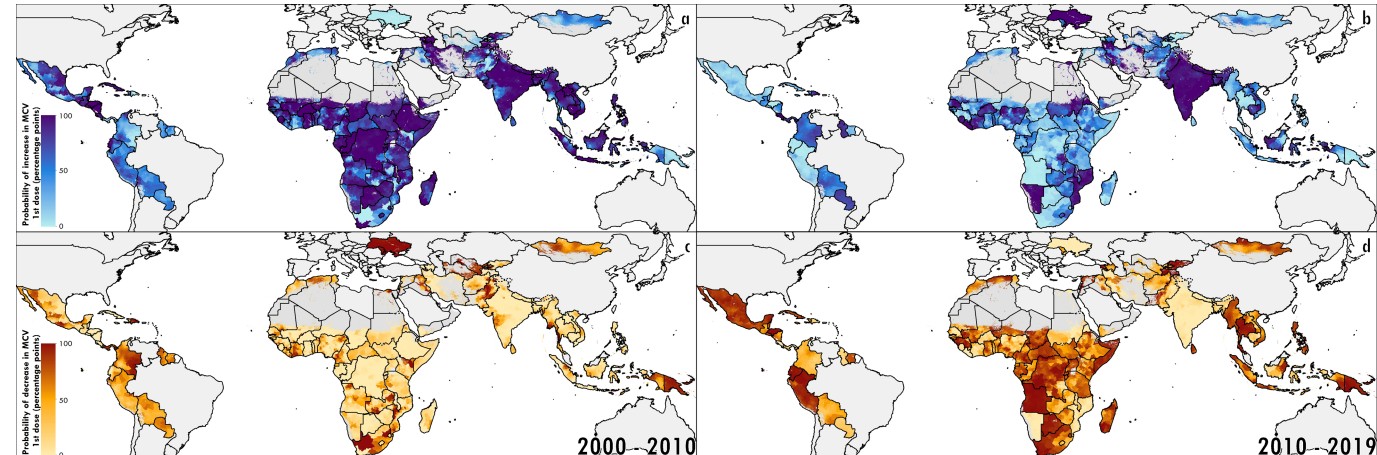

**Extended Data Fig. 8 | Probability of increased or decreased coverage from 2000 to 2010 and 2010 to 2019. a–d**, Probability of an increase in coverage in each district (**a**, **b**) and probability of decrease in coverage in each district (**c**, **d**) from 2000 to 2010 (**a**, **c**) and 2010 to 2019 (**b**, **d**).

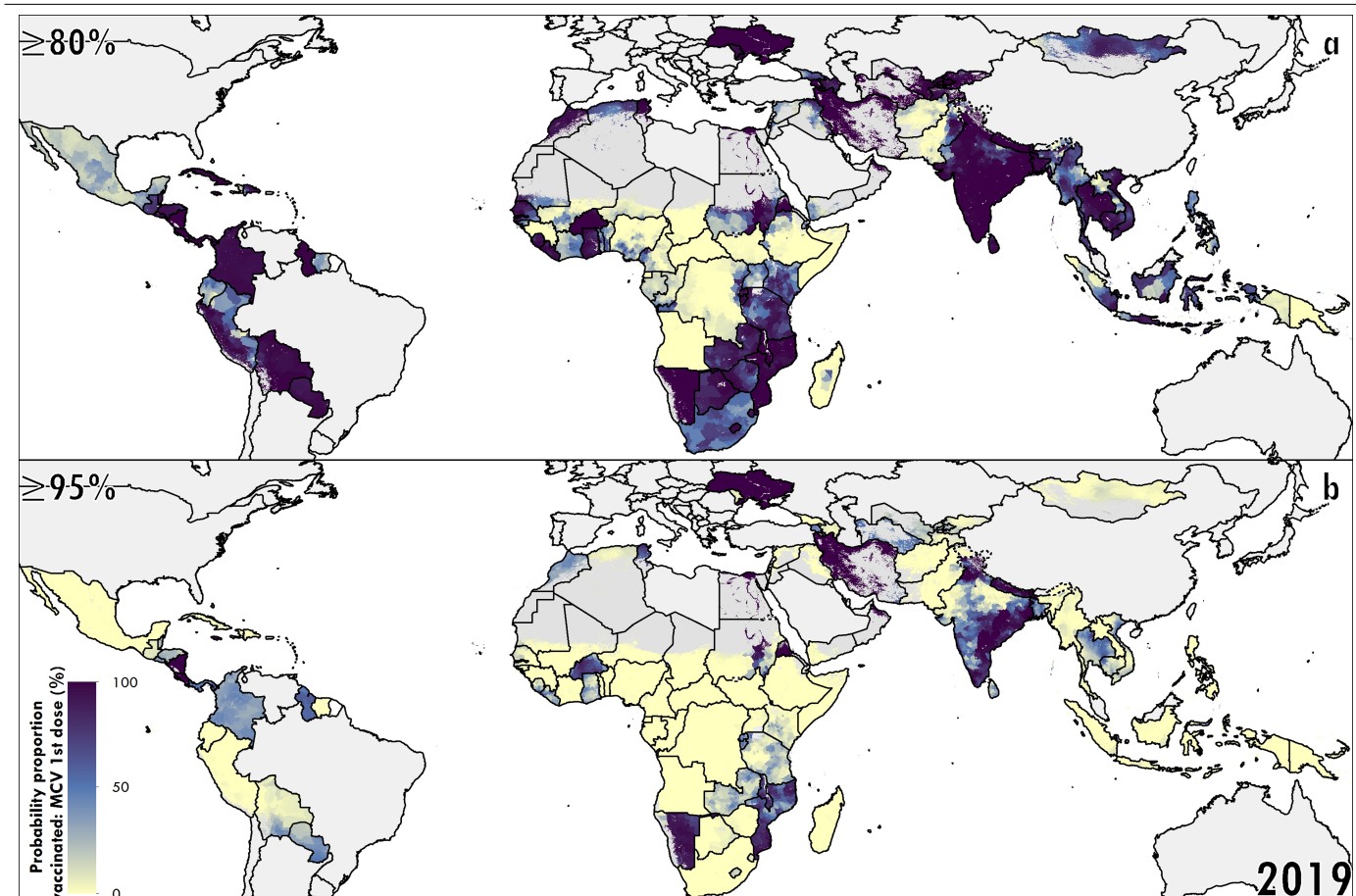

**Extended Data Fig. 9 | Estimated district-level probabilities of reaching MCV1 coverage targets in 2019. a, b,** Probability of districts having achieved 80% GVAP and Measles Rubella Initiative targets (**a**) and 95% critical proportion to immunize coverage targets to reach herd immunity (**b**). Countries excluded from the analysis and pixels classified as 'barren or sparsely vegetated' based on ESA-CCI satellite data or with fewer than 10 people per 1 × 1-km² pixel based on WorldPop estimates are masked in grey[30,50].

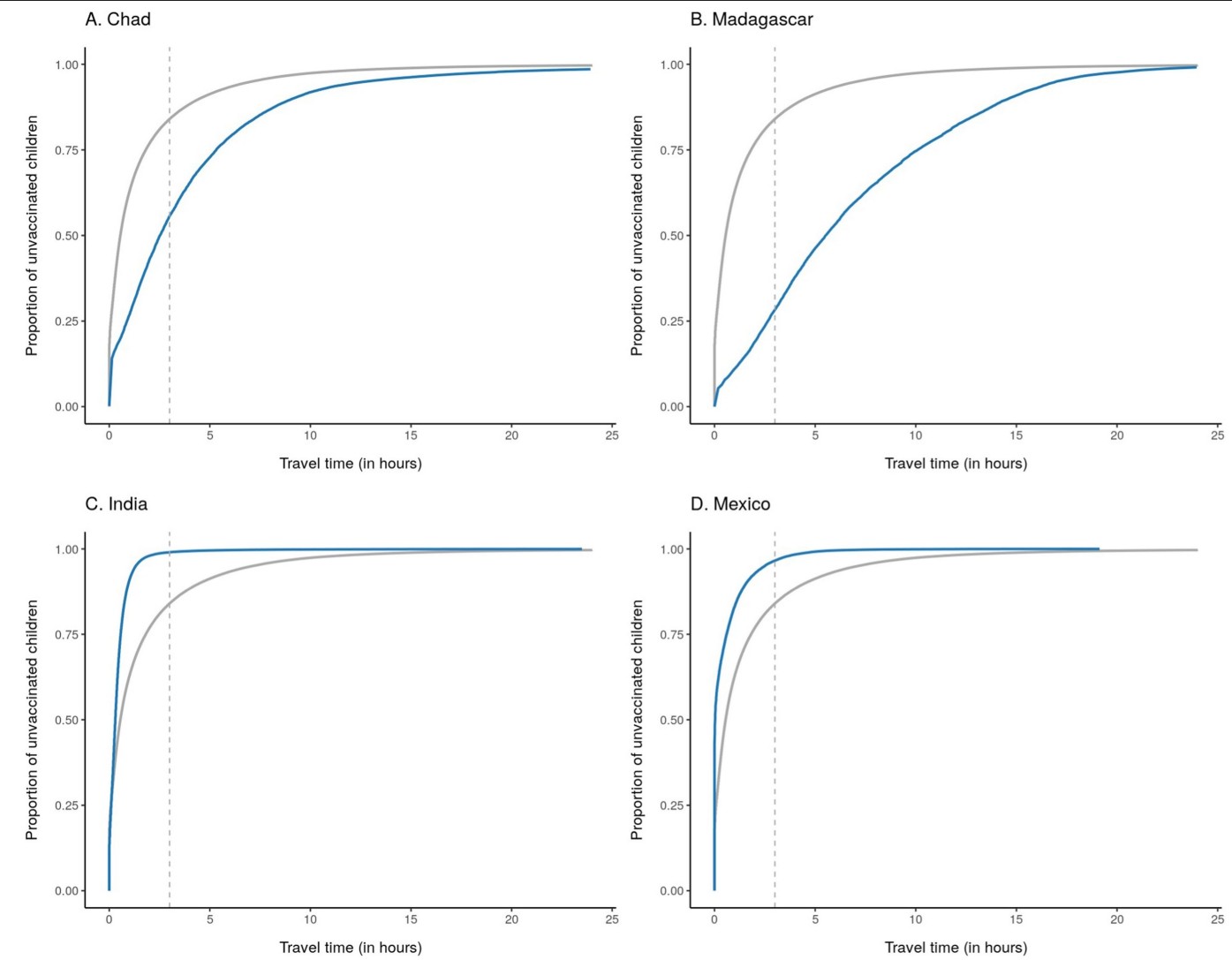

**Extended Data Fig. 10 | Country examples of concentration curves.** Concentration curves of the cumulative proportion of unvaccinated children as a function of travel time (in hours) in Chad (**a**), Madagascar (**b**), India (**c**) and Mexico (**d**). Curves for both the indicated countries (blue) and all LMICs (grey) are shown.

# nature research

# Reporting Summary

Nature Research wishes to improve the reproducibility of the work that we publish. This form provides structure for consistency and transparency in reporting. For further information on Nature Research policies, see Authors & Referees and the Editorial Policy Checklist.

## Statistics

For all statistical analyses, confirm that the following items are present in the figure legend, table legend, main text, or Methods section.

| n/a | Confirmed | |
|---|---|---|
| ☐ | ☒ | The exact sample size (*n*) for each experimental group/condition, given as a discrete number and unit of measurement |
| ☐ | ☒ | A statement on whether measurements were taken from distinct samples or whether the same sample was measured repeatedly |
| ☐ | ☒ | The statistical test(s) used AND whether they are one- or two-sided<br>*Only common tests should be described solely by name; describe more complex techniques in the Methods section.* |
| ☐ | ☒ | A description of all covariates tested |
| ☐ | ☒ | A description of any assumptions or corrections, such as tests of normality and adjustment for multiple comparisons |
| ☐ | ☒ | A full description of the statistical parameters including central tendency (e.g. means) or other basic estimates (e.g. regression coefficient) AND variation (e.g. standard deviation) or associated estimates of uncertainty (e.g. confidence intervals) |
| ☐ | ☒ | For null hypothesis testing, the test statistic (e.g. *F*, *t*, *r*) with confidence intervals, effect sizes, degrees of freedom and *P* value noted<br>*Give P values as exact values whenever suitable.* |
| ☐ | ☒ | For Bayesian analysis, information on the choice of priors and Markov chain Monte Carlo settings |
| ☒ | ☐ | For hierarchical and complex designs, identification of the appropriate level for tests and full reporting of outcomes |
| ☒ | ☐ | Estimates of effect sizes (e.g. Cohen's *d*, Pearson's *r*), indicating how they were calculated |

*Our web collection on statistics for biologists contains articles on many of the points above.*

## Software and code

Policy information about availability of computer code

| Data collection | No primary data collection was carried out for this analysis. |
|---|---|
| Data analysis | This analysis was carried out using R version 3.6.1 and using R-INLA v.20.01.29.9000. Maps were produced using ArcGIS Desktop 10.6 and Python 2.7. All code used for these analyses will be made publicly available upon publication. |

For manuscripts utilizing custom algorithms or software that are central to the research but not yet described in published literature, software must be made available to editors/reviewers. We strongly encourage code deposition in a community repository (e.g. GitHub). See the Nature Research guidelines for submitting code & software for further information.

## Data

Policy information about availability of data

All manuscripts must include a data availability statement. This statement should provide the following information, where applicable:
- Accession codes, unique identifiers, or web links for publicly available datasets
- A list of figures that have associated raw data
- A description of any restrictions on data availability

The findings of this study are supported by data available in public online repositories and data publicly available upon request of the data provider. A detailed table of data sources and availability can be found in Supplementary Table 4 and http://ghdx.healthdata.org/lbd-publication-data-input-sources. Administrative boundaries were modified from the Database for Global Administrative Areas (GADM) dataset. Populations were retrieved from WorldPop and gridded estimates of travel time to nearest city or settlement were available online from work by Weiss, et al 2018. This study complies with the Guidelines for Accurate and Transparent Health Estimates Reporting (GATHER) recommendations. All maps and figures presented in this study are generated by the authors; no permissions are required for publication.

# Field-specific reporting

Please select the one below that is the best fit for your research. If you are not sure, read the appropriate sections before making your selection.

☒ Life sciences  ☐ Behavioural & social sciences  ☐ Ecological, evolutionary & environmental sciences

For a reference copy of the document with all sections, see nature.com/documents/nr-reporting-summary-flat.pdf

# Life sciences study design

All studies must disclose on these points even when the disclosure is negative.

| | |
|---|---|
| Sample size | This observational study incorporated all available survey data sources that met the inclusion criteria as described in detail in the manuscript and supplementary information. The combined dataset from 354 household based surveys contained information on vaccination status from 1.70 million individual children. |
| Data exclusions | Surveys were excluded due to unrealistic national or geographic trends compared to other surveys in nearby country-years, inability to match the microdata to geographic locations, or non-standard methodology. These criteria were pre-established prior to reviewing the data. A full list of excluded surveys is included in Supplementary Table 5. |
| Replication | All code and data are available publicly for reproducibility. |
| Randomization | As this work is an observational mapping study, there were no experimental groups. |
| Blinding | As this work is an observational mapping study, there was no need for blinding. |

# Reporting for specific materials, systems and methods

We require information from authors about some types of materials, experimental systems and methods used in many studies. Here, indicate whether each material, system or method listed is relevant to your study. If you are not sure if a list item applies to your research, read the appropriate section before selecting a response.

## Materials & experimental systems

| n/a | Involved in the study |
|---|---|
| ☒ | Antibodies |
| ☒ | Eukaryotic cell lines |
| ☒ | Palaeontology |
| ☒ | Animals and other organisms |
| ☒ | Human research participants |
| ☒ | Clinical data |

## Methods

| n/a | Involved in the study |
|---|---|
| ☒ | ChIP-seq |
| ☒ | Flow cytometry |
| ☒ | MRI-based neuroimaging |

