## [Peer Review File · Nature]

Manuscript Title: Mapping routine measles vaccination in low- and middle-income countries

Editorial Notes: N/A

Reviewer Comments & Author Rebuttals**Reviewer Reports on the Initial Version:****REFEREE #1 (REMARKS TO THE AUTHOR):**

Summary The paper is absolutely interesting and well-written. The novelty of the work lies in the humongous challenge undertaken by the authors to assemble data from a wide range of sources, including household surveys, to model and map MCV1 coverage in 100 low and middle income countries (LMICs). The modelling approach used by the authors, termed stacked generalization, involves the use of geospatial covariate data to fit child models, predictions from which are included in a geostatistical modelling framework to produce ‘combined’ estimates of coverage. These estimates are then further calibrated to match the global burden of disease estimates.

As I have highlighted below, a major shortcoming of the work which needs to be addressed properly before the paper is published is the birth cohort approach used by the authors to process and reassign MCV1 coverage from different age groups. Because this approach does not distinguish between routine doses and campaign doses, I would recommend that the title of the manuscript is changed to “Mapping measles vaccination coverage in low- and middle-income countries”. Additionally, this approach does not account for time of vaccination, rather it considers different birth cohorts of children to determine whether they have been adequately vaccinated or not. The predicted maps thus relate to the years in which the cohorts were born and not necessarily the years/times they were vaccinated. This should be clarified throughout the manuscript. It is absolutely wrong to assume that catch-up vaccination is negligible, as experiences from many LMICs suggest that vaccination campaigns play a vital role in boosting coverage levels. What the authors have investigated in the current work is “MCV1 coverage” and not “routine MCV1 coverage”.

Other issues that I found in the manuscript are listed below.

Major comments

1. The authors mentioned that areal data were converted to point data before modelling (this is shown in supplementary Fig 1). I understand this was done using survey data as well as administrative estimates from JRF. I was wondering if there were any restrictions as to the size of the areas during this process. This is a very crude approach to handling the spatial misalignment between the areas and the point level at which data are needed for modelling. For large admin 1 areas, this crude approximation is likely to introduce large biases in the data that are modelled.

2. I have a major concern regarding the approach used by the authors to process data on MCV1 coverage. The authors mentioned (see lines 433 – 439) that they first assigned children from each survey to different age groups, namely 12-23 months, 24-35 months, 36- 47 months and 48-59 months based on their age at the time of interview. Then they proceeded to include each age group in the year that they were aged 0-12 months. First, this method is bound to introduce a lot of biases in the data as it does not accurately reflect the years that the children were vaccinated, so the authors cannot claim to have effectively investigated trends in coverage. Using the 2013 Nigeria DHS survey as an example, children aged 36-47 months in that survey would have been aged 0-11 months in 2010 and 12-23 months in 2011. There is a strong possibility that those kids were not vaccinated during 2010-2011, hence the authors will be misclassifying the year that they were vaccinated using their birth cohort approach described above. Moreover, Nigeria and many other countries conduct regular vaccination campaigns and routine vaccination is never restricted to children aged 0-11 (or 12) months. Since surveys do not usually distinguish between a routine dose and a campaign dose, it would be erroneous for the authors to attribute all of the doses to routine vaccination only. To reduce this bias, I strongly recommend that the authors (i) consider only children aged 12-23 months during the survey year - which is a standard age group for measuring routine coverage - and cancel reassigning older kids to their birth years, or (ii) include information on SIAs to determine which birth cohorts have not been exposed to an SIA (or many SIAs) and consider including only those cohorts in the year they were aged 0-12 months – although this will not accurately account for time of vaccination. Secondly, I'm wondering what this data processing method means in terms of the age group that the predicted maps relate to. 0-12 months or 12-23 months? Thirdly, there is also the issue of recall bias, particularly for older children, as the caregivers are more likely to struggle with remembering what vaccines those children would have received. It is mentioned in lines 397 – 400 that the populations used to assess coverage were either 12-23 months or 24-35 months, depending on the country's vaccination schedule. Then at lines 433 – 439, the authors mention that data from each birth cohort were included in the year in which the birth cohort was aged 0-12 months. This would be wrong in countries where the dose is recommended in the 2nd year of life as it does not reflect the time of vaccination.

3. The stacked generalization step that involves using predictions of coverage from different child models in a geostatistical model to predict the same coverage is not a principled approach to combining predictions from a statistical standpoint. First, the authors mention at lines 440 – 448 that the selected covariates used in the child models were not collinear. This makes one wonder why the same rule was not applied when fitting the geostatistical model, since predictions from the child models will be highly correlated. Placing the restriction $\beta c = 1$ on the regression parameters will not make this problem go away. Secondly, this approach does not offer a way of propagating the uncertainties associated with the predictions from the child models to the final models. In a Bayesian context, what the authors have attempted to do is termed Bayesian model averaging, but it has been implemented in the most unusual way. Thirdly, the estimated regression coefficients in the geostatistical model are meaningless – these basically relate coverage to coverage. Since parameter estimates from the child models are not presented, it is difficult to understand how useful the covariates were for predicting vaccination coverage.

4. Both the child models and the geostatistical model were fitted using point/cluster-level data. It's difficult to understand why the authors chose to validate the geostatistical model at the first administrative level. Does this mean that the point level estimates were not reliable enough to be

used for validation? Another important point here is whether the “observed” admin level one data used in calculating the validation metrics were weighted or not. Since some admin data were converted to point data using the centroids of the admin areas, it is unclear how any sampling weights were applied to such data when generating weighted estimates at the admin level. It will not be ideal to compare the modelled estimates to unweighted admin level one data.

Minor comments

1 The geostatistical model is poorly described. It is not clear whether the variables C_i , N_i , and p_i are vectors or scalars. I assume that these are vectors representing time series data for the i th location. This needs to be clarified. A space-time structure is given for the GP model but no temporal index is given in the binomial likelihood and the logit function, etc.

2 How was the restriction $\sum \beta_c = 1$ implemented during model fitting? The prior specified for β_c does not enforce this.

3 The increases in coverage mentioned in lines 86 – 90, were these significant increases? Knowing whether the increases were significant or not will be of interest to many readers.

4 The 1 x1 km resolution data used to mask out some areas as mentioned in Figure 2 does not match the spatial resolution of the modelled 5 x 5 km surfaces.

5 Lines 36-38 – I believe that the 95% effectiveness mentioned relates to valid doses. This needs to be highlighted.

REFeree #2 (REMARKS TO THE AUTHOR):

I have focused upon the spatial data analysis surrounding the Bayesian hierarchical model. These models are widely used in a number of contexts and there is at least one whole book devoted to the subject; see, e.g., Banerjee, Carlin and Gelfand (2014). Hence, the model development is not exactly original, but it is very appropriate for the analysis and the benefits of Bayesian hierarchical modeling have been, and continue to be, well documented in the statistics literature. In this regard, the adoption of these models is commendable.

There are, nevertheless, a few key issues that would benefit from a discussion.

1. The authors claim that they use a great-circle distance for their spatial distance. This may work fine in many specific instances, but there is a technical issue here that the authors should take a closer look at. The most commonly used spatial covariance functions, including the Matern, are theoretically valid (will yield positive definite covariance matrices) for Euclidean (planar) distances. Great circle distances require specialized covariance functions based upon Legendre polynomials. If the spatial domain of interest is small enough that the curvature of the earth is not prominent, then the great circle distance is well approximated by the Euclidean distance and the use of the former in regular covariance functions is practicable. Otherwise, this could lead to numerical issues. See Banerjee (Biometrics, 2005, <https://doi.org/10.1111/j.1541-0420.2005.00320.x>) for an elaborate discussion. If the authors have not encountered any problems using the great circle distance, then they may keep the current analysis but should recognize the pitfalls in the above reference.

2. The tensor (Kronecker) product structure for spatial-temporal covariance implies the notion of

"separability" in space and time. This means that the temporal covariance structure remains the same for each location. Likewise, the spatial covariance structure over the region is the same across time. This has undesirable implications for modeling; see Stein (JASA, 2005; <https://doi.org/10.1198/016214504000000854>) and also Gneiting (JASA, 2002; <https://doi.org/10.1198/016214502760047113>). This issue may not afflict the substantive inference being pursued here, but, again, should be acknowledged and the papers cited.

3. On the same vein, why use AR(1) structure for time? Why not use spatial-temporal covariance functions as in the above references? Or even in a separable model, with time modeled as a Gaussian process using a covariance function?

4. R-INLA is a fine product for implementing the Bayesian hierarchical models. It avoids MCMC and relies upon Laplace approximation and quadrature for inference. However, MCMC, while slower, does have advantages as it samples from the posterior and the identifiability (or lack thereof) of the parameters is sometimes better reflected. Have the authors considered STAN (<https://mc-stan.org/>) as an alternative to confirm the analysis?

REFeree #3 (REMARKS TO THE AUTHOR):

Dear Editor,

Re: Mapping routine measles vaccination in low- and middle-income countries

This manuscript generates estimates of first-dose measles-containing vaccine (MCV1) in low and middle-income countries from 2010-2018.

The research topic is an important one and the data are expansive. However, I have difficulty in following the statistical models they authors are using.

1) First of all, the model input and model outcome are not clearly stated. In the main text, I could not find details of the covariates that are used to fit the models. I appreciate that that may a long list but I think it is important to at least give an overview of these covariates. And only till very end of the main text, I realize that MCV1 is treated as a model outcome. It is also not clear to me why MCV1 is a model outcome rather than a quantity that is observed. The authors should make these very clear at the beginning of the paper. Also, because of the expansiveness of the data and study, the authors need to make their best efforts to present things in a clear and logical way so that readers would not lose track in reading.

2) I am not following most of the description of the statistical models and methods under the section "Geostatistical model". For example in line 456, what are the "optimized covariates" and how did the author choose them? I find most part of the description is too brief and abrupt. I would suggest the author to rewrite the whole section, and, in particular, to give a high-level overview (ideally graphical) of the statistical procedures used.

Author Rebuttals to Initial Comments: (please note that the author has quoted the referees in black and responded in blue)

Referee 1

Summary

The paper is absolutely interesting and well-written. The novelty of the work lies in the humongous challenge undertaken by the authors to assemble data from a wide range of sources, including household surveys, to model and map MCV1 coverage in 100 low and middle income countries (LMICs). The modelling approach used by the authors, termed stacked generalization, involves the use of geospatial covariate data to fit child models, predictions from which are included in a geostatistical modelling framework to produce 'combined' estimates of coverage. These estimates are then further calibrated to match the global burden of disease estimates.

As I have highlighted below, a major shortcoming of the work which needs to be addressed properly before the paper is published is the birth cohort approach used by the authors to process and reassign MCV1 coverage from different age groups. Because this approach does not distinguish between routine doses and campaign doses, I would recommend that the title of the manuscript is changed to "Mapping measles vaccination coverage in low- and middle-income countries". Additionally, this approach does not account for time of vaccination, rather it considers different birth cohorts of children to determine whether they have been adequately vaccinated or not. The predicted maps thus relate to the years in which the cohorts were born and not necessarily the years/times they were vaccinated. This should be clarified throughout the manuscript. It is absolutely wrong to assume that catch-up vaccination is negligible, as experiences from many LMICs suggest that vaccination campaigns play a vital role in boosting coverage levels. What the authors have investigated in the current work is "MCV1 coverage" and not "routine MCV1 coverage". Other issues that I found in the manuscript are listed below.

We thank the referee for their interest in the manuscript and thoughtful feedback. We agree with the referee that the use of data from children of various ages to estimate MCV1 coverage is particularly challenging given variation in the target age range for MCV1 by country and the broad use of frequent supplemental immunization activities (SIAs) in the countries included in this analysis. In the specific responses to the referee's comments below, we aim to clarify these limitations and the implications of our methodology. We also describe sensitivity analyses conducted and modifications to our approach that we have made to address the concerns raised by the referee.

We also appreciate the referee's comment regarding the use of the term "routine coverage" to describe the results of this analysis. We agree that the ability to parse differences between doses given *via* routine immunization programs versus SIA is complex and limited due to the nature of current standards of practice in vaccination record keeping, survey questionnaires and data collection. We also agree that, in many of the countries in this analysis, vaccination campaigns play a vital role in boosting MCV coverage levels, particularly when routine coverage is low. However, we find great value in trying to differentiate dose administered *via* routine and supplemental immunization activities, in order to characterize their successes and shortcomings, and so that policy makers can consider which combination of strategies best meets the needs of each country. For instance, SIAs may reach unimmunized children across

wealth quintiles more equitably than RI¹, but SIAs may be more vulnerable to disruption due to funding challenges, political shifts, or disasters². As routine vaccination serves as the cornerstone of immunization strategies for all countries³, having reliable estimates of routine immunization coverage is imperative for understanding vaccination program vulnerabilities and where supplemental services are most needed to reach unvaccinated children.

In the era of COVID-19, a thorough understanding of baseline routine immunization gaps is particularly important. With the cancellation of almost all planned supplemental immunization programs to date from March 2020 to date⁴, routine immunization services are still recommended globally⁵ and serve as the primary mechanism through which measles vaccination will be delivered in most settings. Understanding gaps in routine immunization services will be crucial to preventing measles outbreaks and excess child morbidity and mortality during this temporary halt on SIAs.

We therefore attempt, within the limits of the available data, to estimate coverage delivered through routine immunization services and remove SIA-delivered doses from the available data. As the referee notes, survey data is highly likely to contain doses of MCV delivered by SIAs, either through miscoding on vaccination cards or through parental recall. In our data processing, however, we remove SIA doses recorded in survey data wherever possible, leveraging codes present in the individual-level survey microdata. In response to the referee's comments, and appreciating that this step in our data processing may have been unclear previously, we have provided a description of this process below and have incorporated additional details on this in Supplemental Information section 1.2.1 (lines 98 – 148) and Supplementary Information section 1.3.4 (lines 198 – 215). We also note that this approach has been used to produce estimates of routine MCV coverage by other researchers in the past, such as in recent studies differentiating between routine and SIA coverage^{1,6,7}.

Despite these efforts, we appreciate that misclassification of some SIA-administered doses is likely to be inevitable in any attempt to estimate routine immunization coverage. We recognize that this is an inherent limitation to our analysis. We have therefore added additional discussion in our Online Methods section (lines 426 – 429) as well as limitations of the manuscript (lines 625 – 629) to be more transparent regarding this shortcoming of the available data and our ability to completely parse the source of MCV doses included in this analysis.

Major comments

1. The authors mentioned that areal data were converted to point data before modelling (this is shown in supplementary Fig 1). I understand this was done using survey data as well as administrative estimates from JRF. I was wondering if there were any restrictions as to the size of the areas during this process. This is a very crude approach to handling the spatial

misalignment between the areas and the point level at which data are needed for modelling. For large admin 1 areas, this crude approximation is likely to introduce large biases in the data that are modelled.

We agree with the referee that including data that can only be geographically resolved to administrative boundaries in our geospatial modelling framework is challenging. To clarify, however, we are not resampling national data, or any administrative data sources, from the JRF or otherwise. In the geostatistical model described in this manuscript, we include only data that can be resolved subnationally (lines 445-449) as follows:

Surveys were included if they contained MCV1 coverage information and subnational geolocation, and excluded if they contained areal data and were missing key survey design variables (strata, primary sampling units, and design weights), did not include children aged 12–59 months, contained no subnational individual-level geographic information, or if coverage estimates were implausible (Supplementary Tables 4 and 5).

We also clarified further in Supplementary Information section 1.5.1 (lines 357 -364) as follows:

Using the results of the geostatistical modelling process above, we calibrated our set of posterior draws to estimates of MCV1 coverage from GBD 2019. The geostatistical model described above uses only subnationally-resolved survey data, while coverage estimates produced by GBD 2019 additionally leverage surveys for which only national-level data are available as well as bias-adjusted national-level administrative data. This calibration step allows our geospatial estimates to reflect national-level trends in coverage that are informed by national-level surveys and administrative data, while preserving the relative spatial patterns estimated by the geospatial model.

We incorporate national-level, bias-adjusted administrative data on vaccine coverage from the Global Burden of Disease study as a covariate in the geospatial modelling process (Supplementary Information lines 376-383), but neither national-level or subnational administrative data are resampled or used as input data in the geostatistical model. We have investigated the use of country-reported subnational administrative data in our geospatial models, but have not yet identified a method to appropriately account for the substantial level of noise and bias in this data.

As an additional clarification, our methods do not assign data to the centroid of each first administrative unit polygon (Major Comment #4), but instead distribute “candidate points” based on underlying population weights using a k-means clustering algorithm. We recognize that our description of these methods was unclear in the original manuscript. In order to increase clarity, we have added an additional Supplementary Fig. 5 as well as expanded and edited the language regarding this method in the manuscript (lines 451-456) as follows:

If GPS information was not collected or was not available, we calculated mean coverage at the most granular geographic area possible while accounting for sampling weights and survey design. These aggregated coverage estimates were then included in the geospatial modelling process using a previously described method that leverages population weights and a k-means clustering algorithm to propose a set of GPS coordinates as a proxy for locations where survey data collection could have occurred. These coordinates were then used to represent the areal data in the geospatial model.

Also, we increased the clarity of our description in Supplementary Section 1.3.5 (lines 229 – 238) to read as follows:

For areally located data that has been matched to an administrative unit but for which no GPS coordinates were available, we used a previously described method to generate a set of candidate cluster locations where survey sampling could plausibly have occurred, using population weights and k-means clustering (Supplementary Fig. 5). In brief, 10,000 points were randomly sampled from the area within the administrative boundary that was matched to the areal data. Points were sampled proportionally to the total population within that space and time as estimated by the WorldPop population raster. A set of integration points (1 per 1,000 5 × 5-km pixels) was generated via k-means clustering, representing candidate locations of survey sampling under the assumption that each survey sampled locations proportional to population.

As noted above, we do not resample national-level data using this process but appreciate the reviewer's comments regarding the limitations of the use of administrative data for large first-administrative units. Because most household-based surveys are powered for analysis at the first administrative level, however, the geo-resolution of children is often only collected and available at the first administrative level. Although we try to resolve geo-location to the most granular location possible (second, third administrative levels or GPS coordinates), often through direct engagement with survey providers and country-level collaborators, this information may not be available or may not have been collected for many surveys. In this work, 125 surveys (about 1/3 of total dataset) are not available at geographic resolutions more granular than at the first administrative unit level. Excluding this data would substantially limit the ability of a spatiotemporal analysis such as this to reflect changes in coverage over time.

Despite the limited geospatial resolution of these data sources, we believe that a model-based geostatistical (MBG) approach has key advantages for analysing large data sets on vaccine coverage, as they have shown to be for many other diseases and interventions⁸⁻¹⁰. These methods produce estimates that are informed by data resolved to the first administrative unit but can also reflect lower-level geographic variation, from spatiotemporal relationships between coverage and covariates, and from coverage patterns present in higher-resolution data across space and time. MBG methods allow data to be included in the model at the most granular geographic resolution possible, and – by estimating coverage at the local level – allow the flexibility to produce consistent estimates across time despite evolving geo-political borders.

To use this MBG approach, however, we currently need all observation-level data to be located to a specific point in space and time, hence the conversion of areally-located data to specific GPS coordinates. The method we present in this work, the resampling method, is one possible solution to this alignment dilemma. We agree with the referee that this approach has important limitations, and we have elaborated on these in the main text and Supplemental Information. These limitations include:

- Under-propagation of uncertainty from the data, as it is not actually known if the candidate points were sampled from the estimated locations
- The potential to obscure covariate relationships, as the covariate values from the locations of the candidate GPS locations produced by this resampling method may differ from the covariate values from the true survey sampling locations
- An overly smooth spatial pattern in the final estimates

Despite these limitations, a recent study¹¹ found that eliminating areal data all together, compared to the resampling method, had worse 95% coverage of the underlying

probability field (i.e. predictive validity), suggesting that the incorporation of areal data using the resampling method is preferable to ignoring the data altogether. We agree with the referee that this is an important topic in our methods, which warrants more emphasis in order to help readers evaluate our results. We have included an additional extension of this method in our limitations section of the manuscript (lines 632 - 636).

Methods to better include areal data in geospatial modelling are currently under development. An ideal version of this method might fit models using all available GPS-coordinate level “point” data. While this model is fitting, coverage would be calculated at each first and second administrative unit and compared to the input data sources that can only be resolved to these spatial levels. The result of this method would be a set of geospatial estimates that would reflect data at each aggregated spatial level (including first and second administrative units). While this type of modelling framework is being explored¹², these models are not currently computationally feasible to use at the geographic scale required by our regional modelling approach.

2. I have a major concern regarding the approach used by the authors to process data on MCV1 coverage. The authors mentioned (see lines 433 – 439) that they first assigned children from each survey to different age groups, namely 12-23 months, 24-35 months, 36-47 months and 48-59 months based on their age at the time of interview. Then they proceeded to include each age group in the year that they were aged 0-12 months.

First, this method is bound to introduce a lot of biases in the data as it does not accurately reflect the years that the children were vaccinated, so the authors cannot claim to have effectively investigated trends in coverage. Using the 2013 Nigeria DHS survey as an example, children aged 36-47 months in that survey would have been aged 0-11 months in 2010 and 12-23 months in 2011. There is a strong possibility that those kids were not vaccinated during 2010-2011, hence the authors will be misclassifying the year that they were vaccinated using their birth cohort approach described above. Moreover, Nigeria and many other countries conduct regular vaccination campaigns and routine vaccination is never restricted to children aged 0-11 (or 12) months. Since surveys do not usually distinguish between a routine dose and a campaign dose, it would be erroneous for the authors to attribute all of the doses to routine vaccination only. To reduce this bias, I strongly recommend that the authors (i) consider only children aged 12-23 months during the survey year - which is a standard age group for measuring routine coverage - and cancel reassigning older kids to their birth years, or (ii) include information on SIAs to determine which birth cohorts have not been exposed to an SIA (or many SIAs) and consider including only those cohorts in the year they were aged 0-12 months – although this will not accurately account for time of vaccination.

We thank the referee for this comment. We confirm that we did assign children to annual age cohorts based on the age at time of interview and included children in model in the year in which they were aged 0-12 months. We appreciate this opportunity for discussion and to improve our models by more closely interrogating our assumptions regarding the inclusion of data from these additional cohorts of children and the subsequent implications for our model results.

First, we would like to place our approach in the broader context of immunisation coverage estimation. Our methods for including cohorts beyond the target age schedule is a practice similarly adapted by the WUENIC group among several locations^{13,14} in historical and recent updates of global immunization coverage. The shift of coverage estimates to the birth year of

a given cohort is also a method used by the WUENIC group. Our models align with GBD coverage estimates, which incorporate bias-adjusted national-level country-reported data following the approach of WUENIC. We agree that a more precise method to align national-level country-reported data and survey data would be ideal, if sufficient data were available. The JRF, however, does not routinely collect nor report information on the precise age cohort that is used by each country in each year to report administrative or official country-reported data. Therefore, we adopt the methodology used by the WUENIC group in both our GBD and geospatial estimates to ensure comparability between these estimates and those national-level estimates commonly used by policymakers, and consistency with the GBD estimates to which our geospatial estimates are calibrated. We have made this rationale and the associated limitations clearer in the Supplementary Information section 1.2.1 (lines 113-121) as follows:

We assign coverage from each birth cohort of children surveyed to the year in which they were aged 0–12 months for modelling, in alignment with the methods used by WUENIC and estimates from the Global Burden of Diseases, Injuries, and Risk Factors study (GBD). The WUENIC and GBD national-level coverage estimates both additionally incorporate administrative and/or official country-reported data on MCV1 coverage, which does not routinely collect nor report information on the precise birth cohort that accompanies the country-reported values of coverage. In order to maintain comparability with these national-level estimates, and because our estimates are calibrated to those produced by GBD, we adopt the same methodology for birth cohort assignment.

We also agree with the referee that the inclusion of older cohorts of children may raise the question of bias. Older children may be more likely to be vaccinated due to additional opportunities for vaccination through catch-up routine immunization services, inclusion in additional supplemental immunization activities, or through a second opportunity for first-dose immunization during the second year of life in countries that have introduced MCV2. Alternatively, in select cases, older cohorts may have lower coverage than younger cohorts in a given survey, due to – for instance – rapidly increasing secular trends in immunization coverage with limited catch-up vaccination. To further investigate these possible sources of bias, we conducted a series of sensitivity analyses described below and in the manuscript and Supplementary Information.

First, we conducted an analysis of cohorts of children for which we have multiple observations of vaccine coverage from surveys conducted at different points in time. We chose this approach because simply looking at coverage by cohort within a given survey presents an age-period-cohort problem, where the effects of changing coverage by age and changing coverage within a country over time are extremely difficult to disentangle. By analysing cohorts with multiple survey observations, we could compare survey estimates of coverage for the cohort obtained at the time in which they were the target age and in at least one other year, for instance when the children were one year older than the target age ($t+1$), two years older than the target age ($t+2$), or three years older than the target age ($t+3$). We computed the log of the coverage ratios between observations that took place when the cohort was of the target age and when the cohort was in the older age category, and regressed this value against indicator variables for each time lag ($t+1$, $t+2$, and $t+3$). If increasing age at the time of survey for a given cohort is associated with increases in coverage (bias), we would expect that a significant effect size for the relevant indicator variables.

The results of this analysis demonstrated a significant effect for the indicator variable for the largest ($t+3$) time lag between observations for a given cohort (Supplementary Information section 2.2). Therefore, we decided to remove these oldest cohorts, i.e. those that were 3 years older than the target age group, from our data set. Removing these cohorts took our overall sample size from 1,879,146 children to 1,568,653 children. In order to further explain the rationale for this approach and the inherent limitations, we have added this sensitivity analysis and an additional discussion of the limitations to Supplementary Information section 2.2 (lines 431-469).

We would note that the determination of precisely which children were eligible for a given SIA is not entirely straightforward in many cases. The SIA database maintained by WHO is an excellent public good, but details are lacking for a number of SIAs. In particular, the entries for SIAs conducted subnationally do not generally contain details on the subnational timing, location, or coverage of the campaign. For a full accounting in a geospatial modelling framework, information on the subnational timing and location of SIAs would be necessary to include or exclude children based on SIA eligibility or attempt to adjust for SIA exposure in a model of coverage. In collaboration with WHO, we have been working on a long-term project to systematically extract and geolocate subnational data from SIA technical reports in order to better inform their future inclusion in geospatial models such as these. At the present time, however, such subnational SIA data is not yet available, limiting our ability to fully adjust for SIA eligibility in older age cohorts.

We also considered the possibility that the addition of MCV2 to routine schedules may provide some children who did not receive MCV1 on the routine schedule with a second-year-of-life opportunity for a first MCV1 vaccination. Among the 355 surveys in this analysis, however, only 28 were conducted after a country had included MCV2 in the routine immunization schedule, suggesting that the overall influence of MCV2 in our modelled estimates is likely to be low.

Last, we do recognize that an ideal approach to this problem might involve the construction of an age-specific model in order to better account for changes in coverage by age and over time. Such a model would be methodologically challenging for several reasons. First, such a model would need to be capable of incorporating age-specific coverage across non-standard and variable age bins from the various survey types included in this analysis. Second, these models would be subject to the same fundamental age-period-cohort challenges as any analysis of coverage both by age and over time. Last, because of the additional interaction with age, these models are substantially more computationally challenging to fit, particularly at the scale of the regions used in this study. Age-specific coverage models designed to account for these challenges at large spatial scales are under development but not yet feasible for this analysis.

Based on the results of the sensitivity analyses described above, we therefore now exclude the $t+3$ age cohort from our input data for the analysis presented in the main manuscript but preserve the younger cohorts. This approach allows us to include as much data on coverage as possible, while excluding the oldest cohorts at highest risk of bias.

Last, to better convey the impact of the inclusion of multiple age cohorts on our estimates and conclusions, we ran three additional versions of the models: a model including all cohorts from each survey (the approach used for the original submission of this manuscript), a model with the target age cohort along with $t+1$ and $t+2$ cohorts (the

approach used for the current main text of the manuscript), a model with only the target age cohort and the t+1 cohort, and a model with the target age cohort alone. The results and major findings of this work did not sustainably change among the four cohort versions. We have included this analysis in the Supplementary Information as a sensitivity analysis (Supplementary Information section 2.2). See Supplementary Table 15 with corresponding results for each version as well as second administrative level estimates from each model in years 2000, 2010, and 2019. For a snapshot of these results, see below:

	Full cohort model	Modified cohort version A (target age, t+1 and t+2 cohorts) model	Modified cohort version B (target age and t+1) model	Target age cohort only model
Proportion of districts with increased coverage from 2000 to 2019	57.0% (95% UI: 50.0 – 64.4%)	56.8% (95% UI: 49.9–64.0%)	55.7% (95% UI: 48.7 – 63.3%)	56.7% (95% UI: 50.2 – 63.5%)
Proportion of districts with increased coverage from 2000 to 2010	69.0% (95% UI: 64.4 – 73.7%)	69.1% (95% UI: 64.3–73.6%)	67.4% (95% UI: 61.8 – 73.2%)	69.1% (95% UI: 64.4 – 73.8%)
Proportion of districts with increased coverage from 2010 to 2019	40.6% (95% UI: 34.6 – 47.3%)	39.8% (95% UI: 34.1–46.5%)	39.6% (95% UI: 33.8 – 47.7%)	39.9% (95% UI: 34.4 – 46.9%)
Proportion of districts meeting GVAP target in 2000	39.1%	38.1%	36.9%	39.3%
Proportion of districts meeting GVAP target in 2019	32.4%	32.2%	32.2%	33.0%

We have clarified the changes to our methodology as well as choice in age cohorts included in the input data set in the manuscript (lines 465 – 468) as follows:

Data corresponding to each birth cohort were included in the modelling process in the year in which that birth cohort was aged 0–12 months old. We included children in the target age cohort and cohorts 1 and 2 years older than the target cohort; children were excluded if they were 3 years older than the target cohort.

We have also made modifications to reflect these changes in Supplementary Information section 1.2.1 (lines 108-112) that now read as:

We have included children among the target age cohort and the cohorts 1 and 2 years older than the target cohort at the time of survey. We excluded children that were three years older than the target cohort at the time of survey. We have tested this assumption in a sensitivity analysis that can be found in Supplementary Information section 2.2.

Secondly, I'm wondering what this data processing method means in terms of the age group that the predicted maps relate to. 0-12 months or 12-23 months?

We thank the referee for the opportunity to clarify. The predicted maps relate to coverage in the year following the target age of vaccination, which varies by country, but are represented in the year the child is born. This aligns our estimates with those produced by the GBD as well as the WUENIC group. We have modified the Supplementary Information section 1.2.1 (lines 98-102) to make this clearer as follows:

Routine coverage of the first dose of any measles-containing vaccine (MCV1), including measles-rubella and measles-mumps-rubella combination vaccines, was defined as the proportion of children born in a given year who received at least one dose of any measles-containing vaccine through routine immunisation services after reaching the target age for immunisation.

Thirdly, there is also the issue of recall bias, particularly for older children, as the caregivers are more likely to struggle with remembering what vaccines those children would have received.

We agree with the referee that the possibility of recall bias is substantial, especially as children age and caregivers are required to remember information for longer periods of time. However, previous research has suggested that there is no consistent direction of the effect of recall bias on coverage found in survey data¹⁶. This is a major limitation in implementing a mechanism to account for this bias across numerous surveys. One approach to this problem might be to develop a modelling framework such that doses from recall are included in a model with less certainty than doses recorded on a vaccination card, with the level uncertainty reflecting the time lag between immunization and survey data collection. This is an area of important future methodological development but currently beyond the capability of our model. To further clarify and outline this limitation to readers, we have added text to the manuscript (lines 625 -629) that reads:

We aim to estimate routine coverage and have excluded doses delivered via SIAs from the analysed survey data wherever possible (Supplementary Information section 1.3.5), but misclassification of SIA doses is still likely, particularly in cases of parental recall -- especially among older children -- and in cases where survey methodology does not distinguish clearly between SIA and routine doses.

It is mentioned in lines 397 – 400 that the populations used to assess coverage were either 12-23 months or 24-35 months, depending on the country's vaccination schedule. Then at lines 433 – 439, the authors mention that data from each birth cohort were included in the year in which the birth cohort was aged 0-12 months. This would be wrong in countries where the dose is recommended in the 2nd year of life as it does not reflect the time of vaccination.

As mentioned above, we have computed coverage among the country-specific target population and have included cohorts in their birth year to be in line with the approach of the WUENIC group. To better explain the rationale for this approach and its associated limitations in our manuscript we have modified lines 465-467 with text mentioned previously.

3. The stacked generalization step that involves using predictions of coverage from different child models in a geostatistical model to predict the same coverage is not a principled approach to combining predictions from a statistical standpoint. First, the authors mention at lines 440 – 448 that the selected covariates used in the child models were not collinear. This makes one wonder why the same rule was not applied when fitting the geostatistical model, since predictions from the child models will be highly correlated. Placing the restriction $\beta c = 1$ on the regression parameters will not make this problem go away.

We thank the referee for their comment and provide additional clarifications and references below. To the referee's first point, we confirm that the covariate selection algorithm was not applied to the child models before fitting the geostatistical model. Covariate selection, *via* a variance inflation factor algorithm, before the child model fitting stage was used to improve child model convergence, rather than out of an attempt to improve the interpretability of the covariates. We have clarified the intention behind this method and have provided more details on how it works in Supplementary Information section 1.3.6 (lines 251-259) as follows:

We used a variance inflation factor (VIF) algorithm for geospatial covariate selection before using the subsequent covariates in child models. We chose to use a covariate selection algorithm before fitting child models to remove collinear covariates to facilitate model convergence. For each coordinate point where data is located, geospatial covariate values were extracted. The VIF algorithm works by computing the variance of each covariate, which is conflated when multicollinearities exist in the covariate values. The VIF was computed for each covariate, removing the largest incrementally, until the VIF for each remaining covariate was less than 3. This algorithm was run for all modelling regions (as discussed in Supplementary Information section 1.4.1).

In this manuscript, we use stacked generalization as a way to allow non-linear relationships and complex interactions between covariates and coverage, as the primary objective of this model is predictive accuracy rather than causal inference. Given the limited number of child

models (3) used in the final geostatistical model, no potentially collinear child models needed to be removed in order for the geostatistical model to converge. Removing any of the child models for collinearity would reduce the benefits and diminish the fundamental intuition of using the stacked generalization framework. Even if the child models are correlated, the differences between the results of the three child models – even if minute – can contribute to improved prediction in the stacked generalization process.

We thank the referee for identifying the lack of clarity in our description of the intent behind the restriction on the regression parameters ($\sum \beta_c = 1$). This method is suggested by Bhatt, et al¹⁷ to improve computational tractability rather than reduce correlation. This has been further clarified in the Online Methods section of the manuscript (lines 530-532).

Secondly, this approach does not offer a way of propagating the uncertainties associated with the predictions from the child models to the final models. In a Bayesian context, what the authors have attempted to do is termed Bayesian model averaging, but it has been implemented in the most unusual way.

We agree with the referee that propagation of uncertainty from the child models throughout the modelling process would be ideal in a Bayesian context – the same would hold true for propagation of uncertainty from the underlying covariates themselves. However, propagation of covariate uncertainty is currently computationally infeasible at our modelling scale and therefore represents a source of uncertainty that is not reflected in our final estimates. We agree with the referee that the formal propagation of uncertainty is an important area for future research, and to better communicate this importance, we have added this as a limitation in the manuscript (lines 646 – 649).

Thirdly, the estimated regression coefficients in the geostatistical model are meaningless – these basically relate coverage to coverage. Since parameter estimates from the child models are not presented, it is difficult to understand how useful the covariates were for predicting vaccination coverage.

We have included the stacked generalization step in our modelling process for three primary reasons. First, our primary objective in this manuscript is to create a model with optimal predictive accuracy, rather than to create a model that can be used for causal inference. Second, unlike geostatistical models of processes that can be largely explained through the use of environmental covariates, there are relatively few available spatial covariates that directly predict immunization coverage. Methods to maximize the predictive power of available covariates are therefore of substantial interest to this work. Third, we anticipate complex interactions and non-linear relationships between outcomes and covariates in our model. In geospatial modelling approaches, stacked generalization has been used to account for complex interactions between outcomes and covariates and been shown to increase predictive accuracy^{17,18}, including in other modelled surfaces of vaccine coverage¹⁹. As our objective is predictive accuracy rather than causal inference, we believe that the benefits of inclusion of a stacked generalization step outweigh this limitation.

We have intentionally chosen not to present the specific importance of each underlying covariate in each of the child models, for several reasons. First, these results could be deceiving, as the model is not designed to provide inference about which covariates are most predictive of coverage. Second, the importance of each covariate in the child models

does not reflect the final contribution of that covariate to the ultimate geostatistical model, due to the use of the stacked generalisation framework. We have included the estimated regression coefficients of the child models to illustrate the fit of the geospatial mode, but our original text did not clearly note that these coefficients are not meant to illustrate the relationships between covariates and outcomes. As the referee notes, the inherent trade-off between the models designed for prediction versus inference is an important consideration. To discuss our approach to stacked generalization overall as well as this inherent trade-off between prediction and inference to readers, we have added an additional discussion of this topic in the limitations section of the manuscript (lines 643 - 646).

4. Both the child models and the geostatistical model were fitted using point/cluster-level data. It's difficult to understand why the authors chose to validate the geostatistical model at the first administrative level. Does this mean that the point level estimates were not reliable enough to be used for validation? Another important point here is whether the "observed" admin level one data used in calculating the validation metrics were weighted or not. Since some admin data were converted to point data using the centroids of the admin areas, it is unclear how any sampling weights were applied to such data when generating weighted estimates at the admin level. It will not be ideal to compare the modelled estimates to unweighted admin level one data.

We thank the referee for the comment and have updated our validation framework in response to this comment. As described above, we are not using the centroids of administrative units to assign the areal data and instead are using a k-means clustering algorithm that takes population-weights in consideration when assigning a set of "candidate" points. The "survey" package in R is used to calculate conservative sample size calculations for areal units at lower administrative levels. To further clarify these methods, we have added additional documentation to the Supplementary Information Section 1.3.5 (lines 243– 246), providing additional details on our sample size calculations using the R "survey" package²⁰.

We originally chose to perform model validation at the first administrative level, as that is the geographic level for which most household-based surveys are powered (and the least geographically granular data used in our spatial modelling framework). Validation at the first administrative unit, therefore, has the benefit of comparing the results of the geospatial model to the underlying survey data at the geospatial level for which estimates generated using traditional, non-geospatial survey analysis methods are most valid.

As the referee notes, however, validation at smaller geographic scales may also be of interest to readers. As Global Vaccine Action Plan and Immunization Agenda 2030 targets are set at the district level (approximately the second administrative level in most countries), we have chosen to present the results of our analysis at this level. We have therefore expanded our validation structure to evaluate performance at both the first and second administrative unit levels separately (Online Methods, lines 590 – 595).

In this analysis, we use geospatial modelling techniques primarily as a method to incorporate data available at a variety of spatial scales and produce second-administrative-level estimates that are robust to changes in administrative boundaries over time. Because the sample size for each individual cluster from a survey is small, and because of the potential that some of the point-level variation in coverage is subsequently due to sampling error, our 5x5 km pixel-level estimates are appropriately uncertain (Supplementary Fig. 17). This uncertainty decreases as these geospatial estimates are aggregated up to policy-relevant spatial scales, however. This provides an additional motivation for our decision to present results at the second administrative level, rather than the 5x5 km pixel level.

We therefore have presented validation metrics at the first and second administrative levels, since these correspond to the spatial scale of the results presented in the manuscript. We believe that this approach best allows readers to judge our model's performance in relationship to the results presented in the manuscript and decide at which geographic levels our model results can be used to inform decision-making.

Additional Supplementary Information Tables 10 and 12 and Figs. 11-12 and 15-16 have been added to display second-administrative level holdout results as well as lines 590-592 in the manuscript modified to reflect this addition. Overall, model accuracy was slightly better at the first administrative level than second administrative level but the magnitude of the differences was small. Out of sample validation resulted in mean absolute errors of 0.073 and 0.105, 0.920 and 0.934 95% coverage, and 0.863 and 0.795 correlation, for both first and second administrative unit validation levels respectively in the last year with at least 10 surveys (2017).

Minor comments

1. The geostatistical model is poorly described. It is not clear whether the variables C_i , N_i , and p_i are vectors or scalars. I assume that these are vectors representing time series data for the i th location. This needs to be clarified. A space-time structure is given for the GP model but no temporal index is given in the binomial likelihood and the logit function, etc.

We apologize for any confusion our notation may have caused and have clarified the text of the manuscript and supplement to address this concern. In our original notation, it was unclear that i indexes a cluster of observations of vaccination coverage while also suppressing the implicit space-time-location of that cluster. We have re-written our parameterization in the manuscript (lines 514 – 547) and supplement (lines 294 – 302) to more clearly describe our model, such that each cluster is now indexed as d at location i at time t .

2. How was the restriction $\sum \beta_c = 1$ implemented during model fitting? The prior specified for β_c does not enforce this.

R-INLA has the ability to apply linear constraints during model fit through a series of combinations. We apply a linear constraint to the β_c 's using the "extraconstr" argument of R-INLA²¹, such that:

$$A\beta_c^T = \mathbf{e}^T$$

where A is a matrix of dimensions (1,1, length of vector of the fixed effects (i.e. 3)) and $\mathbf{e} = 1$. We have added these previously missing details in the Online Methods section (lines 530 - 532) to increase transparency and clarity as well as in Supplementary Information section 1.4.5 (lines 331-336).

3. The increases in coverage mentioned in lines 86 – 90, were these significant increases? Knowing whether the increases were significant or not will be of interest to many readers.

In this modelling framework, we sample the posterior distribution of our model 1,000 times, with each draw (or sample) representing one plausible set of estimates for all locations and years in the analysis. The estimated percent of districts with increasing coverage – and associated uncertainty – are then generated by first calculating the percentage of second administrative units with increasing coverage for each of the 1,000 draws. This results in a distribution of 1,000 possible values of the percent of second administrative units that have experienced coverage increases over time. We then summarize this distribution's mean, 2.5th percentile, and 97.5th percentile. This approach is designed to produce an estimate of the percent of districts that have experienced coverage increases over time across all modelled countries. Because of this Bayesian approach, the 95% UIs presented in the manuscript encapsulate the uncertainty around this point estimate.

As the referee points out, however, readers are likely to also be interested in knowing which specific locations have increased or decreased in coverage and with what degree of certainty. Given our Bayesian framework, we have not applied a specific statistical significance cut-off to assess the certainty in the changes in coverage. Following the referee's suggestion, however, we have included an additional figure in the Supplementary Information (Supplementary Information Fig. 1) of the probability of having increased or decreased coverage in each district from 2000 to 2010 and 2010 to 2018 (computed from each district-level posterior sample) to help the reader interpret the uncertainty in estimated changes in coverage over time. We thank the referee for this suggestion -- paired with Figure 2 from the manuscript, this additional supplemental Figure allows readers to assess both the magnitude of estimated changes and the level of certainty about these changes from the model.

4. The 1 x1 km resolution data used to mask out some areas as mentioned in Figure 2 does not match the spatial resolution of the modelled 5 x 5 km surfaces.

We choose to use the land cover surface to mask areas with low population or have been classified as sparsely vegetated or barren terrain. Cartographically, this prevents visual overrepresentation of the coverage estimates in large, sparsely populated areas, for example in Northern Africa, and helps draw the reader's attention to coverage estimates in more-populated locations. We therefore chose a 1x1 km mask as a cartographic strategy, aiming to direct the reader's attention to population areas while preserving as much of the information content in our estimates as possible. As the referee notes, this leaves the spatial resolution of the landcover mask to be of a different scale than the spatial resolution of the modelled estimates. However, the maps presented in all main figures are at the second administrative level (i.e. districts) rather than at the 5x5 km level. For these district-level maps, a 5x5 km mask would also not be at the same spatial resolution as the mapped coverage estimates. The online visualization tool, however, allows for the district level maps to be viewed without the low-population mask if this is of interest to readers.

We have also considered masking districts – rather than 1x1 km pixels -- below a given population threshold. However, this district-level masking strategy is quite limited in districts where populations are extremely concentrated and surrounded by unpopulated areas – again, for instance, in parts of North Africa. If the district falls below the population threshold, the high-population centre is obscured. If the district is above the population threshold, the geographically large but unpopulated areas around the population centre are again visually overrepresented, which contradicts the purpose of the masking strategy. Given that the resolution of the mask does not play a role in the modelling process and instead is a cartographic strategy, we feel that the current approach is the most appropriate.

5. Lines 36-38 – I believe that the 95% effectiveness mentioned relates to valid doses. This needs to be highlighted.

We thank the referee for pointing this out. We have modified the manuscript text (line 36) to reflect that the approximately 93% effectiveness of measles vaccination relates to valid doses to be more accurate.

Referee 2

I have focused upon the spatial data analysis surrounding the Bayesian hierarchical model. These models are widely used in a number of contexts and there is at least one whole book devoted to the subject; see, e.g., Banerjee, Carlin and Gelfand (2014). Hence, the model development is not exactly original, but it is very appropriate for the analysis and the benefits of Bayesian hierarchical modeling have been, and continue to be, well documented in the statistics literature. In this regard, the adoption of these models is commendable.

There are, nevertheless, a few key issues that would benefit from a discussion.

1. The authors claim that they use a great-circle distance for their spatial distance. This may work fine in many specific instances, but there is a technical issue here that the authors should take a closer look at. The most commonly used spatial covariance functions, including the Matérn, are theoretically valid (will yield positive definite covariance matrices) for Euclidean (planar) distances. Great circle distances require specialized covariance functions based upon Legendre polynomials. If the spatial domain of interest is small enough that the curvature of the earth is not prominent, then the great circle distance is well approximated by the Euclidean distance and the use of the former in regular covariance functions is practicable. Otherwise, this could lead to numerical issues. See Banerjee (Biometrics, 2005, <https://doi.org/10.1111/j.1541-0420.2005.00320.x>) for an elaborate discussion. If the authors have not encountered any problems using the great circle distance, then they may keep the current analysis but should recognize the pitfalls in the above reference.

We thank the referee for their thoughtful insight, comments and opportunity for discussion. To the referee's first point, we have chosen to use the great circle distance because it more accurately represents the true surface of the Earth and, as Banerjee²² notes, to avoid introducing anisotropy *via* a planar coordinate system. We use an SPDE approach to allow for efficient computational inference and thus use the Matérn as a solution to the covariance function, which also has the ability to account for a Matérn with great circle representation.

We are accounting for the spherical nature of our spatial mesh while constructing our SPDE components in our INLA modelling framework. We define the SPDE on \mathbb{S}^2 , a spherical manifold, and use the great circle distance as described in the SPDE framework²³. Practically, this is implemented in our modelling pipeline by first converting data locations and geographic region boundaries to three-dimensional coordinates on the unit-sphere. Next, using the INLA "inla.mesh.create" function, we construct an \mathbb{S}^2 mesh using the three-dimensional data and boundaries. We acknowledge this is an important topic and thank the referee for drawing attention to our previously incomplete description. To clarify to readers, we have added a discussion of this in Supplementary Information section 1.4.4 (lines 313-317).

We define the SPDE on \mathbb{S}^2 , a spherical manifold, and use the great circle distance as described in the SPDE framework. Practically, this is implemented in our modelling pipeline by first converting data locations and geographic region boundaries to three-dimensional coordinates on the unit-sphere. Next, using the INLA "inla.mesh.create" function, we construct an \mathbb{S}^2 , mesh using the three-dimensional data and boundaries.

2. The tensor (Kronecker) product structure for spatial-temporal covariance implies the notion of "separability" in space and time. This means that the temporal covariance structure remains the same for each location. Likewise, the spatial covariance structure over the region is the same across time. This has undesirable implications for modeling; see Stein (JASA, 2005; <https://doi.org/10.1198/016214504000000854>) and also Gneiting (JASA, 2002; <https://doi.org/10.1198/016214502760047113>). This issue may not afflict the substantive

inference being pursued here, but, again, should be acknowledged and the papers cited.

We confirm the referee's note that our models are "separable" in space and time. This does pose the assumptions, as noted by Stein and Gneiting, that models are symmetric across space and time and the covariance between at each year and point in space always has the same functional form regardless of the location of the points. Huang et al²⁴ have found that even among data collected from a non-separable process, such as ours, separable models still yielded reasonable predictions. We agree with the referee that we should include a discussion on this in the manuscript and thus have made additions to the manuscript in the limitations section (lines 653-659) as follows:

Our model is separable, yet symmetric, across time and space. This approach assumes that, for each region, the covariance has the same functional form between years and locations regardless of the locations themselves; the use of a non-separable covariance function could relax these assumptions. However, due the additional computational challenges associated with fitting a non-separable model, as well as data sparsity in several regions throughout space and time, we have determined that fitting a non-separable model would be challenging, complex, and likely yield little benefit compared to our current modelling approach.

3. On the same vein, why use AR(1) structure for time? Why not use spatial-temporal covariance functions as in the above references? Or even in a separable model, with time modeled as a Gaussian process using a covariance function?

We thank the referee for their comment and opportunity for discussion; we agree that selecting a temporal structure for geospatial modelling is a complex process. We note that the ultimate temporal pattern in our modelled estimates of coverage depends both on the choice of spatial-temporal covariance structure and on the temporal patterns present in the predictions of coverage from stacked generalization (the "child models" used as predictors in the final geostatistical model). As the referee notes, using spatio-temporal covariance functions could relax some of the assumptions inherent in our separable model. As described above, we have added a discussion of these limitations to the text of the manuscript. Within the limitations of a separable model, as the referee notes, alternative approaches to modelling time could include the use of a one-dimensional Gaussian process, a higher-order autoregressive model, or other, potentially more flexible parameterizations. In practice, however, we find that the AR(1) temporal covariance structure, though simpler, generally allows our models to fit our data appropriately. As a quantitative illustration of this observation, we observe that our out-of-sample predictive validity metrics demonstrate good model performance at both the first and second administrative level, with low bias and reasonable coverage of the 95% predictive interval, across the years included in this study. We have included more information for the readers in the Supplementary Information section 1.4.6 (lines 345 - 352).

4. R-INLA is a fine product for implementing the Bayesian hierarchical models. It avoids MCMC and relies upon Laplace approximation and quadrature for inference. However, MCMC, while slower, does have advantages as it samples from the posterior and the identifiability (or lack thereof) of the parameters is sometimes better reflected. Have the authors considered STAN (<https://mc-stan.org/>) as an alternative to confirm the analysis?

We agree that there are associated limitations with using R-INLA for model fitting and implementation and MCMC methods, like STAN, would be appropriate alternatives to explore. However, INLA has several key advantages when fitting latent Gaussian models compared to a full MCMC sampler, which become particularly important when running models across large spatial scales. First, from a computational standpoint, INLA models leverage an inherent parallelizing structure to enhance multi-core processing. This allows for large dimensional latent fielded models to run in a computationally reasonable time scale, particularly in a distributed computing environment. Second, INLA is able to generalize across latent Gaussian models and as such, a large portion of inference can also be generalized and automated; this also increases the computational tractability of using INLA for these types of large-scale model types. These advantages have led similar spatial research products and teams to also opt to using the R-INLA infrastructure. For example, recent and historical work mapping vaccination coverage by the WorldPop project use similar INLA-based infrastructure for model fitting^{7,25}, albeit at smaller spatial and temporal scales. Also, the DHS program's series of modelled surfaces similarly use methods based in the INLA-SPDE approach¹⁹.

When using INLA, the posterior is numerically approximated and draws are taken from the calculated posterior distribution. We are therefore not able to assess the "convergence" of models in INLA in the same way MCMC sampler convergence is determined, as the referee notes. As an alternative approach, we use the INLA argument "inla_results\$mode\$status" while validating model fit. In response to the referee's comments, and to better discuss the limitations and benefits to the INLA approach with readers, we have added a commentary in Supplementary Information Section 1.4.5 (lines 320-325).

Models were fit using R-INLA as described previously. We have chosen to use R-INLA for model fitting and interpretation to leverage its computational tractability when running models across large spatial scales. This is possible through maximizing computational efficiency via an inherent parallelizing structure to exploit multi-core processing and also generalizing across latent Gaussian models which allows for a significant amount of inference to be automated.

And continuing on Supplementary Information lines 337-342:

The posterior is numerically approximated as part of the INLA approach and thus convergence cannot be assessed in a comparable way as with a full MCMC sampler. As an alternative, we used the R-INLA “`inla_results$mode$mode.status`” function to ensure that the INLA results have identified the posterior modes of the hyperparameters and the associated Hessian is positive definite. This check was passed for all modelled regions.

Referee 3

This manuscript generates estimates of first-dose measles-containing vaccine (MCV1) in low and middle-income countries from 2010-2018. The research topic is an important one and the data are expansive. However, I have difficulty in following the statistical models they authors are using.

We recognize the challenge in summarizing a complex model, such as the one we have presented in this work, in a clear and concise way; we thank the referee for their feedback on how we can improve upon our descriptions of our methodology. As such, we have added additional information on our geostatistical model to our data and modelling flowchart in the Supplementary Information (Supplementary Information Fig. 2), referenced in the manuscript (line 413), to provide an improved visual description of the modelling pipeline, as a reference for readers.

1) First of all, the model input and model outcome are not clearly stated. In the main text, I could not find details of the covariates that are used to fit the models. I appreciate that that may a long list but I think it is important to at least give an overview of these covariates. And only till very end of the main text, I realize that MCV1 is treated as a model outcome. It is also not clear to me why MCV1 is a model outcome rather than a quantity that is observed. The authors should make these very clear at the beginning of the paper. Also, because of the expansiveness of the data and study, the authors need to make their best efforts to present things in a clear and logical way so that readers would not lose track in reading.

We thank the referee for their comment. We have clarified the language around modelling inputs and outputs, including how observations of MCV1 coverage in household-based surveys are used as data inputs and also as our modelled prediction outcome (lines 410 - 429) as follows:

Building from our previous study of diphtheria-tetanus-pertussis vaccination coverage in Africa¹⁴, we fit a geostatistical model with correlated errors across space and time to predict 5x5-km level estimates of MCV1 coverage from 2000 to 2019 using a suite of geospatial and national-level covariates for 101 LMICs. This overall process has been

summarised in Supplementary Fig. 2. We spatially aggregated estimates using population-weighted averages to second administrative units from a modified version of the Database of Global Administrative Units (GADM), referred to as districts, and performed post-hoc analyses to assess geographic inequality to examine progress towards GVAP targets, absolute geographic inequality and vaccination status as a function of geographic remoteness. This study is compliant with the Guidelines for Accurate and Transparent Health Estimates Reporting (GATHER) recommendations (Supplementary Table 1).

We defined routine MCV1 coverage as evidence of receipt of at least one dose of a measles-containing vaccine from either a home-based record (HBR) or parental recall among the target population in concordance with country-specific vaccination schedules in 2019. For countries recommending the first dose of a measles-containing-vaccine within the first year of life, the population used to assess coverage is 12–23 months old. If the first dose is not recommended until the second year of life, the population used to assess coverage is 24–35 months old. For full list of schedules by country, please refer to Supplementary Table 2. Despite our best efforts to remove doses delivered through SIAs (Supplementary Information section 1.3.5), there is likely to be residual misclassification of some SIA doses due to the limitations of the available data, and these estimates of routine coverage should be viewed in the context of this limitation.

As mentioned above, we have added a spatial modelling flowchart in addition to our data processing flow chart to be more articulate about our modelling inputs and intermediary and final outputs. We also have ensured there are explicit references in the Online Methods to the suite of geospatial covariates and to the section in the Supplementary Information regarding which covariates are being used in the child models for each modelled region (lines 483 – 485; Supplementary Table 7).

2) I am not following most of the description of the statistical models and methods under the section “Geostatistical model”. For example in line 456, what are the “optimized covariates” and how did the author choose them? I find most part of the description is too brief and abrupt. I would suggest the author to rewrite the whole section, and, in particular, to give a high-level overview (ideally graphical) of the statistical procedures used.

We appreciate the referee’s suggestions regarding the clarity of the methodological descriptions and opportunity for revision. In response to this feedback, we have made several substantial revisions and added clarifications to the description of the models and methods and would be happy to take additional editorial guidance in this regard.

As mentioned above, we have added an additional visual representation of our modelling process, as suggested, which can be found in the Supplementary Information (Supplementary Information Figure 2). To allow readers to more easily access more detailed descriptions, we

have added direct references in the Online Methods section to relevant sections of the Supplementary Information, particularly to provide additional details on the covariates and covariate selection process (lines 477 – 478 and 485).

In response to the specific feedback of the reviewer regarding the use of the term “optimized covariates”, we recognize that our initial description was unclear. This section has now been rewritten as follows (lines 475-485):

We included twenty-six geospatial covariates as possible predictors of MCV1 coverage in the modelling process, including maternal education, access to major cities or settlements, a binary urban or rural indicator, total population, and a suite of 22 environmental covariates (Supplementary Fig. 6 and Supplementary Table 6). Four national-level covariates were also included: lag-distributed income, prevalence of the completion of the fourth antenatal care visit among pregnant women, mortality due to war and terror, and bias-adjusted national-level administrative data on MCV1 coverage reported through the WHO’s Joint Reporting Form (Supplementary Information Section 1.5.1). For each region, an optimised set of geospatial covariates was selected from these twenty-six possible covariates, using a variance inflation factor (VIF) algorithm where covariates were selected with a $VIF < 3$. This method was used to ensure non-collinearity between covariates within each region, in order to facilitate model convergence. Selected covariates varied by region (Supplementary Table 7).

Additionally, in response to the suggestions of this referee and the other referees, we have made the following revisions to our Online Methods section:

- Additional clarity on our definition of MCV1 coverage and birth year assignments (lines 464 - 468)
- Re-parameterization of our geostatistical model for improved clarity (lines 514 – 547)

In addition to these changes in the Online Methods section, we have also made the following clarifications and adjustments to our Supplementary Information:

- A sensitivity analysis to further discuss our choice in target age cohort inclusion
- Additional clarity on the methods by which we aim to distinguish between doses delivered in SIAs and routine doses in the underlying survey data (lines 206 - 215)
- A more thorough description of stacked generalization and implementation of constraints (lines 277 - 289)
- Additional information on implementation of Matérn for use on the great circle distance (lines 313-317)
- A re-parameterization of our geostatistical model priors to enhance clarity (lines 292 – 302)
- An additional discussion of the rationale behind the use of R-INLA and the ways in which we evaluated proxies for model convergence (lines 320-342)

- A substantially more detailed explanation of areal data resampling method, including an additional figure to explain this process (Supplementary Information Fig. 5, lines 228-248).

We believe these changes have greatly strengthened this manuscript, and we thank the referee for bringing them to our attention.

References

1. Portnoy, A., Jit, M., Helleringer, S. & Verguet, S. Comparative Distributional Impact of Routine Immunization and Supplementary Immunization Activities in Delivery of Measles Vaccine in Low- and Middle-Income Countries. *Value in Health* (2020) doi:10.1016/j.jval.2020.03.012.
2. Cutts, F. T., Lessler, J. & Metcalf, C. J. E. Measles elimination: progress, challenges and implications for rubella control. *Expert Rev Vaccines* **12**, 917–932 (2013).
3. WHO. *Global Vaccine Action Plan 2011-2020*.
http://www.who.int/immunization/global_vaccine_action_plan/GVAP_doc_2011_2020/en/ (2012).
4. WHO and UNICEF warn of a decline in vaccinations during COVID-19. <https://www.who.int/news-room/detail/15-07-2020-who-and-unicef-warn-of-a-decline-in-vaccinations-during-covid-19>.
5. World Health Organization. COVID-19: Operational guidance for maintaining essential health services during an outbreak. <https://www.who.int/publications-detail/covid-19-operational-guidance-for-maintaining-essential-health-services-during-an-outbreak>.
6. Portnoy, A., Jit, M., Helleringer, S. & Verguet, S. Impact of measles supplementary immunization activities on reaching children missed by routine programs. *Vaccine* **36**, 170–178 (2018).
7. Utazi, C. E. *et al.* Geospatial variation in measles vaccine coverage through routine and campaign strategies in Nigeria: Analysis of recent household surveys. *Vaccine* **38**, 3062–3071 (2020).
8. Burstein, R. *et al.* Mapping 123 million neonatal, infant and child deaths between 2000 and 2017. *Nature* **574**, 353–358 (2019).

9. Dwyer-Lindgren, L. *et al.* Mapping HIV prevalence in sub-Saharan Africa between 2000 and 2017. *Nature* **570**, 189–193 (2019).
10. Graetz, N. *et al.* Mapping local variation in educational attainment across Africa. *Nature* **555**, 48–53 (2018).
11. Marquez, N. & Wakefield, J. Harmonizing Child Mortality Data at Disparate Geographic Levels. *arXiv:2002.00089 [stat]* (2020).
12. Wilson, K. & Wakefield, J. Pointless spatial modeling. *Biostatistics* **21**, e17–e32 (2020).
13. Bolivia (Plurinational State of): WHO and UNICEF estimates of immunization coverage: 2019 revision. https://www.who.int/immunization/monitoring_surveillance/data/bol.pdf
14. Zimbabwe: WHO and UNICEF estimates of immunization coverage: 2019 revision. https://www.who.int/immunization/monitoring_surveillance/data/zwe.pdf
15. WHO | Data, statistics and graphics. *WHO* http://www.who.int/immunization/monitoring_surveillance/data/en/.
16. Dansereau, E., Brown, D., Stashko, L. & Danovaro-Holliday, M. C. A systematic review of the agreement of recall, home-based records, facility records, BCG scar, and serology for ascertaining vaccination status in low and middle-income countries. *Gates Open Res* **3**, 923 (2019).
17. Bhatt, S. *et al.* Improved prediction accuracy for disease risk mapping using Gaussian process stacked generalization. *Journal of The Royal Society Interface* **14**, 20170520 (2017).
18. Mosser, J. F. *et al.* Mapping diphtheria-pertussis-tetanus vaccine coverage in Africa, 2000–2016: a spatial and temporal modelling study. *The Lancet* **393**, 1843–1855 (2019).
19. USAID. Interpolation of DHS Survey Data at Subnational Administrative Level 2. <http://dhsprogram.com/pubs/pdf/SAR17/SAR17.pdf>.
20. Package 'survey'. <https://cran.r-project.org/web/packages/survey/survey.pdf>
21. Gómez-Rubio, V. *Chapter 6 Advanced Features | Bayesian inference with INLA*.
22. Banerjee, S. On Geodetic Distance Computations in Spatial Modeling. *Biometrics* **61**, 617–625 (2005).

23. Lindgren, F., Rue, H. & Lindström, J. An explicit link between Gaussian fields and Gaussian Markov random fields: the stochastic partial differential equation approach. *Journal of the Royal Statistical Society: Series B (Statistical Methodology)* **73**, 423–498 (2011).
24. Huang, H.-C., Martinez, F., Mateu, J. & Montes, F. Model comparison and selection for stationary space–time models. *Computational Statistics & Data Analysis* **51**, 4577–4596 (2007).
25. Utazi, C. E. *et al.* High resolution age-structured mapping of childhood vaccination coverage in low and middle income countries. *Vaccine* **36**, 1583–1591 (2018).

Reviewer Reports on the First Revision:

Referee #1 (Remarks to the Author):

Minor comment:

There is a confusion between the definition of MCV1 coverage provided at lines 420-424 and the method used to process MCV1 coverage data discussed at lines 464 - 468. The authors mentioned that for countries where MCV1 is recommended in the first year of life, the population used to assess coverage was children aged 12-23 months (lines 420-424). But then went on to discuss the birth cohort approach which includes older age groups (lines 464-468) - 1 and 2 years older than the target cohort - from each survey. I think that this needs to be clarified in the manuscript.

Referee #2

The authors have addressed most of the concerns I have raised. My only remaining comment is that the authors have missed a lot of references in the statistical literature related to these types of models. They should, at the very least, provide some additional references to note that Bayesian hierarchical models have been widely used in the sciences and they are essentially adopting these methods for their analysis. Two important monographs in this regard are:

Statistics for Spatio-Temporal Data, by Noel Cressie and Christopher K. Wikle. Wiley, Hoboken, NJ, 2011 (588 pp).

Banerjee, Sudipto; Carlin, Bradley P.; Gelfand, Alan E. (2014), Hierarchical Modeling and Analysis for Spatial Data, Second Edition, Monographs on Statistics and Applied Probability (2nd ed.), Chapman and Hall/CRC, ISBN 9781439819173

Perhaps these (and some other related references) should be added to the bibliography. main figures legends should remain where they are.

Author Rebuttals to First Revision: (please note that the author has quoted the referees in black and responded in blue)

Referee #1 (Remarks to the Author):

Minor comment:

There is a confusion between the definition of MCV1 coverage provided at lines 420-424 and the method used to process MCV1 coverage data discussed at lines 464 - 468. The authors mentioned that for countries where MCV1 is recommended in the first year of life, the population used to assess coverage was children aged 12-23 months (lines 420-424). But then went on to discuss the birth cohort approach which includes older age groups (lines 464-468) - 1 and 2 years older than the target cohort - from each survey. I think that this needs to be clarified in the manuscript.

We thank the referee for this opportunity for clarification. We agree the previous language was confusing and as such, have first removed the references to the handling of age cohorts from the original lines 420-424, which now read in lines 401-406 as follows:

We defined routine MCV1 coverage as evidence of receipt of at least one dose of a measles-containing vaccine from either a home-based record (HBR) or parental recall among the target population in concordance with country-specific vaccination schedules in 2019⁵⁵. Despite our best efforts to remove doses delivered through SIAs (Supplementary Information section 1.3.5), there is likely to be residual misclassification of some SIA doses due to the limitations of the available data, and these estimates of routine coverage should be viewed in the context of this limitation.

We have consolidated and simplified our description of the treatment of age cohorts in the original lines 464-468 to read as follows in lines 441-447:

Individual age, in months, at the time of survey collection was used to assign each child to a birth cohort (12–23 months, 24–35 months, 36–47 months, and 48–59 months). Data corresponding to each birth cohort were included in the modelling process in the year in which that birth cohort was aged 0–12 months old. For countries recommending the first dose of a measles-containing-vaccine within the first year of life, we included data from children aged 12-47 months. If the first dose was not recommended until the second year of life, we included data from children aged 24-59 months. For full list of schedules by country, please refer to Supplementary Table 2. This yielded a dataset of 1,697,570 total children.

Referee #2

The authors have addressed most of the concerns I have raised. My only remaining comment is that the authors have missed a lot of references in the statistical literature related to these types of models. They should, at the very least, provide some additional references to note that Bayesian hierarchical models have been widely used in the sciences and they are essentially adopting these methods for their analysis.

Two important monographs in this regard are:

- Statistics for Spatio-Temporal Data, by Noel Cressie and Christopher K. Wikle. Wiley, Hoboken, NJ, 2011 (588 pp).
- Banerjee, Sudipto; Carlin, Bradley P.; Gelfand, Alan E. (2014), Hierarchical Modeling and Analysis for Spatial Data, Second Edition, Monographs on Statistics and Applied Probability (2nd ed.), Chapman and Hall/CRC, ISBN 9781439819173

Perhaps these (and some other related references) should be added to the bibliography.

We thank the referee for their comment We agree that additional references to the statistical literature related to our underlying methods will help to establish that the use of Bayesian hierarchical models such as ours fits within a robust body of previous scientific inquiry. As such, we have added the two references noted by the referee above to the online methods section (line 501) where we describe our model formulation in more detail.